# A Compositional Atlas for Algebraic Circuits

**Benjie Wang**
University of California, Los Angeles
benjiewang@ucla.edu

**Denis Deratani Mauá**
University of São Paulo
ddm@ime.usp.br

**Guy Van den Broeck**
University of California, Los Angeles
guyvdb@cs.ucla.edu

**YooJung Choi**
Arizona State University
yj.choi@asu.edu

## Abstract

Circuits based on sum-product structure have become a ubiquitous representation to compactly encode knowledge, from Boolean functions to probability distributions. By imposing constraints on the structure of such circuits, certain inference queries become tractable, such as model counting and most probable configuration. Recent works have explored analyzing probabilistic and causal inference queries as compositions of basic operators to derive tractability conditions. In this paper, we take an *algebraic* perspective for *compositional inference*, and show that a large class of queries—including marginal MAP, probabilistic answer set programming inference, and causal backdoor adjustment—correspond to a combination of basic operators over semirings: aggregation, product, and elementwise mapping. Using this framework, we uncover simple and general sufficient conditions for tractable composition of these operators, in terms of circuit properties (e.g., marginal determinism, compatibility) and conditions on the elementwise mappings. Applying our analysis, we derive novel tractability conditions for many such compositional queries. Our results unify tractability conditions for existing problems on circuits, while providing a blueprint for analysing novel compositional inference queries.

## 1 Introduction

Circuit-based representations, such as Boolean circuits, decision diagrams, and arithmetic circuits, are of central importance in many areas of AI and machine learning. For example, a primary means of performing inference in many models, from Bayesian networks [16, 9] to probabilistic programs [20, 24, 26, 43], is to convert them into equivalent circuits; this is commonly known as *knowledge compilation*. Inference via knowledge compilation has also been used for many applications in neuro-symbolic AI, such as constrained generation [2, 54] and neural logic programming [34, 28]. Circuits can also be *learned* as probabilistic generative models directly from data [25, 41, 40, 32], in which context they are known as probabilistic circuits [11]. Compared with neural generative models, probabilistic circuits enjoy tractable evaluation of inference queries such as marginal probabilities, which has been used for tasks such as fair machine learning [12] and causal reasoning [53, 50, 49].

The key feature of circuits is that they enable one to precisely characterize *tractability conditions* (structural properties of the circuit) under which a given *inference query* can be computed exactly and efficiently. One can then enforce these circuit properties when compiling or learning a model to enable tractable inference. For many basic inference queries, such as computing a marginal probability, tractability conditions are well understood [48, 8]. However, for more complex queries, the situation is less clear, and the exercise of deriving algorithms and tractability conditions for a given query has usually been carried out in an instance-specific manner requiring significant effort.

38th Conference on Neural Information Processing Systems (NeurIPS 2024).

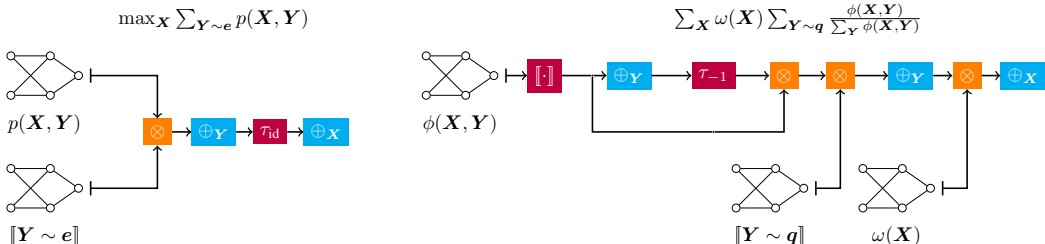

Figure 1: Example applications of our compositional inference framework for *(Left)* MMAP and *(Right)* Success Probability in Prob. Logic Programing under the Stable Model semantics (MaxEnt).

In Figure 1, we illustrate two such queries. The marginal MAP (MMAP) [13] query takes a probabilistic circuit $p$ and some evidence $e$ and asks for the most likely assignment of a subset of variables. The success probability inference in probabilistic logic programming [6, 45] takes a circuit representation $\phi$ of a logic program, a weight function $\omega$ and some query $q$, and computes the probability of the query under the program's semantics (MaxEnt, in the example). At first glance, these seem like very different queries, involving different types of input circuits (logical and probabilistic), and different types of computations. However, they share similar *algebraic structure*: logical and probabilistic circuits can be interpreted as circuits defined over different *semirings*, while maximization and summation can be viewed as *aggregation* over different semirings. In this paper, inspired by the compositional atlas for probabilistic circuits [48], we take a *compositional* approach to algebraic inference problems, breaking them down into a series of basic operators: aggregation, product, and elementwise mapping. For example, the MMAP and probabilistic logic programming queries involve multiple interleaved aggregations and products, along with one elementwise mapping each. Given a circuit algorithm (and associated tractability condition) for each basic operator, we can reuse these algorithms to construct algorithms for arbitrary compositions. The key challenge is then to check if each intermediate circuit satisfies the requisite tractability conditions.

Our contributions can be summarized as follows. We introduce a compositional inference framework for *algebraic* circuits (Section 3) over arbitrary semirings, generalizing existing results on logical [18] and probabilistic [48] circuits. In particular, we provide a language for specifying inference queries involving *different* semirings as a composition of basic operators (Section 3.1). We then prove sufficient conditions for the tractability of each basic operator (Section 3.2) and novel conditions for composing such operators (Section 3.3). We apply our compositional framework to a number of inference problems (Section 4), showing how our compositional approach leads to more systematic derivation of tractability conditions and algorithms, and in some cases improved complexity analysis. In particular, we discover a tractability hierarchy for inference queries captured under the 2AMC framework [29], and reduce the complexity of causal backdoor/frontdoor adjustment on probabilistic circuits [38, 49] from quadratic/cubic to linear/quadratic respectively.

## 2 Preliminaries

**Notation** We use capital letters (e.g., $X, Y$) to denote variables and lowercase for assignments (values) of those variables (e.g., $x, y$). We use boldface to denote sets of variables/assignments (e.g., $\boldsymbol{X}, \boldsymbol{y}$) and write $\text{Assign}(\boldsymbol{V})$ for the set of all assignments to $\boldsymbol{V}$. Given a variable assignment $\boldsymbol{v}$ of $\boldsymbol{V}$, and a subset of variables $\boldsymbol{W} \subseteq \boldsymbol{V}$, we write $\boldsymbol{v}_{\boldsymbol{W}}$ to denote the assignment of $\boldsymbol{W}$ corresponding to $\boldsymbol{v}$.

**Semirings** In this paper, we consider inference problems over commutative *semirings*. Semirings are sets closed w.r.t. operators of addition ($\oplus$) and multiplication ($\otimes$) that satisfy certain properties:

**Definition 1** (Commutative Semiring). *A commutative semiring $\mathcal{S}$ is a tuple $(S, \oplus, \otimes, 0_\mathcal{S}, 1_\mathcal{S})$, where $\oplus$ and $\otimes$ are associative and commutative binary operators on a set $S$ (called the* domain*) such that $\otimes$ distributes over $\oplus$ (i.e., $a \otimes (b \oplus c) = (a \otimes b) \oplus (a \otimes c)$ for all $a, b, c \in S$); $0_\mathcal{S} \in S$ is the additive identity (i.e., $0_\mathcal{S} \oplus a = a$ for all $a \in S$) and annihilates $S$ through multiplication (i.e., $0_\mathcal{S} \otimes a = 0$ for all $a \in S$); and $1_\mathcal{S} \in S$ is the multiplicative identity (i.e., $1_\mathcal{S} \otimes a = a$ for all $a \in S$).*

For example, the probability semiring $\mathcal{P} = (\mathbb{R}_{\geq 0}, +, \cdot, 0, 1)$ employs standard addition and multiplication ($\oplus = +$ and $\otimes = \cdot$) over the non-negative reals, the $(\max, \cdot)$ semiring $\mathcal{M} = (\mathbb{R}_{\geq 0}, \max, \cdot, 0, 1)$

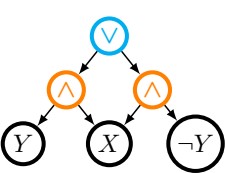
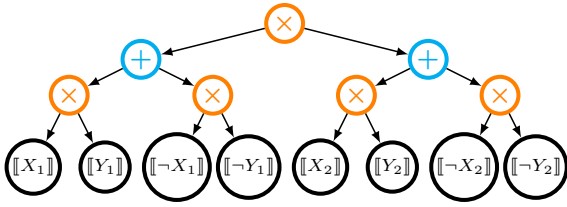

(a) A Boolean circuit that is smooth, decompos-
able, deterministic, but not $\{X\}$-deterministic.

(b) A probabilistic circuit that is smooth, decomposable, and
$\{X_1, X_2\}$-deterministic. $\llbracket \cdot \rrbracket$ maps $\top$ to 1 and $\bot$ to 0.

Figure 2: Examples of Algebraic Circuits. We use ⬤, ⬤, ⬤ to represent input, sum and product nodes respectively.

replaces addition with maximization, while the Boolean semiring $\mathcal{B} = (\{\bot, \top\}, \vee, \wedge, \bot, \top)$ employs disjunction and conjunction operators ($\oplus = \vee$ and $\otimes = \wedge$) over truth values.

**Algebraic Circuits**    We now define the concept of an algebraic circuit, which are computational graph-based representations of functions taking values in an arbitrary semiring.

**Definition 2** (Algebraic Circuit)**.** *Given a semiring $\mathcal{S} = (S, \oplus, \otimes, 0_{\mathcal{S}}, 1_{\mathcal{S}})$, an algebraic circuit $C$ over variables $\boldsymbol{V}$ is a rooted directed acyclic graph (DAG), whose nodes $\alpha$ have the following syntax:*

$$\alpha ::= l \mid +_{i=1}^k \alpha_i \mid \times_{i=1}^k \alpha_i \,,$$

*where $\alpha_i \in C$ are circuit nodes, $k \in \mathbb{N}^{>0}$ and $l : Assign(\boldsymbol{W}) \to S$ is a function over a (possibly empty) subset $\boldsymbol{W} \subseteq \boldsymbol{V}$ of variables, called its* scope*. That is, each circuit node may be an input ($l$), sum (+), or a product ($\times$). The scope of any internal node is defined to be $\mathrm{vars}(\alpha) := \cup_{i=1}^k \mathrm{vars}(\alpha_i)$. Each node $\alpha$ represents a function $p_\alpha$ taking values in $S$, defined recursively by: $p_\alpha(\boldsymbol{w}) ::= l(\boldsymbol{w})$ if $\alpha = l$, $p_\alpha(\boldsymbol{w}) ::= \oplus_{i=1}^k p_{\alpha_i}(\boldsymbol{w})$ if $\alpha = +_{i=1}^k \alpha_i$, and $p_\alpha(\boldsymbol{w}) ::= \otimes_{i=1}^k p_{\alpha_i}(\boldsymbol{w})$ if $\times_{i=1}^k \alpha_i$, where $\boldsymbol{W}$ is the scope of $\alpha$. The function $p_C$ represented by the circuit is defined to be the function of the root node. The size $|C|$ of a circuit is defined to be the number of edges in the DAG.*

For simplicity, we will restrict to circuits with binary products (i.e. $k = 2$ for products); this can be enforced with at most a linear increase in size. Prominent examples of algebraic circuits include negation normal forms (NNF) and binary decision diagrams [4]—which are over the Boolean semiring and represent Boolean functions—and probabilistic circuits [11]—which are over the probabilistic semiring and represent probability distributions.[1] By imposing simple restrictions on the circuit, which we call *circuit properties*, various inference queries that are computationally hard in general become tractable. In particular, smoothness and decomposability ensure tractable marginal inference:

**Definition 3** (Smoothness, Decomposability)**.** *A circuit is* smooth *if for every sum node $\alpha = +_i \alpha_i$, its children have the same scope: $\forall i, j$, $\mathrm{vars}(\alpha_i) = \mathrm{vars}(\alpha_j)$. A circuit is* decomposable *if for every product node $\alpha = \alpha_1 \times \alpha_2$, its children have disjoint scopes: $\mathrm{vars}(\alpha_1) \cap \mathrm{vars}(\alpha_2) = \emptyset$.*

Aside from the scopes of circuit nodes, we can also specify properties relating to their *supports* [11]:

**Definition 4** ($\boldsymbol{X}$-Support)**.** *Given a partition $(\boldsymbol{X}, \boldsymbol{Y})$ of variables $\boldsymbol{V}$ and a node $\alpha$ in circuit $C$, the $\boldsymbol{X}$-support of $\alpha$ is the projection of its support on $\boldsymbol{X}$:*

$$supp_{\boldsymbol{X}}(\alpha) = \{\boldsymbol{x} \in Assign(\boldsymbol{X} \cap \mathrm{vars}(\alpha)) : \exists \boldsymbol{y} \in Assign(\mathrm{vars}(\alpha) \setminus \boldsymbol{X}) \text{ s.t. } p_\alpha(\boldsymbol{x}, \boldsymbol{y}) \neq 0_{\mathcal{S}}\}.$$

**Definition 5** ($\boldsymbol{X}$-Determinism)**.** *Given a circuit $C$ and a partition $(\boldsymbol{X}, \boldsymbol{Y})$ of $\boldsymbol{V}$, we say that $C$ is $\boldsymbol{X}$-deterministic if for all sum nodes $\alpha = +_{i=1}^k \alpha_i$, either: (i) $\mathrm{vars}(\alpha) \cap \boldsymbol{X} = \emptyset$; or (ii) $supp_{\boldsymbol{X}}(\alpha_i) \cap supp_{\boldsymbol{X}}(\alpha_j) = \emptyset$ for all $i \neq j$.*

$\boldsymbol{X}$-determinism refers to a family of properties indexed by sets $\boldsymbol{X}$. In particular $\boldsymbol{V}$-determinism is usually referred to simply as determinism. Note that, as defined, scope and support, and thus these circuit properties, apply to any semiring: the scope only depends on the variable decomposition of the circuit, while the support only refers to scope and the semiring additive identity $0_{\mathcal{S}}$. Figure 2a shows a simple example of a smooth, decomposable, and deterministic circuit that is not $X$-deterministic, while Figure 2b shows a smooth, decomposable, and $\{X_1, X_2\}$-deterministic circuit.

---

[1]Probabilistic circuits are sometimes written with weights on the edges; this can easily be translated to our formalism by replacing the child of a weighted edge with a product of itself and an input function with empty scope corrresponding to the weight [44, 42].

# 3 Compositional Inference: A Unifying Approach

Many inference problems can be written as *compositions of basic operators*, which take as input one or more functions and output another function. For example, the marginal MAP query on probability distributions $\max_{\boldsymbol{x}} \sum_{\boldsymbol{y}} p(\boldsymbol{x}, \boldsymbol{y})$ is a composition of the $\sum$ and $\max$ operators. Similarly, for Boolean functions $\phi, \psi$, the query $\sum_{\boldsymbol{x}} \exists \boldsymbol{y}. \phi(\boldsymbol{x}, \boldsymbol{y}) \wedge \psi(\boldsymbol{x}, \boldsymbol{y})$ composes the $\sum, \exists$ and $\wedge$ operators. Although these queries appear to involve four different operators, three of them ($\sum, \max, \exists$) can be viewed as an *aggregation* operation over *different* semirings. Thus, we begin this section by consolidating to a simple set of three operators applicable to functions taking values in some semiring: namely, aggregation, product, and elementwise mapping (Section 3.1).

Equipped with this language for specifying compositional inference queries, we then move on to analyzing their tractability when the input functions are given as circuits. The thesis of this paper is that algebraic structure is often the right level of abstraction to derive useful sufficient (and sometimes necessary) conditions for tractability. We firstly show *tractability conditions* of each of the basic operators (Section 3.2), before deriving *composability conditions* showing how circuit properties are maintained through operators (Section 3.3). This enables us to systematically derive conditions for the input circuits that enable efficient computation of a compositional inference query. Algorithms and detailed proofs of all theorems can be found in Appendix A.

## 3.1 Basic Operators

**Aggregation**   Given a function $f : \text{Assign}(\boldsymbol{V}) \to S$, *aggregating $f$ over $\boldsymbol{W} \subseteq \boldsymbol{V}$* returns the function $f' : \text{Assign}(\boldsymbol{Z}) \to S$ for $\boldsymbol{Z} = \boldsymbol{V} \setminus \boldsymbol{W}$ defined by $f'(\boldsymbol{z}) := \bigoplus_{\boldsymbol{w}} f(\boldsymbol{z}, \boldsymbol{w})$.

For example, aggregation corresponds to forgetting variables $\boldsymbol{W}$ in the Boolean semiring, marginalizing out $\boldsymbol{W}$ in the probability semiring, and maximizing over assignments in the $(\max, \cdot)$ semiring. Next, some queries, such as divergence measures between probability distributions, take two functions as input, and many others involve combining two or more intermediate results, as is the case in probabilistic answer set programming inference and causal backdoor/frontdoor queries. We define the product operator to encapsulate such "combination" of functions in general.

**Product**   Given two functions $f : \text{Assign}(\boldsymbol{W}) \to S$ and $f' : \text{Assign}(\boldsymbol{W}') \to S$, the *product of $f$ and $f'$* is a function $f'' : \text{Assign}(\boldsymbol{V}) \to S$, where $\boldsymbol{V} = \boldsymbol{W} \cup \boldsymbol{W}'$, defined by $f''(\boldsymbol{v}) := f(\boldsymbol{v}_{\boldsymbol{W}}) \otimes f'(\boldsymbol{v}_{\boldsymbol{W}'})$.

For example, a product corresponds to the conjoin operator $\wedge$ in the Boolean semiring, and standard multiplication $\cdot$ in the probability semiring. Lastly, we introduce the *elementwise mapping* operator, defined by a mapping $\tau$ from a semiring to a (possibly different) semiring. When applied to a function $f$, it returns the function composition $\tau \circ f$. This is the key piece that distinguishes our framework from prior analysis of sum-of-product queries over specific semirings, allowing us to express queries such as causal inference and probabilistic logic programming inference under the same framework.

**Elementwise Mapping**   Given a function $f : \text{Assign}(\boldsymbol{V}) \to S$ and a mapping $\tau : S \to S'$ from semiring $\mathcal{S}$ to $\mathcal{S}'$ satisfying $\tau(0_{\mathcal{S}}) = 0_{\mathcal{S}'}$, an *elementwise mapping of $f$ by $\tau$* results in a function $f' : \text{Assign}(\boldsymbol{V}) \to S'$ defined by $f'(\boldsymbol{v}) := \tau(f(\boldsymbol{v}))$.[2]

In practice, we use elementwise mappings as an abstraction predominantly for two purposes. The first is for switching between semirings, while the second is to map between elements of the same semiring. For the former, one of the most important elementwise mappings we will consider is the *support mapping*, which maps between any two semirings as follows.

**Definition 6** (Support Mapping)**.** *Given a source semiring $\mathcal{S}$ and a target semiring $\mathcal{S}'$, the support mapping $\llbracket \cdot \rrbracket_{\mathcal{S} \to \mathcal{S}'}$ is defined as: $\llbracket a \rrbracket_{\mathcal{S} \to \mathcal{S}'} = 0_{\mathcal{S}'}$ if $a = 0_{\mathcal{S}}$; $\llbracket a \rrbracket_{\mathcal{S} \to \mathcal{S}'} = 1_{\mathcal{S}'}$ otherwise.*

In particular we will often use the source semiring $\mathcal{S} = \mathcal{B}$, in which case the support mapping maps $\bot$ to the $0_{\mathcal{S}'}$ and $\top$ to the $1_{\mathcal{S}'}$ in the target semiring. This is useful for encoding a logical function for inference in another semiring, e.g. probabilistic inference in the probabilistic semiring.

---

[2]In a slight abuse of notation, we will write $\tau : \mathcal{S} \to \mathcal{S}'$ to indicate that $\tau$ maps between the respective sets.

**Example 1** (Marginal MAP). *Suppose that we are given a Boolean formula $\phi(\boldsymbol{X}, \boldsymbol{Y})$ and a weight function $w : Assign(\boldsymbol{X} \cup \boldsymbol{Y}) \to \mathbb{R}_{\geq 0}$. The marginal MAP query for variables $\boldsymbol{X}$ is defined by*

$$\mathrm{MMAP}(\phi, \omega) = \max_{\boldsymbol{x}} \sum_{\boldsymbol{y}} \phi(\boldsymbol{x}, \boldsymbol{y}) \cdot \omega(\boldsymbol{x}, \boldsymbol{y}),$$

*where we interpret $\top$ as 1 and $\bot$ as 0. We can break this down into a compositional query as follows:*

$$\bigoplus_{\boldsymbol{x}} \tau_{id, \mathcal{P} \to \mathcal{M}} \left[ \bigoplus_{\boldsymbol{y}} [\![\phi(\boldsymbol{x}, \boldsymbol{y})]\!]_{\mathcal{B} \to \mathcal{P}} \otimes \omega(\boldsymbol{x}, \boldsymbol{y}) \right].$$

*The support mapping ensures $\phi$ and $\omega$ are both functions over the probabilistic semiring, so that we can apply the product operation. Notice also the inclusion of an identity mapping $\tau_{id, \mathcal{P} \to \mathcal{M}}$ from the probability to the $(\max, \cdot)$ semiring defined by $\tau_{id, \mathcal{P} \to \mathcal{M}}(x) = x$ for all $x \in \mathbb{R}_{\geq 0}$. While differentiating between semirings over the same domain may seem superfluous, the explicit identity operator will become important when we analyze the tractability of these compositions on circuits.*

### 3.2 Tractability Conditions for Basic Operators

We now consider the tractability of applying each basic operator to circuits: that is, computing a circuit whose function corresponds to the result of applying the operator to the functions given by the input circuit(s). First, it is well known that forgetting and marginalization of any subset of variables can be performed in polynomial time if the input circuits in the respective semirings (NNF and PC) are smooth and decomposable [18, 11]. This can be generalized to arbitrary semirings:

**Theorem 1** (Tractable Aggregation). *Let $C$ be a smooth and decomposable circuit representing a function $p : Assign(\boldsymbol{V}) \to S$. Then for any $\boldsymbol{W} \subseteq \boldsymbol{V}$, it is possible to compute the aggregate as a smooth and decomposable circuit $C'$ (i.e., $p_{C'}(\boldsymbol{Z}) = \bigoplus_{\boldsymbol{w}} p_C(\boldsymbol{Z}, \boldsymbol{w})$) in $O(|C|)$ time and space.*

Next, let us consider the product operator. In the Boolean circuits literature, it is well known that the conjoin operator can be applied tractably if the circuits both follow a common structure known as a *vtree* [17]. In [48] a more general property known as *compatibility* was introduced that directly specifies conditions with respect to two (probabilistic) circuits, without reference to a vtree. We now define a generalization of this property ($\boldsymbol{X}$-compatibility) and also identify a new condition ($\boldsymbol{X}$-support-compatibility) that enables tractable products.

**Definition 7** ($\boldsymbol{X}$-Compatibility). *Given two smooth and decomposable circuits $C, C'$ over variables $\boldsymbol{V}, \boldsymbol{V}'$ respectively, and a variable set $\boldsymbol{X} \subseteq \boldsymbol{V} \cap \boldsymbol{V}'$, we say that $C, C'$ are $\boldsymbol{X}$-compatible if for every product node $\alpha = \alpha_1 \times \alpha_2 \in C$ and $\alpha' = \alpha_1' \times \alpha_2' \in C'$ such that $\mathrm{vars}(\alpha) \cap \boldsymbol{X} = \mathrm{vars}(\alpha') \cap \boldsymbol{X}$, the scope is partitioned in the same way, i.e. $\mathrm{vars}(\alpha_1) \cap \boldsymbol{X} = \mathrm{vars}(\alpha_1') \cap \boldsymbol{X}$ and $\mathrm{vars}(\alpha_2) \cap \boldsymbol{X} = \mathrm{vars}(\alpha_2') \cap \boldsymbol{X}$. We say that $C, C'$ are compatible if they are $(\boldsymbol{V} \cap \boldsymbol{V}')$-compatible.*

Intuitively, compatibility states that the scopes of the circuits decompose in the same way at product nodes. Compatibility of two circuits suffices to be able to tractably compute their product:

**Theorem 2** (Tractable Product - Compatibility). *Let $C, C'$ be compatible circuits over variables $\boldsymbol{V}, \boldsymbol{V}'$, respectively, and the same semiring. Then it is possible to compute their product as a circuit $C$ compatible with them (i.e., $p_{C''}(\boldsymbol{V} \cup \boldsymbol{V}') = p_C(\boldsymbol{V}) \otimes p_{C'}(\boldsymbol{V}')$) in $O(|C||C'|)$ time and space.*

We remark that if we are given a fully factorized function $f(\boldsymbol{V}) = \bigotimes_{V_i \in \boldsymbol{V}} f_i(V_i)$, this can be arranged as a circuit (series of binary products) compatible with any other decomposable circuit; thus, we say this type of function is *omni-compatible*. We also say that a circuit is *structured decomposable* if it is compatible with itself. Now, our more general definition of $\boldsymbol{X}$-compatibility states that the scopes of the circuits *restricted to $\boldsymbol{X}$* decompose in the same way at product nodes. This will be important when we consider composing products with other operators, such as aggregation. The following result shows that compatibility w.r.t. a subset is a weaker condition:

**Proposition 1** (Properties of $\boldsymbol{X}$-Compatibility). *If two circuits $C, C'$ are $\boldsymbol{X}$-compatible, then they are $\boldsymbol{X}'$-compatible for any subset $\boldsymbol{X}' \subseteq \boldsymbol{X}$.*

Compatibility is a sufficient but not necessary condition for tractable products. Some non-compatible circuits can be efficiently *restructured* to be compatible, such that we can then apply Theorem 2; we refer readers to [55] for details. Alternatively, it is also known that deterministic circuits can be multiplied with themselves in linear time, even when they are not structured decomposable [48, 27]. We formalize this idea with a new property that we call *support-compatibility*.

**Definition 8** ($X$-Support Compatibility). *Given two smooth and decomposable circuits $C, C'$ over variables $V, V'$ respectively, and a set of variables $X \subseteq V \cap V'$, let $C[X], C'[X]$ be the DAGs obtained by restricting to nodes with scope overlapping with $X$. We say that $C, C'$ are $X$-support-compatible if there is an isomorphism $\iota$ between $C[X], C'[X]$ such that: (i) for any node $\alpha \in C[X]$, $\mathrm{vars}(\alpha) \cap X = \mathrm{vars}(\iota(\alpha)) \cap X$; (ii) for any sum node $\alpha \in C[X]$, $supp_X(\alpha_i) \cap supp_X(\iota(\alpha_j)) = \emptyset$ whenever $i \neq j$. We say that $C, C'$ are support-compatible if they are $(V \cap V')$-support-compatible.*

To unpack this definition, we note that any smooth, decomposable, and $X$-deterministic circuit is $X$-support-compatible with itself, with the obvious isomorphism. However, this property is more general in that it allows for circuits over different sets of variables and does not require that the nodes represent exactly the same function; merely that the sum nodes have "compatible" support decompositions. As we will later see, the significance of this property is that it can be often maintained through applications of operators, making it useful for compositions.

**Theorem 3** (Tractable Product - Support Compatibility). *Let $C, C'$ be support-compatible circuits over variables $V, V'$, respectively, and the same semiring. Then, given the isomorphism $\iota$, it is possible to compute their product as a smooth and decomposable circuit $C''$ support-compatible with them (i.e., $p_{C''}(V \cup V') = p_C(V) \otimes p_{C'}(V')$) in $O(\max(|C|, |C'|))$ time and space.*

We now examine the tractability of general elementwise mappings $\tau : \mathcal{S} \to \mathcal{S}'$ on a circuit $C$. It is tempting here to simply construct a new circuit $C'$ over the semiring $\mathcal{S}'$ with the same structure as $C$, and replace each input function $l$ in the circuit with $\tau(l)$. However, the resulting circuit $p_{C'}(V)$ is not guaranteed to correctly compute $\tau(p_C(V))$ in general. For example, consider the support mapping $[\![ \cdot ]\!]_{\mathcal{B} \to \mathcal{S}}$—which maps $\perp$ to $0_{\mathcal{S}}$ and $\top$ to $1_{\mathcal{S}}$ —for the probability semiring $\mathcal{S} = (\mathbb{R}_{\geq 0}, +, \cdot, 0, 1)$. Then the transformation of the smooth and decomposable circuit $C = X \vee X$ produces $C' = [\![ X ]\!] + [\![ X ]\!]$, which evaluates to $p_{C'}(X = \top) = 2$ whereas $\tau(p_C(X = \top)) = 1$. In order for this simple algorithm to be correct, we need to impose certain conditions on the elementwise mapping $\tau$ and/or the circuit $C$ it is being applied to.

**Theorem 4** (Tractable Mapping). *Let $C$ be a smooth and decomposable circuit over semiring $\mathcal{S}$, and $\tau : \mathcal{S} \to \mathcal{S}'$ a mapping such that $\tau(0_{\mathcal{S}}) = 0_{\mathcal{S}'}$. Then it is possible to compute the mapping of $C$ by $\tau$ as a smooth and decomposable circuit $C'$ (i.e., $p_{C'}(V) = \tau(p_C(V))$) in $O(|C|)$ time and space if $\tau$ distributes over sums and over products.*

*$\tau$ distributes over sums if: either **(Additive)** $\tau$ is an additive homomorphism, i.e. $\tau(a \oplus b) = \tau(a) \oplus \tau(b)$; or **(Det)** $C$ is deterministic.*

*$\tau$ distributes over products if: either **(Multiplicative)** $\tau$ is an multiplicative homomorphism, i.e. $\tau(a \otimes b) = \tau(a) \otimes \tau(b)$; or **(Prod 0/1)** $\tau(1_{\mathcal{S}}) = 1_{\mathcal{S}'}$, and for all product nodes $\alpha = \alpha_1 \times \alpha_2 \in C$, and for every value $v \in Assign(\mathrm{vars}(\alpha))$, either $p_{\alpha_1}(v_{\mathrm{vars}(\alpha_1)}) \in \{0_{\mathcal{S}}, 1_{\mathcal{S}}\}$ or $p_{\alpha_2}(v_{\mathrm{vars}(\alpha_2)}) \in \{0_{\mathcal{S}}, 1_{\mathcal{S}}\}$.*

We can apply Theorem 4 to immediately derive the following property of support mappings:

**Corollary 1** (Support Mapping). *Given a circuit $C$ over a semiring $\mathcal{S}$ and any target semiring $\mathcal{S}'$, a circuit representing $[\![ p_C ]\!]_{\mathcal{S} \to \mathcal{S}'}$ can be computed tractably if (i) $\mathcal{S}$ satisfies $a \oplus b = 0_{\mathcal{S}} \implies a = b = 0_{\mathcal{S}}$ and $\mathcal{S}'$ is idempotent (i.e., $1_{\mathcal{S}'} \oplus 1_{\mathcal{S}'} = 1_{\mathcal{S}'}$), or (ii) $C$ is deterministic.*

*Proof.* First note that $[\![ \cdot ]\!]_{\mathcal{S} \to \mathcal{S}'}$ satisfies (Multiplicative), and thus distributes over products. If (i) holds, consider $[\![ a \oplus b ]\!]_{\mathcal{S} \to \mathcal{S}'}$. If $a = b = 0_{\mathcal{S}}$, then this is equal to $[\![ 0_{\mathcal{S}} ]\!]_{\mathcal{S} \to \mathcal{S}'} = [\![ a ]\!]_{\mathcal{S} \to \mathcal{S}'} + [\![ b ]\!]_{\mathcal{S} \to \mathcal{S}'} = 0_{\mathcal{S}'}$; otherwise $a, b, a \oplus b \neq 0_{\mathcal{S}}$ and $[\![ a \oplus b ]\!]_{\mathcal{S} \to \mathcal{S}'} = [\![ a ]\!]_{\mathcal{S} \to \mathcal{S}'} \oplus [\![ b ]\!]_{\mathcal{S} \to \mathcal{S}'} = 1_{\mathcal{S}'}$ (by idempotence of $\mathcal{S}'$). Thus $[\![ \cdot ]\!]_{\mathcal{S} \to \mathcal{S}'}$ satisfies (Additive). Alternatively, if (ii) holds, then (Det) holds. In either case $[\![ \cdot ]\!]_{\mathcal{S} \to \mathcal{S}'}$ distributes over sums in the circuit. $\square$

The following examples illustrate the generality of elementwise mappings and Theorem 4:

**Example 2** (Partition Function and MAP). *Given a probability distribution $p(V)$, consider the task of computing the partition function $\sum_v p(v)$ and MAP $\max_v p(v)$. These can be viewed as aggregations over the probability and $(\max, \cdot)$ semirings respectively.*

*$p$ is often either a probabilistic circuit $C_{prob}$, or a combination of a Boolean circuit $C_{bool}$ and weights $w$ (in weighted model counting). In the former case, the partition function is tractable because the circuit is already over the probability semiring, while in the latter case, MAP is tractable because the $\mathcal{S}' = (\max, \cdot)$ semiring is idempotent so $[\![ C_{bool} ]\!]_{\mathcal{B} \to \mathcal{S}'}$ is tractable. On the other hand, the partition*

Table 1: Tractability Conditions for Operations on Algebraic Circuits. Sm: Smoothness, Dec: Decomposability; $X$-Det(erminism), $X$-Cmp: $X$-Compatibility, $X$-SCmp: $X$-Support-Compatibility.

| | Conditions | If the Input Circuit(s) are ... | | | Complexity |
| | | $X$-Det | $X$-Cmp w/ $C_{\text{other}}$ | $X$-SCmp w/ $C_{\text{other}}$ | |
| | | Then the Output Circuit is ... | | (A.4) | |
|---|---|---|---|---|---|
| **Aggr. ($W$)** | Sm, Dec | $X$-Det if $W \cap X = \emptyset$ | $X$-Cmp w/ $C_{\text{other}}$ if $W \cap X = \emptyset$ | $X$-SCmp w/ $C_{\text{other}}$ if $W \cap X = \emptyset$ | $O(|C|)$ (A.1) |
| **Product** | Cmp | $X$-Det | $X$-Cmp w/ $C_{\text{other}}$ | N/A | $O(|C||C'|)$ (A.2.1) |
| | SCmp | $X$-Det | $X$-Cmp w/ $C_{\text{other}}$ | $X$-SCmp w/ $C_{\text{other}}$ | $O(\max(|C|, |C'|))$ (A.2.2) |
| **Elem. Mapping** | Sm, Dec, (Add/Det), (Mult/Prod01) | $X$-Det | $X$-Cmp w/ $C_{\text{other}}$ | $X$-SCmp w/ $C_{\text{other}}$ | $O(|C|)$ (A.3) |

*function for Boolean circuits and MAP for PCs require determinism for the conditions of Theorem 4 to hold; in fact, these problems are known to be NP-hard without determinism [18, 39].*

**Example 3** (Power Function in Probability Semiring). *For the probability semiring $S = S' = (\mathbb{R}_{\geq 0}, +, \cdot, 0, 1)$, consider the power function $\tau_\beta(a) := \begin{cases} a^\beta & \text{if } a \neq 0 \\ 0 & \text{if } a = 0 \end{cases}$ for some $\beta \in \mathbb{R}$. This mapping satisfies (Multiplicative), and is tractable if we enforce (Det) on the circuit.*

It is worth noting that semiring homomorphisms (i.e. additive and multiplicative) are always tractable. In the case when $S = S' = \mathcal{P}$, it was shown in [48] that the only such mapping is the identity function. However this is not the case for other semirings: the power function $\tau_\beta$ is an example in the $(\max, \cdot)$ semiring. To summarize, we have shown sufficient tractability conditions for aggeregation, products, and elementwise mappings. Notice that the conditions for aggregation and products only depend on variable scopes and supports, and as such apply to any semiring; in contrast, for elementwise mappings, we take advantage of specific properties of the semiring(s) in question.

### 3.3 Tractable Composition of Operators

We now analyze compositions of these basic operators. As such, we need to consider not only circuit properties that enable tractability, but how these properties are maintained through each operator, so that the output circuit can be used as input to another operator. We call these *composability conditions*. In all cases, the output circuit is smooth and decomposable. Thus, we focus on the properties of $X$-determinism, $X$-compatibility, and $X$-support-compatibility. We emphasize that these are not singular properties, but rather families of properties indexed by a variable set $X$. We present the intuitive ideas behind our results below, while deferring full proofs to the Appendix.

**Theorem 5** (Composability Conditions). *The results in Table 1 hold.*

**$X$-determinism** Intuitively, $X$-determinism is maintained through products because the resulting sum nodes partition the $X$-support in a "finer" way to the original circuits, and through elementwise mappings since they do not expand the support of any node (since $\tau(0_S) = 0_{S'}$). For aggregation, the $X$-support is maintained if aggregation does not occur over any of the variables in $X$.

**$X$-compatibility** Here, we are interested in the following question: if the input circuit(s) to some operator are $X$-compatible with some other circuit $C_{\text{other}}$ for any fixed $X$, is the same true of the output of the operator? $X$-compatibility with $C_{\text{other}}$ is maintained through aggregation because it weakens the condition (by Proposition 1) and through elementwise mapping as it does not change variable scopes. As for taking the product of circuits, the output circuit will maintain similar variable partitionings at products, such that it remains $X$-compatible with $C_{\text{other}}$. Notably, this result does *not* hold for compatibility where the scope $X$ may be different for each pair of circuits under consideration; we show a counterexample in Example 4 in the Appendix.

**$X$-support-compatibility** $X$-support-compatibility is maintained through elementwise mappings and aggregation (except on $X$) for similar reasons to $X$-determinism. For products, the result retains a similar $X$-support structure, so $X$-support compatibility is maintained.

We conclude by remarking that, once we determine that a compositional query is tractable, then one immediately obtains a correct algorithm for computing the query by application of the generic

Table 2: Tractability Conditions and Complexity for Compositional Inference Problems. We denote new results with an asterisk.

| | Problem | Tractability Conditions | Complexity |
|---|---|---|---|
| **2AMC** | PASP (Max-Credal)* | Sm, Dec, $\boldsymbol{X}$-Det | $O(|C|)$ |
| | PASP (MaxEnt)*, MMAP | Sm, Dec, Det, $\boldsymbol{X}$-Det | $O(|C|)$ |
| | SDP* | Sm, Dec, Det, $\boldsymbol{X}$-Det, $\boldsymbol{X}$-First | $O(|C|))$ |
| **Causal Inference** | Backdoor* | Sm, Dec, SD, $(\boldsymbol{X} \cup \boldsymbol{Z})$-Det | $O(|C|^2)$ |
| | | Sm, Dec, $\boldsymbol{Z}$-Det, $(\boldsymbol{X} \cup \boldsymbol{Z})$-Det | $O(|C|)$ |
| | Frontdoor* | Sm, Dec, SD, $\boldsymbol{X}$-Det, $(\boldsymbol{X} \cup \boldsymbol{Z})$-Det | $O(|C|^2)$ |
| **Other** | MFE* | Sm, Dec, $\boldsymbol{H}$-Det, $\boldsymbol{I}^-$-Det, $(\boldsymbol{H} \cup \boldsymbol{I}^-)$-Det | $O(|C|)$ |
| | Reverse-MAP | Sm, Dec, $\boldsymbol{X}$-Det | $O(|C|)$ |

algorithms for aggregation, product, and elementwise mapping (see Appendix A). An upper bound on the complexity (attained by the algorithm) is also given by considering the complexities of each individual operator; in particular, the algorithm is polytime for a bounded number of operators.

## 4 Case Studies

In this section, we apply our compositional framework to analyze the tractability of several different problems involving circuits found in the literature (Table 2). Some of the results are known, but can now be cast in a general framework (with often simpler proofs). We also present new results, deriving tractability conditions that are less restrictive than reported in existing literature.

**Theorem 6** (Tractability of Compositional Queries). *The results in Table 2 hold.*

### 4.1 Algebraic Model Counting

In algebraic model counting [30] (a generalization of weighted model counting), one is given a Boolean function $\phi(\boldsymbol{V})$, and a fully-factorized labeling function $\omega(\boldsymbol{V}) = \bigotimes_{V_i \in \boldsymbol{V}} \omega_i(V_i)$ in some semiring $\mathcal{S}$, and the goal is to aggregate these labels for all satisfying assignments of $\phi$. This can be easily cast in our framework as $\bigoplus_{\boldsymbol{v}} \big( \llbracket (\phi(\boldsymbol{v})) \rrbracket_{\mathcal{B} \to \mathcal{S}} \otimes \omega(\boldsymbol{v}) \big)$. Here, the support mapping $\llbracket \cdot \rrbracket_{\mathcal{B} \to \mathcal{S}}$ transfers the Boolean function to the semiring $\mathcal{S}$ over which aggregation occurs. Assuming that $\phi(\boldsymbol{V})$ is given as a smooth and decomposable Boolean circuit (DNNF), then by Corollary 1 AMC is tractable if $\mathcal{S}$ is idempotent or if the circuit is additionally deterministic (note that $\omega(\boldsymbol{V})$ is omni-compatible, so the product is tractable); this matches the results of [30].

**2AMC** A recent generalization of algebraic model counting is the 2AMC (second-level algebraic model counting) problem [29], which encompasses a number of important bilevel inference problems such as marginal MAP and inference in probabilistic answer set programs. Given a partition of the variables $\boldsymbol{V} = (\boldsymbol{X}, \boldsymbol{Y})$, a Boolean function $\phi(\boldsymbol{X}, \boldsymbol{Y})$, *outer* and *inner* semirings $\mathcal{S}_{\boldsymbol{X}}, \mathcal{S}_{\boldsymbol{Y}}$, labeling functions $\omega_{\boldsymbol{Y}}(\boldsymbol{Y}) = \bigotimes_{Y_i \in \boldsymbol{Y}} \omega_{\boldsymbol{Y},i}(Y_i)$ over $\mathcal{S}_{\boldsymbol{Y}}$ and $\omega_{\boldsymbol{X}}(\boldsymbol{X}) = \bigotimes_{X_i \in \boldsymbol{X}} \omega_{\boldsymbol{X},i}(X_i)$ over $\mathcal{S}_{\boldsymbol{X}}$, and an elementwise mapping $\tau_{\mathcal{S}_{\boldsymbol{Y}} \to \mathcal{S}_{\boldsymbol{X}}} : \mathcal{S}_{\boldsymbol{Y}} \to \mathcal{S}_{\boldsymbol{X}}$, the 2AMC problem is given by:

$$\bigoplus_{\boldsymbol{x}} \left( \tau_{\mathcal{S}_{\boldsymbol{Y}} \to \mathcal{S}_{\boldsymbol{X}}} \left( \bigoplus_{\boldsymbol{y}} \llbracket \phi(\boldsymbol{x}, \boldsymbol{y}) \rrbracket_{\mathcal{B} \to \mathcal{S}_{\boldsymbol{Y}}} \otimes \omega(\boldsymbol{y}) \right) \otimes \omega'(\boldsymbol{x}) \right) \tag{1}$$

To tackle this type of bilevel inference problem, [29] identified a circuit property called $\boldsymbol{X}$-firstness.

**Definition 9** ($\boldsymbol{X}$-Firstness). *Suppose $C$ is a circuit over variables $\boldsymbol{V}$ and $(\boldsymbol{X}, \boldsymbol{Y})$ a partition of $\boldsymbol{V}$. We say that a node $\alpha \in C$ is $\boldsymbol{X}$-only if $\mathrm{vars}(\alpha) \subseteq \boldsymbol{X}$, $\boldsymbol{Y}$-only if $\mathrm{vars}(\alpha) \subseteq \boldsymbol{Y}$, and* mixed *otherwise. Then we say that $C$ is $\boldsymbol{X}$-first if for all product nodes $\alpha = \alpha_1 \times \alpha_2$, we have that either: (i) each $\alpha_i$ is $\boldsymbol{X}$-only or $\boldsymbol{Y}$-only; (ii) or exactly one $\alpha_i$ is mixed, and the other is $\boldsymbol{X}$-only.*

It was stated in [29] that smoothness, decomposability, determinism, and $\boldsymbol{X}$-firstness suffice to ensure tractable computation of 2AMC problems, by simply evaluating the circuit in the given semirings (caching values if necessary). We now show that this is neither sufficient nor necessary in general. To build intuition, consider the simple NNF circuit $\phi(X, Y) = (X \wedge Y) \vee (X \wedge \neg Y)$. Note that $\phi$ trivially satisfies $X$-firstness and is smooth, decomposable, and deterministic. Let $\mathcal{S}$

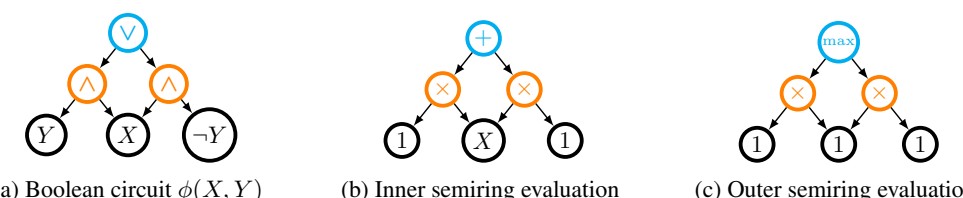

| (a) Boolean circuit $\phi(X, Y)$ | (b) Inner semiring evaluation | (c) Outer semiring evaluation |

Figure 3: Failure case of 2AMC algorithm on smooth, decomposable, X-first circuit.

be the probability semiring, $\mathcal{S}'$ be the $(\max, \cdot)$-semiring, labeling functions be $\omega(y) = \omega(\neg y) = 1$, $\omega'(x) = \omega'(\neg x) = 1$, and the mapping function be the identity $\tau(a) = a$. Then, noting that the labels are the multiplicative identity 1, the 2AMC value is $\max_X \tau(\sum_Y [\![\phi(X, Y)]\!]_{\mathcal{B} \to \mathcal{S}}) = \max(\tau([\![\phi(x, y)]\!]_{\mathcal{B} \to \mathcal{S}} + [\![\phi(x, \neg y)]\!]_{\mathcal{B} \to \mathcal{S}}), \tau([\![\phi(\neg x, y)]\!]_{\mathcal{B} \to \mathcal{S}} + [\![\phi(\neg x, \neg y)]\!]_{\mathcal{B} \to \mathcal{S}})) = \max(\tau(1 + 1), \tau(0)) = 2$. On the other hand, the algorithm of [29] returns the value 2AMC $= 1$, as shown in Figure 3. This is not just a flaw in the specific evaluation algorithm, but rather a provable intractability of the problem given these properties:

**Theorem 7** (Hardness of 2AMC with $X$-firstness). *2AMC is #P-hard, even for circuits that are smooth, decomposable, deterministic, and $X$-first, and a constant-time elementwise mapping.*

Analyzing using our compositional framework, the issue is that the tractability conditions for $\tau$ do not hold; whilst the Boolean circuit is deterministic, this is not true once $Y$ is aggregated. In fact, we show that also enforcing $X$-determinism suffices to tractably compute arbitrary 2AMC instances.

**Theorem 8** (Tractability Conditions for 2AMC). *Every 2AMC instance is tractable in $O(|C|)$ time for Boolean circuits that are smooth, decomposable, deterministic, $X$-first, and $X$-deterministic.*

*Proof sketch.* The key point to notice is that the elementwise mapping relative to the transformation of inner to outer semiring operates over an aggregation of an $X$-first and $X$-deterministic circuit, obtained by the product of a Boolean function (mapped to the inner semiring by a support mapping) and a weight function of $Y$. Hence, it satisfies (Det) and (Prod 0/1): all of the $X$-only children of a product node are 0/1 valued (in the inner semiring). $\square$

For specific instances of 2AMC, depending on the semirings $\mathcal{S}, \mathcal{S}'$ and mapping function $\tau$, we also find that it is possible to remove the requirement of $X$-firstness or $(V)$-determinism, as we summarize in Table 2. One might thus wonder if there is a difference in terms of compactness between requiring $X$-determinism and $X$-firstness, as opposed to $X$-determinism alone. For example, for sentential decision diagrams (SDD) [17], a popular knowledge compilation target, these notions coincide: a SDD is $X$-deterministic iff it is $X$-first (in which context this property is known as $X$-constrainedness [37, 22]). However, as shown in Figure 2b, there exist $X$-deterministic but not $X$-first circuits. We now show that $X$-deterministic circuits can be exponentially more succinct than $X$-deterministic circuits that are additionally $X$-first, as the size of $X$ grows.[3]

**Theorem 9** (Exponential Separation). *Given sets of variables $X = \{X_1, ..., X_n\}, Y = \{Y_1, ..., Y_n\}$, there exists a smooth, decomposable and $X$-deterministic circuit $C$ of size $poly(n)$ such that the smallest smooth, decomposable, and $X$-first circuit $C'$ such that $p_C \equiv p_{C'}$ has size $2^{\Omega(n)}$.*

Thus, to summarize, some instances of 2AMC can be solved efficiently when $\phi$ is smooth, decomposable and $X$-deterministic. A larger number of instances can be solved when additionally, $\phi$ is deterministic; and all 2AMC problems are tractable if we also impose $X$-firstness.

## 4.2 Causal Inference

In causal inference, one is often interested in computing *interventional distributions*, denoted using the $do(\cdot)$ operator, as a function of the observed distribution $p$. This function depends on the causal graph linking the variables, and can be derived using the do-calculus [38]. For example, the well-known *backdoor* and *frontdoor* graphs induce the following formulae:

$$p(\boldsymbol{y}|do(\boldsymbol{x})) = \sum_{\boldsymbol{z}} p(\boldsymbol{z})p(\boldsymbol{y}|\boldsymbol{x}, \boldsymbol{z}), \qquad (2)$$

---

[3]If the size of $X$ is fixed, a circuit can always be rearranged to be $X$-first with at most a $2^{|X|}$ blowup.

$$p(\boldsymbol{y}|do(\boldsymbol{x})) = \sum_{\boldsymbol{z}} p(\boldsymbol{z}|\boldsymbol{x}) \sum_{\boldsymbol{x'}} p(\boldsymbol{x'})p(\boldsymbol{y}|\boldsymbol{x'}, \boldsymbol{z}). \tag{3}$$

Assuming that the observed joint distribution $p(\boldsymbol{X}, \boldsymbol{Y}, \boldsymbol{Z})$ is given as a probabilistic circuit $C$, we consider the problem of obtaining a probabilistic circuit $C'$ over variables $\boldsymbol{X} \cup \boldsymbol{Y}$ representing $p(\boldsymbol{Y}|do(\boldsymbol{X}))$. Tractability conditions for the backdoor/frontdoor cases were derived by [49], with quadratic/cubic complexity respectively. However, we observe that in some cases we can avoid the requirement of structured decomposability and/or obtain reduced complexity relative to their findings.

In the backdoor case, it is known that structured decomposability and $(\boldsymbol{X} \cup \boldsymbol{Z})$-determinism suffices for a quadratic time algorithm. This can be seen by decomposing into a compositional query:

$$\bigoplus_{\boldsymbol{z}} \left( \left( \bigoplus_{\boldsymbol{x},\boldsymbol{y}} p(\boldsymbol{v}) \right) \otimes p(\boldsymbol{v}) \otimes \tau_{-1}\left( \bigoplus_{\boldsymbol{y}} p(\boldsymbol{v}) \right) \right). \tag{4}$$

where $\boldsymbol{V} = (\boldsymbol{X}, \boldsymbol{Y}, \boldsymbol{Z})$, and $\tau_{-1}(a) = \begin{cases} a^{-1} & \text{if } a \neq 0 \\ 0 & \text{if } a = 0 \end{cases}$. Assuming $(\boldsymbol{X} \cup \boldsymbol{Z})$-determinism and structured decomposability, then $\tau_{-1}\left(\bigoplus_{\boldsymbol{y}} p(\boldsymbol{V})\right)$ is tractable by (Det) and (Multiplicative), the product $p(\boldsymbol{V}) \otimes \tau_{-1}\left(\bigoplus_{\boldsymbol{y}} p(\boldsymbol{V})\right)$ by support-compatibility, and the final product by compatibility. However, if we additionally have $\boldsymbol{Z}$-determinism, then the final product becomes tractable by support compatibility. This has linear rather than quadratic complexity, and does not require the circuit to be structured decomposable. In the frontdoor case, [49] showed that $\boldsymbol{X}$-determinism, $(\boldsymbol{X} \cup \boldsymbol{Z})$-determinism, and structured decomposability suffices for cubic complexity. However, we note that under such conditions, the inner product $p(\boldsymbol{X'}) \otimes p(\boldsymbol{Y}|\boldsymbol{X'}, \boldsymbol{Z})$ is tractable by support-compatibility. As such, the complexity of this query is actually quadratic rather than cubic as previously shown. We summarize our findings in Table 2 and refer the reader to the Appendix for full proofs.

## 5 Related Work

Our work builds upon the observation that many inference problems can be characterized as a composition of basic operators. Prior works have considered compositional inference for circuits in the Boolean [18] and probabilistic semirings [48, 49], deriving tractability conditions for operators specific to these semirings. Aside from generalizing to arbitrary semirings, we also introduce extended composability conditions that enable interleaving of aggregation, products, and mappings. Meanwhile, algebraic model counting [30] deals (implicitly) with mappings from the Boolean semiring to an arbitrary semiring, but does not consider compositional queries. Closest to our work, [29] consider a generalization of algebraic model counting that allows for an additional semiring translation; however, this still assumes input Boolean circuits and has incomplete tractability characterizations. Our framework resolves these limitations, permitting arbitrary compositional queries over semirings.

Many works have considered (unbounded) sums-of-products queries on arbitrary semirings [21, 5, 1, 23], encompassing many important problems such as constraint satisfaction problems [7], graphical model inference [56], and database queries [52], which are often computationally hard in the worst-case. Algorithms for such queries often utilize compact intermediate representations and/or assume compact input representations, such as circuits [35, 17, 36, 3]. Our framework focuses on queries where the number of operators is bounded, and characterizes conditions under which inference is tractable in polynomial time. It also includes elementwise mappings as a key additional abstraction that can be used to express queries involving more than sums and products.

## 6 Conclusion

In summary, we have introduced a framework for analysing compositional inference problems on circuits, based on algebraic structure. In doing so, we were able to derive new tractability conditions and simplified algorithms for a number of existing problems, including 2AMC and causal inference. Our framework focuses on simple and composable *sufficient* tractability conditions for aggregations, products and elementwise mappings operators; a limitation of this generality is these conditions may not be necessary for specific queries on specific semirings. Our work motivates the development of knowledge compilation and learning algorithms that target the requisite circuit properties, such as $\boldsymbol{X}$-determinism. Finally, while we focus on exact inference, for many problems (e.g. marginal MAP) approximate algorithms exist and are of significant interest; an interesting direction for future work is to investigate if these can be also be generalized using the compositional approach.

## Acknowledgements

We thank Antonio Vergari for helpful discussions, and acknowledge him for proposing an early version of support compatibility and Theorem 3, and for pointing out a potential reduction in complexity for the causal inference queries. This work was done in part while the authors were visiting the Simons Institute for the Theory of Computing. This work was funded in part by the DARPA ANSR program under award FA8750-23-2-0004, the DARPA PTG Program under award HR00112220005, and NSF grant #IIS-1943641. DM received generous support from the IBM Corporation, the Center for Artificial Intelligence at University of São Paulo (C4AI-USP), the São Paulo Research Foundation (FAPESP grants #2019/07665-4 and 2022/02937-9), the Brazilian National Research Council (CNPq grant no. 305136/2022-4) and CAPES (Finance Code 001). YC was partially supported by a gift from Cisco University Research Program.

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

---

**Algorithm 1:** AGG

---

**Input:** Smooth and decomposable algebraic circuit $C(\boldsymbol{V})$; node $\alpha \in C$; Subset of variables $\boldsymbol{W} \subseteq \text{vars}(\alpha)$

**Output:** Node encoding $\bigoplus_{\boldsymbol{W}} p_\alpha(\boldsymbol{V})$

1 **if** $\alpha$ is input node **then**
2 $\quad$ **return** AGG-INPUT$(\alpha; \boldsymbol{W})$
3 **else if** $\alpha$ is product or sum node and vars$(\alpha) = \boldsymbol{W}$ **then**
4 $\quad$ **return** NEWNODE$(\otimes_{i=1}^{k} p_{\text{AGG}(\alpha_i; \boldsymbol{W} \cap \text{vars}(\alpha_i))})$ **if** $\alpha$ is product **else** NEWNODE$(\oplus_{i=1}^{k} p_{\text{AGG}(\alpha_i; \boldsymbol{W})})$
5 **else if** $\alpha$ is product or sum node and $\boldsymbol{W} \subset \text{vars}(\alpha)$ **then**
6 $\quad$ **return** $\times_{i=1}^{k} \text{AGG}(\alpha_i; \boldsymbol{W} \cap \text{vars}(\alpha_i))$ **if** $\alpha$ is product **else** $+_{i=1}^{k} \text{AGG}(\alpha_i; \boldsymbol{W})$

---

---

**Algorithm 2:** PROD-CMP

---

**Input:** Compatible algebraic circuits $C(\boldsymbol{V}), C'(\boldsymbol{V}')$; nodes $\alpha \in C, \alpha' \in C'$ s.t. $\text{vars}(\alpha) \cap (\boldsymbol{V} \cap \boldsymbol{V}') = \text{vars}(\alpha') \cap (\boldsymbol{V} \cap \boldsymbol{V}')$

**Output:** Node encoding $p_C(\boldsymbol{V}) \otimes p_{C'}(\boldsymbol{V}')$

1 **if** $\text{vars}(\alpha) \cap \text{vars}(\alpha') = \emptyset$ **then**
2 $\quad$ **return** $\alpha \times \alpha'$
3 **else if** $\alpha$ is a product or input node and $\alpha' = +_{j=1}^{k'}$ is a sum node **then**
4 $\quad$ **return** $+_{j=1}^{k'} \text{PROD-CMP}(\alpha, \alpha'_j)$
5 **else if** $\alpha, \alpha'$ are input nodes **then**
6 $\quad$ **return** PROD-INPUT$(\alpha, \alpha')$
7 **else if** $\alpha = \alpha_1 \times \alpha_2, \alpha' = \alpha'_1 \times \alpha'_2$ are product nodes **then**
8 $\quad$ **return** PROD-CMP$(\alpha_1, \alpha'_1) \times$ PROD-CMP$(\alpha_2, \alpha'_2)$
9 **else if** $\alpha = +_{i=1}^{k}\alpha_i, \alpha' = +_{j=1}^{k'}\alpha'_j$ are sum nodes **then**
10 $\quad$ **return** $+_{i=1}^{k} +_{j=1}^{k'} \text{PROD-CMP}(\alpha_i, \alpha'_j)$

---

## A Algorithms and Proofs

In Algorithms 1-4 we present algorithms for the aggregation, product (with compatibility), product (with support-compatiblity), and elementwise mapping operators respectively (the initial call is to the root of the circuit(s)). In the following, we present proofs that the algorithms soundly compute smooth and decomposable output circuits for the respective operators.

### A.1 Tractable Aggregation

**Theorem 1** (Tractable Aggregation). *Let $C$ be a smooth and decomposable circuit representing a function $p : \text{Assign}(\boldsymbol{V}) \to S$. Then for any $\boldsymbol{W} \subseteq \boldsymbol{V}$, it is possible to compute the aggregate as a smooth and decomposable circuit $C'$ (i.e., $p_{C'}(\boldsymbol{Z}) = \bigoplus_{\boldsymbol{w}} p_C(\boldsymbol{Z}, \boldsymbol{w})$) in $O(|C|)$ time and space.*

*Proof.* We prove this inductively, starting from the input nodes of the circuit. Our claim is that for each node $\alpha \in C$, AGG$(\alpha; \boldsymbol{W})$ (Algorithm 1) returns a node $\alpha'$ with scope vars$(\alpha') = \text{vars}(\alpha) \setminus \boldsymbol{W}$ such that $p_{\alpha'}(\text{vars}(\alpha')) = \bigoplus_{\boldsymbol{w}} p_\alpha(\text{vars}(\alpha))$, and is decomposable (if product) and smooth (if sum).

If $\alpha$ is an input node (Lines 1-2), then this is possible by assumption; we denote this with AGG-INPUT in the algorithm. Note that if vars$(\alpha) = \boldsymbol{W}$, then this is just a scalar/constant (i.e. input node with empty scope).

---

**Algorithm 3:** `PROD-SCMP`

---

**Input:** Support-compatible algebraic circuits $C(\mathbf{V}), C'(\mathbf{V}')$; nodes $\alpha \in C, \alpha' \in C'$ s.t.
$\iota(\alpha) = \alpha'$
**Output:** Circuit encoding $p_C(\mathbf{V}) \otimes p_{C'}(\mathbf{V}')$

1 **if** $\text{vars}(\alpha) \cap \text{vars}(\alpha') = \emptyset$ **then**
2    |   **return** $\alpha \times \alpha'$
3 **else if** $\alpha, \alpha'$ are input nodes **then**
4    |   **return** `PROD-INPUT`$(\alpha, \alpha')$
5 **else if** $\alpha = \alpha_1 \times \alpha_2, \alpha' = \alpha'_1 \times \alpha'_2$ are product nodes **then**
6    |   **return** `PROD-SCMP`$(\alpha_1, \alpha'_1) \times$ `PROD-SCMP`$(\alpha_2, \alpha'_2)$
7 **else if** $\alpha = +_{i=1}^{k}\alpha_i, \alpha' = +_{i=1}^{k}\alpha'_i$ are sum nodes **then**
8    |   **return** $+_{i=1}^{k}$`PROD-SCMP`$(\alpha_i, \alpha'_i)$

---

---

**Algorithm 4:** `MAPPING`

---

**Input:** Smooth and decomposable algebraic circuit $C(\mathbf{V})$ over semiring $\mathcal{S}$; Node $\alpha \in C$;
Mapping function $\tau : \mathcal{S} \to \mathcal{S}'$
**Output:** Node encoding $\tau(p_C(\mathbf{V}))$

1 **if** $\alpha$ is input node **then**
2    |   **return** `MAPPING-INPUT`$(\alpha; \tau)$
3 **else if** $\alpha$ is product or sum node **then**
4    |   **return** $\otimes_{i=1}^{k}$`MAPPING`$(\alpha_i; \tau)$ **if** $\alpha$ is product **else** $\oplus_{i=1}^{k}$`MAPPING`$(\alpha_i; \tau)$

---

If $\alpha$ is a product node $\alpha_1 \times \alpha_2$, then by decomposability, $\mathbf{W} \cap \text{vars}(\alpha_1)$ and $\mathbf{W} \cap \text{vars}(\alpha_2)$ partition $\mathbf{W}$. Thus we have that:

$$
\begin{aligned}
\bigoplus_{\boldsymbol{w}} p_\alpha(\text{vars}(\alpha)) &= \bigoplus_{\boldsymbol{w}} \big(p_{\alpha_1}(\text{vars}(\alpha_1)) \otimes p_{\alpha_2}(\text{vars}(\alpha_2))\big) \\
&= \bigoplus_{\boldsymbol{w}\cap\text{vars}(\alpha_1)} \bigoplus_{\boldsymbol{w}\cap\text{vars}(\alpha_2)} \big(p_{\alpha_1}(\text{vars}(\alpha_1)) \otimes p_{\alpha_2}(\text{vars}(\alpha_2))\big) \\
&= \left(\bigoplus_{\boldsymbol{w}\cap\text{vars}(\alpha_1)} p_{\alpha_1}(\text{vars}(\alpha_1))\right) \otimes \left(\bigoplus_{\boldsymbol{w}\cap\text{vars}(\alpha_2)} p_{\alpha_2}(\text{vars}(\alpha_2))\right) \\
&= p_{\text{AGG}(\alpha_1;\mathbf{W}\cap\text{vars}(\alpha_1))}(\text{vars}(\alpha_1) \setminus \mathbf{W}) \otimes p_{\text{AGG}(\alpha_2;\mathbf{W}\cap\text{vars}(\alpha_2))}(\text{vars}(\alpha_2) \setminus \mathbf{W})
\end{aligned}
$$

The second equality follows by the partition (and associativity of the addition and multiplication), while the third follows by distributivity of multiplication over addition. In the case where $\text{vars}(\alpha) = \mathbf{W}$ (Lines 3-4), then $p_{\text{AGG}(\alpha_i;\mathbf{W}\cap\text{vars}(\alpha_i))}(\text{vars}(\alpha_i))$ is just a scalar for each $i$, so we can directly perform this computation, returning a new scalar node $\alpha'$. Otherwise (Lines 5-6), we construct a new product node $\alpha' = \alpha'_1 \times \alpha'_2 = \text{AGG}(\alpha_1; \mathbf{W} \cap \text{vars}(\alpha_1)) \times \text{AGG}(\alpha_2; \mathbf{W} \cap \text{vars}(\alpha_2))$. By the inductive hypothesis, $\alpha'_i$ has scope $\text{vars}(\alpha'_i) = \text{vars}(\alpha_i) \setminus \mathbf{W}$, so $\alpha'$ is clearly decomposable and has scope $\text{vars}(\alpha') = (\text{vars}(\alpha_1) \setminus \mathbf{W}) \cup (\text{vars}(\alpha_2) \setminus \mathbf{W}) = \text{vars}(\alpha) \setminus \mathbf{W}$.

If $\alpha = +_{i=1}^{k} \alpha_i$ is a sum node, then we note that by smoothness, $\text{vars}(\alpha_i) = \text{vars}(\alpha)$ for all $i$. Thus we have that:

$$\bigoplus_{\boldsymbol{w}} p_\alpha(\text{vars}(\alpha)) = \bigoplus_{\boldsymbol{w}} \bigoplus_{i=1}^{k} p_{\alpha_i}(\text{vars}(\alpha))$$

$$= \bigoplus_{i=1}^{k} \bigoplus_{\boldsymbol{w}} p_{\alpha_i}(\text{vars}(\alpha))$$

$$= \bigoplus_{i=1}^{k} \bigoplus_{\boldsymbol{w}} p_{\alpha_i}(\text{vars}(\alpha_i))$$

$$= \bigoplus_{i=1}^{k} p_{\text{AGG}(\alpha_i; \boldsymbol{W})}(\text{vars}(\alpha_i))$$

In the case where $\text{vars}(\alpha) = \boldsymbol{W}$ (Lines 3-4), then $p_{\text{AGG}(\alpha_i; \boldsymbol{W})}(\text{vars}(\alpha_i))$ is just a scalar, so we can directly perform this computation, returning a new scalar node $\alpha'$. Otherwise (Lines 5-6), we construct a new sum node $\alpha' = +_{i=1}^{k} \alpha_i' = +_{i=1}^{k} \text{AGG}(\alpha_i; \boldsymbol{W})$. By the inductive hypothesis, each $\alpha_i'$ has scope $\text{vars}(\alpha_i) \setminus \boldsymbol{W} = \text{vars}(\alpha) \setminus \boldsymbol{W}$, so $\alpha'$ is smooth and also has scope $\text{vars}(\alpha) \setminus \boldsymbol{W}$. $\qquad\square$

## A.2 Tractable Product

### A.2.1 Tractable Product with Compatibility

**Theorem 2** (Tractable Product - Compatibility). *Let $C, C'$ be compatible circuits over variables $\boldsymbol{V}, \boldsymbol{V}'$, respectively, and the same semiring. Then it is possible to compute their product as a circuit $C$ compatible with them (i.e., $p_{C''}(\boldsymbol{V} \cup \boldsymbol{V}') = p_C(\boldsymbol{V}) \otimes p_{C'}(\boldsymbol{V}'))$ in $O(|C||C'|)$ time and space.*

*Proof.* We prove this inductively bottom up, for nodes $\alpha \in C, \alpha' \in C$ such that $\text{vars}(\alpha) \cap (\boldsymbol{V} \cap \boldsymbol{V}') = \text{vars}(\alpha') \cap (\boldsymbol{V} \cap \boldsymbol{V}')$. Our claim is that $\texttt{PROD-SCMP}(\alpha, \alpha')$ (Algorithm 2) returns a node $\alpha''$ such that $p_{\alpha''} = p_\alpha \otimes p_{\alpha'}$, has scope $\text{vars}(\alpha'') = \text{vars}(\alpha) \cup \text{vars}(\alpha')$, and is decomposable (if product) and smooth (if sum).

If $\text{vars}(\alpha) \cap \text{vars}(\alpha') = \emptyset$ (i.e. $\text{vars}(\alpha) \cap (\boldsymbol{V} \cap \boldsymbol{V}') = \text{vars}(\alpha') \cap (\boldsymbol{V} \cap \boldsymbol{V}')$ is empty), then the algorithm (Lines 1-2) simply constructs a new product node $\alpha'' = \alpha \times \alpha'$. By definition, $p_{\alpha''} = p_\alpha \otimes p_{\alpha'}$, has scope $\text{vars}(\alpha'') = \text{vars}(\alpha) \cup \text{vars}(\alpha')$, and $\alpha''$ is decomposable.

If $\alpha, \alpha'$ are input nodes, then we can construct a new input node $\alpha''$ satisfying the requisite properties (Lines 5-6).

If $\alpha$ is an input or product node and $\alpha' = +_{j=1}^{k'} \alpha_j'$ is a sum node, then the algorithm constructs a new sum node $\alpha'' = +_{j=1}^{k'} \texttt{PROD-CMP}(\alpha, \alpha_j')$. This computes the correct function as $p_{\alpha''} = \oplus_{j=1}^{k'} \left( p_\alpha \otimes p_{\alpha_j'} \right) = p_\alpha \otimes \left( \oplus_{j=1}^{k'} p_{\alpha_j'} \right) = p_\alpha \otimes p_{\alpha'}$. Each child has scope $\text{vars}(\alpha) \cup \text{vars}(\alpha_j') = \text{vars}(\alpha) \cup \text{vars}(\alpha')$, so smoothness is retained.

If $\alpha = \alpha_1 \times \alpha_2, \alpha' = \alpha_1' \times \alpha_2'$ are product nodes such that $\text{vars}(\alpha) \cap (\boldsymbol{V} \cap \boldsymbol{V}') = \text{vars}(\alpha') \cap (\boldsymbol{V} \cap \boldsymbol{V}')$ is non-empty, then writing $\boldsymbol{X} := \boldsymbol{V} \cap \boldsymbol{V}'$, by compatibility we also have $\text{vars}(\alpha_1) \cap \boldsymbol{X} = \text{vars}(\alpha_1') \cap \boldsymbol{X}$ and $\text{vars}(\alpha_2) \cap \boldsymbol{X} = \text{vars}(\alpha_2') \cap \boldsymbol{X}$, so we can apply the inductive hypothesis for $\texttt{PROD-CMP}(\alpha_1, \alpha_1')$ and $\texttt{PROD-CMP}(\alpha_2, \alpha_2')$. Algorithm 2 constructs a new product node $\alpha'' = \texttt{PROD-CMP}(\alpha_1, \alpha_1') \times \texttt{PROD-CMP}(\alpha_2, \alpha_2')$. To show that this is decomposable, we need the following lemma:

**Lemma 1** (Decomposability of Product). *Suppose $\alpha \in C, \alpha' \in C'$ are decomposable product nodes which decompose in the same way over $\boldsymbol{X}$, i.e. $\text{vars}(\alpha_1) \cap \boldsymbol{X} = \text{vars}(\alpha_1') \cap \boldsymbol{X}$ and $\text{vars}(\alpha_2) \cap \boldsymbol{X} = \text{vars}(\alpha_2') \cap \boldsymbol{X}$. Then $(\text{vars}(\alpha_1) \cup \text{vars}(\alpha_1')) \cap (\text{vars}(\alpha_2) \cup \text{vars}(\alpha_2')) = \emptyset$.*

*Proof.* We have that:

$(\text{vars}(\alpha_1) \cup \text{vars}(\alpha_1')) \cap (\text{vars}(\alpha_2) \cup \text{vars}(\alpha_2'))$

$= (\text{vars}(\alpha_1) \cap \text{vars}(\alpha_2)) \cup (\text{vars}(\alpha_1') \cap \text{vars}(\alpha_2')) \cup (\text{vars}(\alpha_1) \cap \text{vars}(\alpha_2')) \cup (\text{vars}(\alpha_2) \cap \text{vars}(\alpha_1'))$

Note that the first two intersections are empty due to decomposability of $\alpha, \alpha'$. For the third intersection $(\text{vars}(\alpha_1) \cap \text{vars}(\alpha_2'))$, any variable in this intersection must be in the common variables $\boldsymbol{X}$. But we know that $\text{vars}(\alpha_2') \cap \boldsymbol{X} = \text{vars}(\alpha_2) \cap \boldsymbol{X}$ in both cases above; by decomposability, $(\text{vars}(\alpha_2') \cap \boldsymbol{X}) \cap (\text{vars}(\alpha_1) \cap \boldsymbol{X}) = \emptyset$. Thus the third intersection is also empty; a similar argument applies for the fourth. $\qquad\square$

Applying this Lemma, we see that $\alpha''$ is decomposable as $\text{vars}(\texttt{PROD-CMP}(\alpha_1, \alpha_1')) = (\text{vars}(\alpha_1) \cup \text{vars}(\alpha_1'))$ and $\text{vars}(\texttt{PROD-CMP}(\alpha_2, \alpha_2')) = (\text{vars}(\alpha_2) \cup \text{vars}(\alpha_2'))$. We can also verify that $p_{\alpha''} = p_{\texttt{PROD-CMP}(\alpha_1,\alpha_1')} \otimes p_{\texttt{PROD-CMP}(\alpha_2,\alpha_2')} = p_{\alpha_1} \otimes p_{\alpha_1'} \otimes p_{\alpha_2} \otimes p_{\alpha_2'} = p_\alpha \otimes p_{\alpha'}$ by the inductive hypothesis, and associativity of $\otimes$.

If $\alpha = +_{i=1}^k \alpha_i$, $\alpha' = +_{i=1}^{k'} \alpha_i'$ are sum nodes, then the algorithm produces a new sum node $\alpha'' = +_{i=1}^k +_{j=1}^{k'} \texttt{PROD-CMP}(\alpha_i, \alpha_j')$ (Lines 7-8). This computes the correct function as $p_{\alpha''} = \oplus_{i=1}^k \oplus_{j=1}^{k'} \texttt{PROD-CMP}(\alpha_i, \alpha_j') = \oplus_{i=1}^k \oplus_{j=1}^{k'} p_{\alpha_i} p_{\alpha_j'} = (\oplus_{i=1}^k p_{\alpha_i}) \otimes (\oplus_{j=1}^{k'} p_{\alpha_j'}) = p_\alpha \otimes p_{\alpha'}$. It also retains smoothness.

The complexity of this algorithm is $O(|C||C'|)$ because we perform recursive calls for pairs of nodes in $C$ and $C'$. $\qquad\square$

### A.2.2  Linear-time Product with Support Comptibility

**Theorem 3** (Tractable Product - Support Compatibility). *Let $C, C'$ be support-compatible circuits over variables $\boldsymbol{V}, \boldsymbol{V}'$, respectively, and the same semiring. Then, given the isomorphism $\iota$, it is possible to compute their product as a smooth and decomposable circuit $C''$ support-compatible with them (i.e., $p_{C''}(\boldsymbol{V} \cup \boldsymbol{V}') = p_C(\boldsymbol{V}) \otimes p_{C'}(\boldsymbol{V}'))$ in $O(\max(|C|, |C'|))$ time and space.*

*Proof.* We prove this inductively bottom up, for nodes $\alpha \in C$ such that $\alpha' \in C$ either satisfies $\alpha' = \iota(\alpha)$ or $\text{vars}(\alpha) \cap \text{vars}(\alpha') = \emptyset$. Our claim is that $\texttt{PROD-SCMP}(\alpha, \alpha')$ (Algorithm 3) returns a node $\alpha''$ such that $p_{\alpha''} = p_\alpha \otimes p_{\alpha'}$, has scope $\text{vars}(\alpha'') = \text{vars}(\alpha) \cup \text{vars}(\alpha')$, and is decomposable (if product) and smooth (if sum).

If $\text{vars}(\alpha) \cap \text{vars}(\alpha') = \emptyset$, then the algorithm (Lines 1-2) simply constructs a new product node $\alpha'' = \alpha \times \alpha'$. By definition, $p_{\alpha''} = p_\alpha \otimes p_{\alpha'}$, has scope $\text{vars}(\alpha'') = \text{vars}(\alpha) \cup \text{vars}(\alpha')$, and $\alpha''$ is decomposable.

If the $\alpha, \alpha'$ are input nodes, then we can construct a new input node $\alpha''$ satisfying the requisite properties (Lines 3-4).

If $\alpha = \alpha_1 \times \alpha_2, \alpha' = \alpha_1' \times \alpha_2'$ are product nodes and $\iota(\alpha) = \alpha'$, then the Algorithm (Lines 5-6) constructs a product node $\alpha'' = \texttt{PROD-SCMP}(\alpha_1, \alpha_1') \times \texttt{PROD-SCMP}(\alpha_2, \alpha_2')$. Define $\boldsymbol{X} = \boldsymbol{V} \cup \boldsymbol{V}'$. By support compatibility (i.e. $\boldsymbol{X}$-support compatibility), $\alpha, \alpha'$ are part of the restricted circuits $C[\boldsymbol{X}], C'[\boldsymbol{X}]$ respectively and so $\text{vars}(\alpha) \cap \boldsymbol{X} \neq \emptyset, \text{vars}(\alpha') \cap \boldsymbol{X} \neq \emptyset$. There are two cases to consider; we first show that in both of these cases, we can apply the inductive hypothesis to $\texttt{PROD-SCMP}(\alpha_1, \alpha_1')$ and $\texttt{PROD-SCMP}(\alpha_2, \alpha_2')$.

- Firstly, suppose that both $\alpha_1$ and $\alpha_2$ have scope overlapping with $\boldsymbol{X}$. Then by the isomorphism, we have $\alpha_1' = \iota(\alpha_1)$, $\alpha_2' = \iota(\alpha_2)$. By the definition of support compatibility, this also means $\text{vars}(\alpha_1) \cap \boldsymbol{X} = \text{vars}(\alpha_1') \cap \boldsymbol{X}$ and $\text{vars}(\alpha_2) \cap \boldsymbol{X} = \text{vars}(\alpha_2') \cap \boldsymbol{X}$ and these are both non-empty; thus we can apply the inductive hypothesis for $\texttt{PROD-SCMP}(\alpha_1, \alpha_1')$ and $\texttt{PROD-SCMP}(\alpha_2, \alpha_2')$.

- Second, suppose instead that only $\alpha_1$ has scope overlapping with $\boldsymbol{X}$, and so $\text{vars}(\alpha_2) \cap \boldsymbol{X} = \emptyset$. Then $\alpha_1' = \iota(\alpha_1)$ and $\text{vars}(\alpha_1) \cap \boldsymbol{X} = \text{vars}(\alpha_1') \cap \boldsymbol{X} = \text{vars}(\alpha) \cap \boldsymbol{X} = \text{vars}(\alpha') \cap \boldsymbol{X}$. Since $\text{vars}(\alpha_2') = \text{vars}(\alpha') \setminus \text{vars}(\alpha_1')$, it follows that $\text{vars}(\alpha_2) \cap \boldsymbol{X} = (\text{vars}(\alpha') \cap \boldsymbol{X}) \setminus (\text{vars}(\alpha_1') \cap \boldsymbol{X}) = \emptyset$, i.e. $\alpha_2'$ also does not have scope overlapping with $\boldsymbol{X}$. Since $\boldsymbol{X}$ are the shared variables $\boldsymbol{V}, \boldsymbol{V}'$, it follows that $\text{vars}(\alpha_2) \cap \text{vars}(\alpha_2') = \emptyset$, and so we can apply the inductive hypothesis for $\texttt{PROD-SCMP}(\alpha_2, \alpha_2')$ (and for $\texttt{PROD-SCMP}(\alpha_1, \alpha_1')$).

By the inductive hypothesis, $\texttt{PROD-SCMP}(\alpha_1, \alpha_1')$ has scope $\text{vars}(\alpha_1) \cup \text{vars}(\alpha_1')$ and $\texttt{PROD-SCMP}(\alpha_2, \alpha_2')$ has scope $\text{vars}(\alpha_2) \cup \text{vars}(\alpha_2')$. We can thus apply Lemma 1. Thus $\texttt{PROD-SCMP}(\alpha_1, \alpha_1')$ and $\texttt{PROD-SCMP}(\alpha_2, \alpha_2')$ have disjoint scopes and $\alpha''$ is decomposable. We

can also verify that $p_{\alpha''} = p_{\mathtt{PROD\text{-}SCMP}(\alpha_1,\alpha'_1)} \otimes p_{\mathtt{PROD\text{-}SCMP}(\alpha_2,\alpha'_2)} = p_{\alpha_1} \otimes p_{\alpha'_1} \otimes p_{\alpha_2} \otimes p_{\alpha'_2} = p_\alpha \otimes p_{\alpha'}$ by the inductive hypothesis, and associativity of $\otimes$.

If $\alpha = +_{i=1}^{k} \alpha_i$, $\alpha' = +_{i=1}^{k'} \alpha'_i$ are sum nodes and $\iota(\alpha) = \alpha'$, then by smoothness, all of the children of $\alpha$ have the same support and all the children of $\alpha'$ have the same support; thus all the children are in $C[\boldsymbol{X}], C'[\boldsymbol{X}]$ respectively, $k = k'$, and $\iota(\alpha_i) = \alpha'_i$. By support compatibility, we also that (i) $\mathrm{vars}(\alpha_i) \cap \boldsymbol{X} = \mathrm{vars}(\alpha'_j) \cap \boldsymbol{X}$ for all $i, j$; and (ii) that $\mathrm{supp}_{\boldsymbol{X}}(\alpha_i) \cap \mathrm{supp}_{\boldsymbol{X}}(\alpha'_j)$ for $i \neq j$.

We claim that $p_{\alpha_i} \otimes p_{\alpha'_j} \equiv 0_{\mathcal{S}}$ whenever $i \neq j$. To see this, recall the definition of $\boldsymbol{X}$-support: we have that:

$$\mathrm{supp}_{\boldsymbol{X}}(\alpha_i) = \{\boldsymbol{x} \in \mathrm{Assign}(\boldsymbol{X} \cap \mathrm{vars}(\alpha_i)) : \exists \boldsymbol{y} \in \mathrm{Assign}(\mathrm{vars}(\alpha_i) \setminus \boldsymbol{X}) \text{ s.t. } p_{\alpha_i}(\boldsymbol{x}, \boldsymbol{y}) \neq 0_{\mathcal{S}}\}$$
$$\mathrm{supp}_{\boldsymbol{X}}(\alpha'_j) = \{\boldsymbol{x} \in \mathrm{Assign}(\boldsymbol{X} \cap \mathrm{vars}(\alpha'_j)) : \exists \boldsymbol{y} \in \mathrm{Assign}(\mathrm{vars}(\alpha'_j) \setminus \boldsymbol{X}) \text{ s.t. } p_{\alpha'_j}(\boldsymbol{x}, \boldsymbol{y}) \neq 0_{\mathcal{S}}\}$$

Since $\boldsymbol{X} \cap \mathrm{vars}(\alpha_i) = \boldsymbol{X} \cap \mathrm{vars}(\alpha'_j)$ and is nonempty, by (ii) we know that there is no assignment of $\boldsymbol{X} \cap \mathrm{vars}(\alpha_i)$ such that $p_{\alpha_i}$ and $p_{\alpha'_j}$ can be simultaneously not equal to $0_{\mathcal{S}}$. Thus there is no assignment of $\boldsymbol{X} \cap \mathrm{vars}(\alpha_i)$ such that $p_{\alpha_i} \otimes p_{\alpha'_j}$ is not $0_{\mathcal{S}}$, since $0_{\mathcal{S}}$ is the multiplicative annihilator.

Thus, the product function is given by:

$$p_\alpha \otimes p_{\alpha'} = \bigoplus_{i=1}^{k} \bigoplus_{j=1}^{k} (p_{\alpha_i} \otimes p_{\alpha'_j})$$
$$= \bigoplus_{i=1}^{k} (p_{\alpha_i} \otimes p_{\alpha'_i})$$
$$= \bigoplus_{i=1}^{k} \mathtt{PROD\text{-}SCMP}(\alpha_i, \alpha'_i)$$

The second equality follows by the Lemma and the fact that $0_{\mathcal{S}}$ is the additive identity, and the third equality by the inductive hypothesis. Thus $\alpha'' = +_{i=1}^{k} \mathtt{PROD\text{-}SCMP}(\alpha_i, \alpha'_i)$ computes the correct function (Lines 7-8). We conclude by noting that $\mathrm{vars}(\alpha'') = \bigcup_{i=1}^{k} (\mathrm{vars}(\alpha_i) \cup \mathrm{vars}(\alpha_i)) = \bigcup_{i=1}^{k} \mathrm{vars}(\alpha_i) \cup \bigcup_{i=1}^{k} \mathrm{vars}(\alpha_i) = \mathrm{vars}(\alpha) \cup \mathrm{vars}(\alpha')$.

The complexity of this procedure applied to the root nodes is $O(\max(|C|, |C'|))$, as we only perform recursive calls for (i) $\alpha \in C[\boldsymbol{X}]$ and its corresponding node $\alpha' = \iota(\alpha)$ and (ii) nodes with non-overlapping scope, upon which the recursion ends; so the overall number of recursive calls is linear in the size of the circuits.

$\square$

### A.3 Tractable Elementwise Mapping

**Theorem 4** (Tractable Mapping). *Let $C$ be a smooth and decomposable circuit over semiring $\mathcal{S}$, and $\tau : \mathcal{S} \to \mathcal{S}'$ a mapping such that $\tau(0_{\mathcal{S}}) = 0_{\mathcal{S}'}$. Then it is possible to compute the mapping of $C$ by $\tau$ as a smooth and decomposable circuit $C'$ (i.e., $p_{C'}(\boldsymbol{V}) = \tau(p_C(\boldsymbol{V}))$) in $O(|C|)$ time and space if $\tau$ distributes over sums and over products.*

*$\tau$ distributes over sums if: either **(Additive)** $\tau$ is an additive homomorphism, i.e. $\tau(a \oplus b) = \tau(a) \oplus \tau(b)$; or **(Det)** $C$ is deterministic.*

*$\tau$ distributes over products if: either **(Multiplicative)** $\tau$ is an multiplicative homomorphism, i.e. $\tau(a \otimes b) = \tau(a) \otimes \tau(b)$; or **(Prod 0/1)** $\tau(1_{\mathcal{S}}) = 1_{\mathcal{S}'}$, and for all product nodes $\alpha = \alpha_1 \times \alpha_2 \in C$, and for every value $\boldsymbol{v} \in \mathrm{Assign}(\mathrm{vars}(\alpha))$, either $p_{\alpha_1}(\boldsymbol{v}_{\mathrm{vars}(\alpha_1)}) \in \{0_{\mathcal{S}}, 1_{\mathcal{S}}\}$ or $p_{\alpha_2}(\boldsymbol{v}_{\mathrm{vars}(\alpha_2)}) \in \{0_{\mathcal{S}}, 1_{\mathcal{S}}\}$.*

*Proof.* First, let us consider sum nodes. Given any sum node $\alpha = +_{i=1}^{k} \alpha_i \in C$, we consider computing a circuit representing

$$\tau(p_\alpha(\mathrm{vars}(\alpha))) \equiv \tau\left(\bigoplus_{i=1}^{k} p_{\alpha_i}(\mathrm{vars}(\alpha))\right) \tag{5}$$

If (Additive) holds, then we immediately have that $\tau\big(\bigoplus_{i=1}^{k} p_{\alpha_i}(\text{vars}(\alpha))\big) \equiv \bigoplus_{i=1}^{k} \tau(p_{\alpha_i}(\text{vars}(\alpha)))$ by associativity of $\oplus$. Alternatively, if (Det) holds, then given any $\boldsymbol{v} \in \text{Assign}((\text{vars}(\alpha)))$, there is at most one child, say $\alpha_j$, such that $p_{\alpha_j}(\boldsymbol{v}) \neq 0_{\mathcal{S}}$. Then we have that

$$
\begin{aligned}
\tau\big(\oplus_{i=1}^{k} p_{\alpha_i}(\boldsymbol{v})\big) &= \tau\Big(p_{\alpha_j}(\boldsymbol{v}) \oplus \Big(\bigoplus_{i=1,i\neq j}^{k} p_{\alpha_i}(\boldsymbol{v})\Big)\Big) \\
&= \tau\Big(p_{\alpha_j}(\boldsymbol{v}) \oplus \Big(\bigoplus_{i=1,i\neq j}^{k} 0_{\mathcal{S}}\Big)\Big) \\
&= \tau\big(p_{\alpha_j}(\boldsymbol{v})\big) \\
&= \tau\big(p_{\alpha_j}(\boldsymbol{v})\big) \oplus \Big(\bigoplus_{i=1,i\neq j}^{k} 0_{\mathcal{S}'}\Big) \\
&= \tau\big(p_{\alpha_j}(\boldsymbol{v})\big) \oplus \Big(\bigoplus_{i=1,i\neq j}^{k} \tau(0_{\mathcal{S}})\Big) \\
&= \tau\big(p_{\alpha_j}(\boldsymbol{v})\big) \oplus \Big(\bigoplus_{i=1,i\neq j}^{k} \tau(p_{\alpha_i}(\boldsymbol{v}))\Big) \\
&= \bigoplus_{i=1}^{k} \tau(p_{\alpha_i}(\boldsymbol{v}))
\end{aligned}
$$

and so again $\tau\big(\bigoplus_{i=1}^{k} p_{\alpha_i}(\boldsymbol{v})\big) \equiv \bigoplus_{i=1}^{k} \tau(p_{\alpha_i}(\boldsymbol{v}))$.

Second, let us consider product nodes. If (Multiplicative) holds, then we immediately have that $\tau\big(\bigotimes_{i=1}^{k} p_{\alpha_i}(\text{vars}(\alpha))\big) \equiv \bigotimes_{i=1}^{k} \tau(p_{\alpha_i}(\text{vars}(\alpha)))$ by associativity of $\otimes$. Otherwise, if (Prod 0/1) holds, then given any $\boldsymbol{v} \in \text{Assign}(\text{vars}(\alpha))$, there is at most one child, say $\alpha_j$, such that $p_{\alpha_j}(\boldsymbol{v}) \notin \{0_{\mathcal{S}}, 1_{\mathcal{S}}\}$. Thus, we have that:

$$
\begin{aligned}
\tau\Big(\bigotimes_{i=1}^{k} p_{\alpha_i}(\boldsymbol{v})\Big) &= \tau\Big(p_{\alpha_j}(\boldsymbol{v}) \otimes \Big(\bigotimes_{i=1,i\neq j}^{k} p_{\alpha_i}(\boldsymbol{v})\Big)\Big) \\
&= \tau\big(p_{\alpha_j}(\boldsymbol{v})\big) \otimes \tau\Big(\bigotimes_{i=1,i\neq j}^{k} p_{\alpha_i}(\boldsymbol{v})\Big) \\
&= \tau\big(p_{\alpha_j}(\boldsymbol{v})\big) \otimes \Big(\bigotimes_{i=1,i\neq j}^{k} \tau(p_{\alpha_i}(\boldsymbol{v}))\Big) \\
&= \bigotimes_{i=1}^{k} \tau(p_{\alpha_i}(\boldsymbol{v}))
\end{aligned}
$$

The second equality follows because $\big(\bigotimes_{i=1,i\neq j}^{k} p_{\alpha_i}(\boldsymbol{v})\big) \in \{0_{\mathcal{S}}, 1_{\mathcal{S}}\}$, and we have that $\tau(a \otimes 0_{\mathcal{S}}) = 0_{\mathcal{S}'} = \tau(a) \otimes \tau(0_{\mathcal{S}})$ and $\tau(a \otimes 1_{\mathcal{S}}) = 1_{\mathcal{S}'} = \tau(a) \otimes \tau(1_{\mathcal{S}})$ for any $a \in \mathcal{S}$. The third equality follows as both $\tau\big(\bigotimes_{i=1,i\neq j}^{k} p_{\alpha_i}(\boldsymbol{v})\big)$ and $\bigotimes_{i=1,i\neq j}^{k} \tau(p_{\alpha_i}(\boldsymbol{v}))$ are equal to $1_{\mathcal{S}'}$ iff no $p_{\alpha_i}(\boldsymbol{v})$ is $0_{\mathcal{S}}$. Thus, we have that $\tau\big(\bigotimes_{i=1}^{k} p_{\alpha_i}(\boldsymbol{v})\big) \equiv \bigotimes_{i=1}^{k} \tau(p_{\alpha_i}(\boldsymbol{v}))$.

By applying these identities recursively to sum and product nodes, and assuming that $\tau$ can be applied tractably to input nodes, we obtain a circuit $C'$ such that $p_{C'}(\boldsymbol{V}) \equiv \tau(p_C(\boldsymbol{V}))$. $\qquad\square$

### A.4 Tractable Composition of operators

**Theorem 5** (Composability Conditions). *The results in Table 1 hold.*

| | Tractability Conditions | If the Input Circuit(s) are ... | | | Complexity |
| | | $X$-Det | $X$-Cmp w/ $C_{\text{other}}$ | $X$-SCmp w/ $C_{\text{other}}$ | |
| | | Then the Output Circuit is ... | | | |
| **Aggregation ($W$)** | Sm AND Dec | $X$-Det if $W \cap X = \emptyset$ (5.1) | $X$-Cmp w/ $C_{\text{other}}$ if $W \cap X = \emptyset$ (5.5) | $X$-SCmp w/ $C_{\text{other}}$ if $W \cap X = \emptyset$ (5.9) | $O(\|C\|)$ |
| **Product** | Cmp | $X$-Det (5.2) | $X$-Cmp w/ $C_{\text{other}}$ (5.6) | N/A | $O(\|C\|\|C'\|)$ |
| | SCmp | $X$-Det (5.3) | $X$-Cmp w/ $C_{\text{other}}$ (5.7) | $X$-SCmp w/ $C_{\text{other}}$ (5.10) | $O(\max(\|C\|, \|C'\|))$ |
| **Elem. Mapping** | (Sm AND Dec) AND (Add OR Det) AND (Mult OR Prod01) | $X$-Det (5.4) | $X$-Cmp w/ $C_{\text{other}}$ (5.8) | $X$-SCmp w/ $C_{\text{other}}$ (5.11) | $O(\|C\|)$ |

Table 3: Tractability Conditions for Operations on Algebraic Circuits. Sm: Smoothness, Dec: Decomposability; $X$-Det(erminism), $X$-Cmp: $X$-Compatibility, $X$-SCmp: $X$-Support-Compatibility.

*Proof.* We look at each property in turn, and show that they are maintained under the aggregation, product, and mapping operators as stated in the Table. For convenience, we reproduce the table in Table 3, with each result highlighted with a number that is referenced in the proof below.

**$X$-determinism**   Suppose that circuit $C$ is $X$-deterministic; that is, for any sum node $\alpha = +_{i=1}^{k} \alpha_i \in C$, either (i) $\text{vars}(\alpha) \cap X = \emptyset$, or else (ii) $\text{supp}_X(\alpha_i) \cap \text{supp}_X(\alpha_j) = \emptyset$ for all $i \neq j$.

**(5.1)** Consider aggregating with respect to a set of variables $W$ such that $W \cap X = \emptyset$. According to Algorithm 1 and the proof of Theorem 1, this produces an output circuit where each node $\alpha'$ corresponds to some node $\alpha$ in the original circuit, such that $p_{\alpha'} = \bigoplus_{w \cap \text{vars}(\alpha)} p_\alpha$ and with scope $\text{vars}(\alpha) \setminus W$. In particular, for sum nodes $\alpha = +_{i=1}^{k} \alpha_i \in C$, either $\text{vars}(\alpha) \subseteq W$, in which case $\alpha'$ is an input node (and $X$-determinism is not applicable), or else $\alpha' = +_{i=1}^{k} \alpha_i'$ is also a sum node, where each $\alpha_i'$ corresponds to $\alpha_i$. If (i) $\text{vars}(\alpha) \cap X = \emptyset$, then $\text{vars}(\alpha') \cap X = \emptyset$ also.

If (ii) $\text{supp}_X(\alpha_i) \cap \text{supp}_X(\alpha_j) = \emptyset$ for all $i \neq j$, we claim that $\text{supp}_X(\alpha_i') \subseteq \text{supp}_X(\alpha_i)$ for all $i$. To see this, first note that by smoothness, $\text{vars}(\alpha_i') = \text{vars}(\alpha_j') = \text{vars}(\alpha')$. Suppose that $x_i \in \text{Assign}(X \cap \text{vars}(\alpha'))$ satisfies $x \in \text{supp}_X(\alpha_i')$. Then there exists $y_i \in \text{Assign}(\text{vars}(\alpha') \setminus X)$ such that $p_{\alpha_i'}(x_i, y_i) \neq 0_\mathcal{S}$. Since $\alpha_i'$ corresponds to $\alpha_i$ in the original circuit, we have:

$$\bigoplus_{w \in \text{Assign}(W) \cap \text{vars}(\alpha)} p_{\alpha_i}(x_i, y_i, w_i) = p_{\alpha_i'}(x, y) \neq 0_\mathcal{S}$$

This means that there must be some $w_i \in \text{Assign}(W) \cap \text{vars}(\alpha)$ such that $p_{\alpha_i}(x, y_i, w_i) \neq 0_\mathcal{S}$ (since $0_\mathcal{S}$ is the additive identity); thus $x \in \text{supp}_X(\alpha_i)$. To finish the proof, note that $\text{supp}_X(\alpha_i') \subseteq \text{supp}_X(\alpha_i)$ and $\text{supp}_X(\alpha_l') \subseteq \text{supp}_X(\alpha_l)$ are disjoint unless $i = l$ (by $X$-determinism of $\alpha$, i.e. $\text{supp}_X(\alpha_i) \cap \text{supp}_X(\alpha_l) = \emptyset$ unless $i = l$). Thus (ii) holds for $\alpha'$. In either case, we have shown that $\alpha'$ is also $X$-deterministic.

**(5.2)** Consider taking the product of two compatible circuits $C, C'$ over variables $V, V'$, outputting a circuit $C''$. According to Algorithm 2 and the proof of Theorem 2, every sum node $\alpha'' \in C''$ corresponds to either the product of (a) an input or product node $\alpha \in C$ and a sum node $\alpha' = +_{j=1}^{k'} \alpha_j' \in C'$, such that $\alpha'' = +_{j=1}^{k'} \alpha_j''$ or (b) two sum nodes $\alpha = +_{i=1}^{k} \alpha_i \in C$ and $\alpha' = +_{j=1}^{k'} \alpha_j' \in C'$, such that $\alpha'' = +_{i=1}^{k} +_{j=1}^{k'} \alpha_{ij}''$. Further, $\alpha$ and $\alpha'$ have the same scope over the common variables $V \cap V'$, i.e. $\text{vars}(\alpha) \cap (V \cap V') = \text{vars}(\alpha') \cap (V \cap V')$.

Assume that $C$ and $C'$ are both $X$-deterministic; then $X \subseteq V \cap V'$. We note that since $\alpha, \alpha'$ have the same scope over the common variables, they also have the same scope over $X$, i.e. $\text{vars}(\alpha) \cap X = \text{vars}(\alpha') \cap X$.

In case (a), $X$-determinism of $\alpha'$ means that either (i) $\text{vars}(\alpha') \cap X = \emptyset$ or (ii) $\text{supp}_X(\alpha_i') \cap \text{supp}_X(\alpha_j') = \emptyset$ for all $i \neq j$. If (i), then $\text{vars}(\alpha'') \cap X = (\text{vars}(\alpha) \cup \text{vars}(\alpha')) \cap X = \emptyset$ also. If (ii), note that $\text{supp}_X(\alpha_j'') \subseteq \text{supp}_X(\alpha_j')$ for all $j$ as $a \otimes 0_\mathcal{S} = 0_\mathcal{S}$ for any semiring $\mathcal{S}$ and $a \in \mathcal{S}$. Thus $\text{supp}_X(\alpha_i'') \cap \text{supp}_X(\alpha_j'') = \emptyset$ for all $i \neq j$. Thus $\alpha''$ is $X$-deterministic.

In case (b), since $\alpha, \alpha'$ have the same scope over $X$, either (i) holds for both $\alpha, \alpha'$, or (ii) holds for both. If (i), then $\text{vars}(\alpha'') \cap X = (\text{vars}(\alpha) \cup \text{vars}(\alpha')) \cap X = \emptyset$ also. If (ii), then for any $i, j$, consider the restricted support $\text{supp}_X(\alpha_{ij}'')$. Noting that $\text{vars}(\alpha_i) \cap X = \text{vars}(\alpha_j') \cap X = \text{vars}(\alpha_{ij}'') \cap$

$\boldsymbol{X}$ by smoothness, we claim that $\mathrm{supp}_{\boldsymbol{X}}(\alpha_{ij}'') \subseteq \mathrm{supp}_{\boldsymbol{X}}(\alpha_i) \cap \mathrm{supp}_{\boldsymbol{X}}(\alpha_j')$. Suppose that $\boldsymbol{x} \in \mathrm{supp}_{\boldsymbol{X}}(\alpha_{ij}'')$. Then there exists some $\boldsymbol{y} \in \mathrm{vars}(\alpha_{ij}'')\backslash\boldsymbol{X}$ such that $p_{\alpha_{ij}''}(\boldsymbol{x}, \boldsymbol{y}) = p_{\alpha_i}(\boldsymbol{x}, \boldsymbol{y}_{\mathrm{vars}(\alpha_i))\backslash\boldsymbol{X}})\otimes p_{\alpha_j'}(\boldsymbol{x}, \boldsymbol{y}_{\mathrm{vars}(\alpha_j')\backslash\boldsymbol{X}}) \neq 0_{\mathcal{S}}$. This means that both $p_{\alpha_i}(\boldsymbol{x}, \boldsymbol{y}_{\mathrm{vars}(\alpha_i))\backslash\boldsymbol{X}}), p_{\alpha_j'}(\boldsymbol{x}, \boldsymbol{y}_{\mathrm{vars}(\alpha_j')\backslash\boldsymbol{X}})$ cannot be $0_{\mathcal{S}}$, and so $\boldsymbol{x} \in \mathrm{supp}_{\boldsymbol{X}}(\alpha_i)$ and $\boldsymbol{x} \in \mathrm{supp}_{\boldsymbol{X}}(\alpha_j')$ also. To finish the proof, we note that $\mathrm{supp}_{\boldsymbol{X}}(\alpha_{ij}'') \subseteq \mathrm{supp}_{\boldsymbol{X}}(\alpha_i)\cap\mathrm{supp}_{\boldsymbol{X}}(\alpha_j')$ and $\mathrm{supp}_{\boldsymbol{x}}(\alpha_{lm}'') \subseteq \mathrm{supp}_{\boldsymbol{X}}(\alpha_l)\cap\mathrm{supp}_{\boldsymbol{X}}(\alpha_m')$ are disjoint unless $i = l, j = m$ (by $\boldsymbol{X}$-determinism of $\alpha$ and $\alpha'$). Thus $\alpha''$ is $\boldsymbol{X}$-deterministic by (ii).

**(5.3)** Consider taking the product of two support-compatible circuits $C, C'$ over variables $\boldsymbol{V}, \boldsymbol{V}'$, outputting a circuit $C''$. According to Algorithm 3 and the proof of Theorem 3, every sum node $\alpha'' = +_{i=1}^{k}\alpha_i'' \in C''$ corresponds to some sum nodes $\alpha = +_{i=1}^{k}\alpha_i \in C$ and $\alpha' = +_{i=1}^{k}\alpha_i' \in C'$ such that $\alpha' = \iota(\alpha)$, $p_{\alpha_i''} = p_{\alpha_i} \otimes p_{\alpha_i'}$, and has scope $\mathrm{vars}(\alpha) \cup \mathrm{vars}(\alpha')$. Further, $\alpha$ and $\alpha'$ have the same scope over the common variables $\boldsymbol{V} \cap \boldsymbol{V}'$, i.e. $\mathrm{vars}(\alpha) \cap (\boldsymbol{V} \cap \boldsymbol{V}') = \mathrm{vars}(\alpha') \cap (\boldsymbol{V} \cap \boldsymbol{V}')$.

Assume that $C$ and $C'$ are both $\boldsymbol{X}$-deterministic; then $\boldsymbol{X} \subseteq \boldsymbol{V} \cap \boldsymbol{V}'$. We note that since $\alpha, \alpha'$ have the same scope over the common variables, they also have the same scope over $\boldsymbol{X}$, i.e. $\mathrm{vars}(\alpha) \cap \boldsymbol{X} = \mathrm{vars}(\alpha') \cap \boldsymbol{X}$. Thus, either (i) holds for both $\alpha, \alpha'$, or (ii) holds for both. If (i), then $\mathrm{vars}(\alpha'') \cap \boldsymbol{X} = (\mathrm{vars}(\alpha) \cup \mathrm{vars}(\alpha')) \cap \boldsymbol{X} = \emptyset$ also. If (ii), then for any $i$, consider the restricted support $\mathrm{supp}_{\boldsymbol{X}}(\alpha_{ij}'')$. Noting that $\mathrm{vars}(\alpha_i) \cap \boldsymbol{X} = \mathrm{vars}(\alpha_j') \cap \boldsymbol{X} = \mathrm{vars}(\alpha_i'') \cap \boldsymbol{X}$ by smoothness, we claim that $\mathrm{supp}_{\boldsymbol{X}}(\alpha_i'') \subseteq \mathrm{supp}_{\boldsymbol{X}}(\alpha_i) \cap \mathrm{supp}_{\boldsymbol{X}}(\alpha_i')$. Suppose that $\boldsymbol{x} \in \mathrm{supp}_{\boldsymbol{X}}(\alpha_i'')$. Then there exists some $\boldsymbol{y} \in \mathrm{vars}(\alpha_i'') \backslash \boldsymbol{X}$ such that $p_{\alpha_i''}(\boldsymbol{x}, \boldsymbol{y}) = p_{\alpha_i}(\boldsymbol{x}, \boldsymbol{y}_{\mathrm{vars}(\alpha_i))\backslash\boldsymbol{X}}) \otimes p_{\alpha_i'}(\boldsymbol{x}, \boldsymbol{y}_{\mathrm{vars}(\alpha_i')\backslash\boldsymbol{X}}) \neq 0_{\mathcal{S}}$. This means that both $p_{\alpha_i}(\boldsymbol{x}, \boldsymbol{y}_{\mathrm{vars}(\alpha_i))\backslash\boldsymbol{X}}), p_{\alpha_i'}(\boldsymbol{x}, \boldsymbol{y}_{\mathrm{vars}(\alpha_i')\backslash\boldsymbol{X}})$ cannot be $0_{\mathcal{S}}$, and so $\boldsymbol{x} \in \mathrm{supp}_{\boldsymbol{X}}(\alpha_i)$ and $\boldsymbol{x} \in \mathrm{supp}_{\boldsymbol{X}}(\alpha_i')$ also. To finish the proof, we note that $\mathrm{supp}_{\boldsymbol{X}}(\alpha_i'') \subseteq \mathrm{supp}_{\boldsymbol{X}}(\alpha_i) \cap \mathrm{supp}_{\boldsymbol{X}}(\alpha_i')$ and $\mathrm{supp}_{\boldsymbol{x}}(\alpha_l'') \subseteq \mathrm{supp}_{\boldsymbol{X}}(\alpha_l) \cap \mathrm{supp}_{\boldsymbol{X}}(\alpha_l')$ are disjoint unless $i = l$ (by $\boldsymbol{X}$-determinism of $\alpha$ and $\alpha'$). Thus $\alpha''$ is $\boldsymbol{X}$-deterministic by (ii).

**(5.4)** Consider applying an elementwise mapping $\tau$ to a circuit $C$, outputting a circuit $C'$. According to Algorithm 4 and Theorem 4, every sum node $\alpha' = +_{i=1}^{k}\alpha_i' \in C'$ corresponds to some node $\alpha = +_{i=1}^{k}\alpha_i \in C'$, such that $p_{\alpha'} = \tau(p_\alpha)$, and further $p_{\alpha_i'} = \tau(p_{\alpha_i})$ and $\mathrm{vars}(\alpha_i') = \mathrm{vars}(\alpha_i)$ for each $i$.

Assume that $C$ is $\boldsymbol{X}$-deterministic. If (i) $\mathrm{vars}(\alpha) \cap \boldsymbol{X} = \emptyset$, then $\mathrm{vars}(\alpha')\boldsymbol{X} = \emptyset$ also. Otherwise, (ii) $\mathrm{supp}_{\boldsymbol{X}}(\alpha_i) \cap \mathrm{supp}_{\boldsymbol{X}}(\alpha_j) = \emptyset$ for all $i \neq j$. We claim that $\mathrm{supp}_{\boldsymbol{X}}(\alpha_i') \subseteq \mathrm{supp}_{\boldsymbol{X}}(\alpha_i)$ for each $i$. To see this, recall that elementwise mappings satisfy $\tau(0_{\mathcal{S}}) = 0_{\mathcal{S}'}$. If $\boldsymbol{x} \in \mathrm{supp}_{\boldsymbol{X}}(\alpha_i')$, then there exists $\boldsymbol{y}$ s.t. $p_{\alpha_i'}(\boldsymbol{x}, \boldsymbol{y}) \neq 0_{\mathcal{S}'}$. Since $p_{\alpha_i'}(\boldsymbol{x}, \boldsymbol{y}) = \tau(p_{\alpha_i}(\boldsymbol{x}, \boldsymbol{y}))$, $p_{\alpha_i}(\boldsymbol{x}, \boldsymbol{y}) \neq 0_{\mathcal{S}}$. So $\boldsymbol{x} \in \mathrm{supp}_{\boldsymbol{X}}(\alpha_i)$. To finish the proof, note that $\mathrm{supp}_{\boldsymbol{x}}(\alpha_i') \subseteq \mathrm{supp}_{\boldsymbol{X}}(\alpha_i)$ and $\mathrm{supp}_{\boldsymbol{x}}(\alpha_l') \subseteq \mathrm{supp}_{\boldsymbol{X}}(\alpha_l)$ are disjoint unless $i = l$ (by $\boldsymbol{X}$-determinism of $\alpha$). Thus $\alpha'$ is $\boldsymbol{X}$-deterministic by (ii).

**$\boldsymbol{X}$-compatibility** Recall that two smooth and decomposable circuits $C, C_{\mathrm{other}}$ over variables $\boldsymbol{V}, \boldsymbol{V}_{\mathrm{other}}$ are $\boldsymbol{X}$-compatible for $\boldsymbol{X} \subseteq \boldsymbol{V} \cap \boldsymbol{V}_{\mathrm{other}}$ if for every product node $\alpha = \alpha_1 \times \alpha_2 \in C$ and $\alpha_{\mathrm{other}} = \alpha_{\mathrm{other},1} \times \alpha_{\mathrm{other},2} \in C_{\mathrm{other}}$ such that $\mathrm{vars}(\alpha) \cap \boldsymbol{X} = \mathrm{vars}(\alpha_{\mathrm{other}}) \cap \boldsymbol{X}$, it holds that $\mathrm{vars}(\alpha_1) \cap \boldsymbol{X} = \mathrm{vars}(\alpha_{\mathrm{other},1}) \cap \boldsymbol{X}$ and $\mathrm{vars}(\alpha_2) \cap \boldsymbol{X} = \mathrm{vars}(\alpha_{\mathrm{other},2}) \cap \boldsymbol{X}$.

**(5.5)** Suppose that $C, C_{\mathrm{other}}$ are $\boldsymbol{X}$-compatible. We wish to show that $C_{\mathrm{other}}, C'$ are $\boldsymbol{X}$-compatible where $C'$ is the output circuit from Algorithm 1 that aggregates $C$ over $\boldsymbol{W}$, where $\boldsymbol{W} \cap \boldsymbol{X} = \emptyset$.

Suppose $\alpha' = \alpha_1' \times \alpha_2' \in C'$ and $\alpha_{\mathrm{other}} = \alpha_{\mathrm{other},1} \times \alpha_{\mathrm{other},2} \in C_{\mathrm{other}}$ are product nodes such that $\mathrm{vars}(\alpha') \cap \boldsymbol{X} = \mathrm{vars}(\alpha_{\mathrm{other}}) \cap \boldsymbol{X}$. Let $\alpha = \alpha_1 \times \alpha_2$ be the corresponding node in $C$ such that $p_{\alpha'} = \bigoplus_{\boldsymbol{w}} p_\alpha$. The scope $\mathrm{vars}(\alpha') = \mathrm{vars}(\alpha) \setminus \boldsymbol{W}$; since $\boldsymbol{W} \cap \boldsymbol{X} = \emptyset$, we have $\mathrm{vars}(\alpha) \cap \boldsymbol{X} = \mathrm{vars}(\alpha_{\mathrm{other}}) \cap \boldsymbol{X}$ also. Thus, by $\boldsymbol{X}$-compatibility of $C, C_{\mathrm{other}}$, we have that $\mathrm{vars}(\alpha_1) \cap \boldsymbol{X} = \mathrm{vars}(\alpha_{\mathrm{other},1}) \cap \boldsymbol{X}$ and $\mathrm{vars}(\alpha_2) \cap \boldsymbol{X} = \mathrm{vars}(\alpha_{\mathrm{other},2}) \cap \boldsymbol{X}$. Since $\mathrm{vars}(\alpha_1') = \mathrm{vars}(\alpha_1) \setminus \boldsymbol{W}$ and $\mathrm{vars}(\alpha_2') = \mathrm{vars}(\alpha_2) \setminus \boldsymbol{W}$, this means that $\mathrm{vars}(\alpha_1') \cap \boldsymbol{X} = \mathrm{vars}(\alpha_{\mathrm{other},1}) \cap \boldsymbol{X}$ and $\mathrm{vars}(\alpha_2') \cap \boldsymbol{X} = \mathrm{vars}(\alpha_{\mathrm{other},2}) \cap \boldsymbol{X}$. Thus $C', C_{\mathrm{other}}$ are $\boldsymbol{X}$-compatible.

**(5.6)** Suppose that $C$ over $\boldsymbol{V}$ and $C'$ over $\boldsymbol{V}'$ are both $\boldsymbol{X}$-compatible with $C_{\mathrm{other}}$. We wish to show that $C_{\mathrm{other}}, C''$ are $\boldsymbol{X}$-compatible where $C''$ is the output circuit from Algorithm 2 that computes the product of the two compatible (i.e. $(\boldsymbol{V} \cup \boldsymbol{V}')$-compatible) circuits $C, C'$.

Suppose $\alpha'' = \alpha_1'' \times \alpha_2'' \in C''$ is a product node, and $\alpha_{\mathrm{other}} = \alpha_{\mathrm{other},1} \times \alpha_{\mathrm{other},2} \in C_{\mathrm{other}}$ such that $\mathrm{vars}(\alpha'') \cap \boldsymbol{X} = \mathrm{vars}(\alpha_{\mathrm{other}}) \cap \boldsymbol{X}$; we need to show that these decompose in the same way over $\boldsymbol{X}$. By Algorithm 2 and the proof of Theorem 2, this was created as the product of nodes $\alpha = \alpha_1 \times \alpha_2 \in C$ and $\alpha' = \alpha_1' \times \alpha_2' \in C'$ such that $\mathrm{vars}(\alpha'') \cap (\boldsymbol{V} \cap \boldsymbol{V}') = \mathrm{vars}(\alpha) \cap (\boldsymbol{V} \cap \boldsymbol{V}') =$

$\text{vars}(\alpha') \cap (\boldsymbol{V} \cap \boldsymbol{V'})$ (and similarly for their children). Thus by $(\boldsymbol{V} \cup \boldsymbol{V'})$-compatibility of $C, C', \alpha$ and $\alpha'$ decompose the same way over $(\boldsymbol{V} \cup \boldsymbol{V'})$, i.e. $\text{vars}(\alpha_1) \cap (\boldsymbol{V} \cup \boldsymbol{V'}) = \text{vars}(\alpha'_1) \cap (\boldsymbol{V} \cup \boldsymbol{V'})$ and $\text{vars}(\alpha_2) \cap (\boldsymbol{V} \cup \boldsymbol{V'}) = \text{vars}(\alpha'_2) \cap (\boldsymbol{V} \cup \boldsymbol{V'})$. Since $\boldsymbol{X} \subseteq \boldsymbol{V} \cap \boldsymbol{V'}$ (by definition of compatibility), this also holds over $\boldsymbol{X}$, i.e. $\text{vars}(\alpha_1) \cap \boldsymbol{X} = \text{vars}(\alpha'_1) \cap \boldsymbol{X}$ and $\text{vars}(\alpha_2) \cap \boldsymbol{X} = \text{vars}(\alpha'_2) \cap \boldsymbol{X}$.

Now, since $\text{vars}(\alpha''_1) = \text{vars}(\alpha_1) \cup \text{vars}(\alpha'_1)$ and $\text{vars}(\alpha''_2) = \text{vars}(\alpha_2) \cup \text{vars}(\alpha'_2)$, we have that:

$$\text{vars}(\alpha'') \cap \boldsymbol{X} = (\text{vars}(\alpha) \cap \boldsymbol{X}) \cup (\text{vars}(\alpha') \cap \boldsymbol{X}) = \text{vars}(\alpha) \cap \boldsymbol{X}$$
$$\text{vars}(\alpha''_1) \cap \boldsymbol{X} = (\text{vars}(\alpha_1) \cap \boldsymbol{X}) \cup (\text{vars}(\alpha'_1) \cap \boldsymbol{X}) = \text{vars}(\alpha_1) \cap \boldsymbol{X}$$
$$\text{vars}(\alpha''_2) \cap \boldsymbol{X} = (\text{vars}(\alpha_2) \cap \boldsymbol{X}) \cup (\text{vars}(\alpha'_2) \cap \boldsymbol{X}) = \text{vars}(\alpha_2) \cap \boldsymbol{X}$$

By compatibility of $C, C_{\text{other}}$, we have that $\text{vars}(\alpha_{\text{other}_1}) \cap \boldsymbol{X} = \text{vars}(\alpha_1) \cap \boldsymbol{X}$ and $\text{vars}(\alpha_{\text{other}_2}) \cap \boldsymbol{X} = \text{vars}(\alpha_2) \cap \boldsymbol{X}$. Thus $\text{vars}(\alpha_{\text{other}_1}) \cap \boldsymbol{X} = \text{vars}(\alpha''_1) \cap \boldsymbol{X}$ and $\text{vars}(\alpha_{\text{other}_2}) \cap \boldsymbol{X} = \text{vars}(\alpha''_2) \cap \boldsymbol{X}$. This shows $\boldsymbol{X}$-compatibility of $C'', C_{\text{other}}$.

**Example 4** (Counterexample to (5.6) for Compatibility). *While $\boldsymbol{X}$-compatibility is maintained through multiplying compatible circuits, the same is not true for compatibility, due to the different variable overlaps between the circuits. For example, suppose that $C$ over variable sets $\boldsymbol{A}, \boldsymbol{B}, \boldsymbol{C}$ has product nodes with scope decomposing as $\alpha = \alpha_1(\boldsymbol{A}) \times \alpha_2(\boldsymbol{B} \cup \boldsymbol{C})$, and $C'$ over variable sets $\boldsymbol{A}, \boldsymbol{B}, \boldsymbol{D}$ has product nodes with scope decomposing as $\alpha' = \alpha'_1(\boldsymbol{A}) \times \alpha'_2(\boldsymbol{B} \cup \boldsymbol{D})$. Then these circuits are compatible (i.e. $\boldsymbol{A} \cup \boldsymbol{B}$-compatible), and their product is a circuit with product nodes with scope decomposing as $\alpha'' = \alpha'_1(\boldsymbol{A}) \times \alpha'_2(\boldsymbol{B} \cup \boldsymbol{C} \cup \boldsymbol{D})$. Now consider $C_{\text{other}}$ with product nodes with scope decomposing as $\alpha_{\text{other}} = \alpha_{\text{other}}(\boldsymbol{C}) \times \alpha_{\text{other}}(\boldsymbol{D})$. This is compatible with $\alpha$ and $\alpha'$, but not with $\alpha''$.*

**(5.7)** This holds by the same argument as (5.6).

**(5.8)** The circuit $C'$ obtained by applying an elementwise mapping to $C$ does not change the scopes of any node. Thus, if $C$ is compatible with $C_{\text{other}}$, then $C'$ is also compatible with $C_{\text{other}}$.

**X-support-compatibility**    Recall that two smooth and decomposable circuits $C, C_{\text{other}}$ over variables $\boldsymbol{V}, \boldsymbol{V}_{\text{other}}$ are $\boldsymbol{X}$-support-compatible for $\boldsymbol{X} \subseteq \boldsymbol{V} \cap \boldsymbol{V}_{\text{other}}$ if there is an isomorphism $\iota$ between the nodes $C[\boldsymbol{X}]$ and $C_{\text{other}}[\boldsymbol{X}]$, such that:

- For any node $\alpha \in C[\boldsymbol{X}]$, $\text{vars}(\alpha) \cap \boldsymbol{X} = \text{vars}(\iota(\alpha)) \cap \boldsymbol{X}$;

- For all sum nodes $\alpha = +_{i=1}^{k} \alpha_i \in C[\boldsymbol{X}]$, we have that $\text{supp}_{\boldsymbol{X}}(\alpha_i) \cap \text{supp}_{\boldsymbol{X}}(\iota(\alpha_j)) = \emptyset$ whenever $i \neq j$.

**(5.9)** Suppose that $C, C_{\text{other}}$ are $\boldsymbol{X}$-support-compatible; and let $\iota_{C_{\text{other}}, C}$ be the isomorphism from $C_{\text{other}}[\boldsymbol{X}]$ to $C[\boldsymbol{X}]$. We wish to show that $C_{\text{other}}, C'$ are $\boldsymbol{X}$-support-compatible where $C'$ is the output circuit from Algorithm 1 that aggregates $C$ over $\boldsymbol{W}$, where $\boldsymbol{W} \cap \boldsymbol{X} = \emptyset$.

We define the isomorphism as follows. Consider the set of nodes $C'[\boldsymbol{X}]$. Since $\boldsymbol{W} \cap \boldsymbol{X} = \emptyset$, these nodes are not scalars and so are not propagated away by Lines 3-4. Moreover, since the algorithm retains the node types and connectivity of the circuit, there is an isomorphism $\iota_{C, C'}$ between $C[\boldsymbol{X}]$ and $C'[\boldsymbol{X}]$. There is thus an isomorphism $\iota_{C_{\text{other}}, C'} := \iota_{C, C'} \circ \iota_{C_{\text{other}}, C}$ between $C_{\text{other}}[\boldsymbol{X}]$ and $C'[\boldsymbol{X}]$. It remains to show the two conditions.

Given a node $\alpha_{\text{other}} \in C_{\text{other}}$, let us write $\alpha := \iota_{C_{\text{other}}, C}(\alpha_{\text{other}})$ and $\alpha' := \iota_{C, C'}(\alpha)$. By $\boldsymbol{X}$-support compatibility of $C_{\text{other}}, C$, we have that $\text{vars}(\alpha_{\text{other}}) \cap \boldsymbol{X} = \text{vars}(\alpha) \cap \boldsymbol{X}$. By the proof of Theorem 1, we know that $\text{vars}(\alpha') = \text{vars}(\alpha) \setminus \boldsymbol{W}$. Since $\boldsymbol{W} \cap \boldsymbol{X} = \emptyset$, this implies that $\text{vars}(\alpha_{\text{other}}) \cap \boldsymbol{X} = \text{vars}(\alpha') \cap \boldsymbol{X}$ as required. For the second part, suppose that these are sum nodes, i.e. $\alpha_{\text{other}} = +_{i=1}^{k} \alpha_{\text{other},i}$, $\alpha = +_{i=1}^{k} \alpha_i$ and $\alpha' = +_{i=1}^{k} \alpha'_i$. We know by $\boldsymbol{X}$-support-compatibility that $\text{supp}_{\boldsymbol{X}}(\alpha_{\text{other},i}) \cap \text{supp}_{\boldsymbol{X}}(\alpha_j) = \emptyset$ whenever $i \neq j$. By the same argument as in (5.1), we have that $\text{supp}_{\boldsymbol{X}}(\alpha'_i) \subseteq \text{supp}_{\boldsymbol{X}}(\alpha_i)$ for all $i$. Thus we can conclude that $\text{supp}_{\boldsymbol{X}}(\alpha_{\text{other},i}) \cap \text{supp}_{\boldsymbol{X}}(\alpha'_j) = \emptyset$ whenever $i \neq j$. So $C_{\text{other}}, C'$ are $\boldsymbol{X}$-support-compatible.

**(5.10)** Suppose that $C$ over $\boldsymbol{V}$ and $C'$ over $\boldsymbol{V'}$ are both $\boldsymbol{X}$-support-compatible with $C_{\text{other}}$; write $\iota_{C_{\text{other}}, C}$ for the isomorphism from $C_{\text{other}}[\boldsymbol{X}]$ to $C$, and $\iota_{C_{\text{other}}, C'}$ for the isomorphism from $C_{\text{other}}[\boldsymbol{X}]$ to $C'$. We wish to show that $C_{\text{other}}, C''$ are $\boldsymbol{X}$-support-compatible where $C''$ is the output circuit

from Algorithm 3 that computes the product of the two support-compatible (i.e. $(\boldsymbol{V} \cup \boldsymbol{V}')$-support-compatible) circuits $C, C'$.

We define the isomorphism as follows. Consider the set of nodes $C''[\boldsymbol{X}]$. The algorithm for multiplying $C, C'$ makes use of the isomorphism $\iota_{C,C'}$ between $C[\boldsymbol{V} \cap \boldsymbol{V}']$ and $C'[\boldsymbol{V} \cap \boldsymbol{V}']$, with $C''[\boldsymbol{V} \cap \boldsymbol{V}']$ retaining the same connectivity and node types; thus there is an isomorphism $\iota_{C,C''}$ from $C[\boldsymbol{V} \cap \boldsymbol{V}']$ to $C''[\boldsymbol{V} \cap \boldsymbol{V}']$, also. Since $\boldsymbol{X} \subseteq (\boldsymbol{V} \cap \boldsymbol{V}')$, this isomorphism also holds between the circuits restricted to $\boldsymbol{X}$. Thus, we define the isomorphism $\iota = \iota_{C,C''} \circ \iota_{C_{\text{other}},C}$ between $C_{\text{other}}[\boldsymbol{X}]$ and $C''[\boldsymbol{X}]$. It remains to show the two conditions.

Given a node $\alpha_{\text{other}} \in C_{\text{other}}$, let us write $\alpha := \iota_{C_{\text{other}},C}(\alpha_{\text{other}})$, $\alpha' = \iota_{C,C'}(\alpha)$ and $\alpha'' := \iota_{C,C''}(\alpha)$. By $\boldsymbol{X}$-support-compatibility of $C_{\text{other}}, C$, we have that $\text{vars}(\alpha_{\text{other}}) \cap \boldsymbol{X} = \text{vars}(\alpha) \cap \boldsymbol{X}$. By support-compatibility of $C, C'$, we have that $\text{vars}(\alpha) \cap (\boldsymbol{V} \cap \boldsymbol{V}') = \text{vars}(\alpha') \cap (\boldsymbol{V} \cap \boldsymbol{V}')$ and so $\text{vars}(\alpha) \cap \boldsymbol{X} = \text{vars}(\alpha') \cap \boldsymbol{X}$, and both are equal to $\text{vars}(\alpha'') \cap \boldsymbol{X}$ since $\text{vars}(\alpha'') = \text{vars}(\alpha) \cup \text{vars}(\alpha')$ (as in Theorem 3). Thus $\text{vars}(\alpha_{\text{other}}) \cap \boldsymbol{X} = \text{vars}(\alpha'') \cap \boldsymbol{X}$ as required. For the second part, suppose that these are sum nodes, i.e. $\alpha_{\text{other}} = +_{i=1}^{k} \alpha_{\text{other},i}$, $\alpha = +_{i=1}^{k} \alpha_i$, $\alpha' = +_{i=1}^{k} \alpha'_i$ and $\alpha' = +_{i=1}^{k} \alpha''_i$. We know by $\boldsymbol{X}$-support-compatibility that $\text{supp}_{\boldsymbol{X}}(\alpha_{\text{other},i}) \cap \text{supp}_{\boldsymbol{X}}(\alpha_j) = \emptyset$ whenever $i \neq j$. By the same argument as in (5.3), we have that $\text{supp}_{\boldsymbol{X}}(\alpha'') \subseteq \text{supp}_{\boldsymbol{X}}(\alpha) \cap \text{supp}_{\boldsymbol{X}}(\alpha')$. Thus we can conclude that $\text{supp}_{\boldsymbol{X}}(\alpha_{\text{other},i}) \cap \text{supp}_{\boldsymbol{X}}(\alpha'') = \emptyset$. So $C_{\text{other}}, C''$ are $\boldsymbol{X}$-support-compatible.

**(5.11)** Suppose that $C, C_{\text{other}}$ are $\boldsymbol{X}$-support-compatible; and let $\iota_{C_{\text{other}},C}$ be the isomorphism from $C_{\text{other}}[\boldsymbol{X}]$ to $C[\boldsymbol{X}]$. We wish to show that $C_{\text{other}}, C'$ are $\boldsymbol{X}$-support-compatible where $C'$ is the output circuit from Algorithm 4 that applies an elementwise mapping $\tau$ to $C$. Algorithm 4 maps each node $\alpha \in C$ to another node $\alpha' \in C$, keeping the node type and connectivity; this defines an isomorphism $\iota_{C,C'}$ from $C[\boldsymbol{X}]$ to $C'[\boldsymbol{X}]$. Thus we have an isomorphism $\iota_{C_{\text{other}},C'} := \iota_{C,C'} \circ \iota_{C_{\text{other}},C}$. It remains to show the two conditions.

Given a node $\alpha_{\text{other}} \in C_{\text{other}}$, let us write $\alpha := \iota_{C0,C}(\alpha_{\text{other}})$ and $\alpha' := \iota_{C,C'}(\alpha)$. By $\boldsymbol{X}$-support-compatibility of $C_{\text{other}}, C$, we have that $\text{vars}(\alpha_{\text{other}}) \cap \boldsymbol{X} = \text{vars}(\alpha) \cap \boldsymbol{X}$. The mapping algorithm does not change the scope of the nodes, i.e. $\text{vars}(\alpha') = \text{vars}(\alpha)$, so we have that $\text{vars}(\alpha_{\text{other}}) \cap \boldsymbol{X} = \text{vars}(\alpha') \cap \boldsymbol{X}$ as required. For the second part, suppose that these are sum nodes, i.e. $\alpha_{\text{other}} = +_{i=1}^{k} \alpha_{\text{other},i}$, $\alpha = +_{i=1}^{k} \alpha_i$ and $\alpha' = +_{i=1}^{k} \alpha'_i$. We know by $\boldsymbol{X}$-support-compatibility that $\text{supp}_{\boldsymbol{X}}(\alpha_{\text{other},i}) \cap \text{supp}_{\boldsymbol{X}}(\alpha_j) = \emptyset$ whenever $i \neq j$. We know by the same argument as in (5.4) that $\text{supp}_{\boldsymbol{X}}(\alpha'_i) \subseteq \text{supp}_{\boldsymbol{X}}(\alpha_i)$ for all $i$. Thus we can conclude that $\text{supp}_{\boldsymbol{X}}(\alpha_{\text{other},i}) \cap \text{supp}_{\boldsymbol{X}}(\alpha'_j) = \emptyset$ whenever $i \neq j$. So $C_{\text{other}}, C'$ are $\boldsymbol{X}$-support-compatible. $\square$

**Theorem 7** (Hardness of 2AMC with $\boldsymbol{X}$-firstness). *2AMC is #P-hard, even for circuits that are smooth, decomposable, deterministic, and $\boldsymbol{X}$-first, and a constant-time elementwise mapping.*

*Proof.* Take a DNF $\phi$ with terms $\phi_1, \dots, \phi_m$ over variables $X_1, \dots, X_n$. Let $l = \lceil \log m \rceil + 1$. Let us construct another DNF $\phi'$ with terms $\phi'_1, \dots, \phi'_m$ over variables $X_1 \dots, X_n$ and $Y_1, \dots, Y_{l+1}$ such that each $\phi'_i$ is the conjunction of $\phi_i$, $Y_{l+1}$ and a term over $Y_1, \dots, Y_l$ encoding a binary representation of $i$. For example:

$$\phi'_5 = \phi_5 \wedge Y_1 \wedge \neg Y_2 \wedge Y_3 \wedge \neg Y_4 \wedge \cdots \wedge \neg Y_l \wedge Y_{l+1}.$$

Now, efficiently manipulate $\phi'$ to make it smooth [15]. The circuit $\phi'$ is thus smooth, decomposable, deterministic and trivially satisfies X-firstness (since the children to every $\wedge$-gate are literals). Take the probability semiring as $\mathcal{S}_{\boldsymbol{X}}$, and $\mathcal{S}_{\boldsymbol{Y}} = (\mathbb{N}^2, +_2, \times_2, (0,0), (1,1))$ and $\tau((n1,n2)) = n1/n2$ (define $0/0 = 0$). Also, define $\omega(x) = 1$, and $\omega'(Y_{l+1} = 0) = (0,1)$ and $\omega'(y) = 1$ for all other literals. Then 2AMC counts the models of $\phi$, which is #P-hard [46]:

$$2AMC = \sum_{\boldsymbol{x}} \frac{\sum_{\boldsymbol{y}:y_{l+1}=1} \phi'(\boldsymbol{x}, \boldsymbol{y})}{\sum_{\boldsymbol{y}} \phi'(\boldsymbol{x}, \boldsymbol{y})} = \sum_{\boldsymbol{x}} \phi(\boldsymbol{x}),$$

where we assume $0/0 = 0$. The last equality follows because the circuit is deterministic (hence $\sum_{\boldsymbol{y}} \phi'(\boldsymbol{x}, \boldsymbol{y}) = \max_{\boldsymbol{y}} \phi'(\boldsymbol{x}, \boldsymbol{y}) \leq 1$) and logically equivalent to $\phi$ (i.e., $\forall \boldsymbol{x} : \phi(\boldsymbol{x}) = 1 \Leftrightarrow \exists \boldsymbol{y} : \phi'(\boldsymbol{x}, \boldsymbol{y}) = 1$). $\square$

**Theorem 8** (Tractability Conditions for 2AMC). *Every 2AMC instance is tractable in $O(|C|)$ time for Boolean circuits that are smooth, decomposable, deterministic, $\boldsymbol{X}$-first, and $\boldsymbol{X}$-deterministic.*

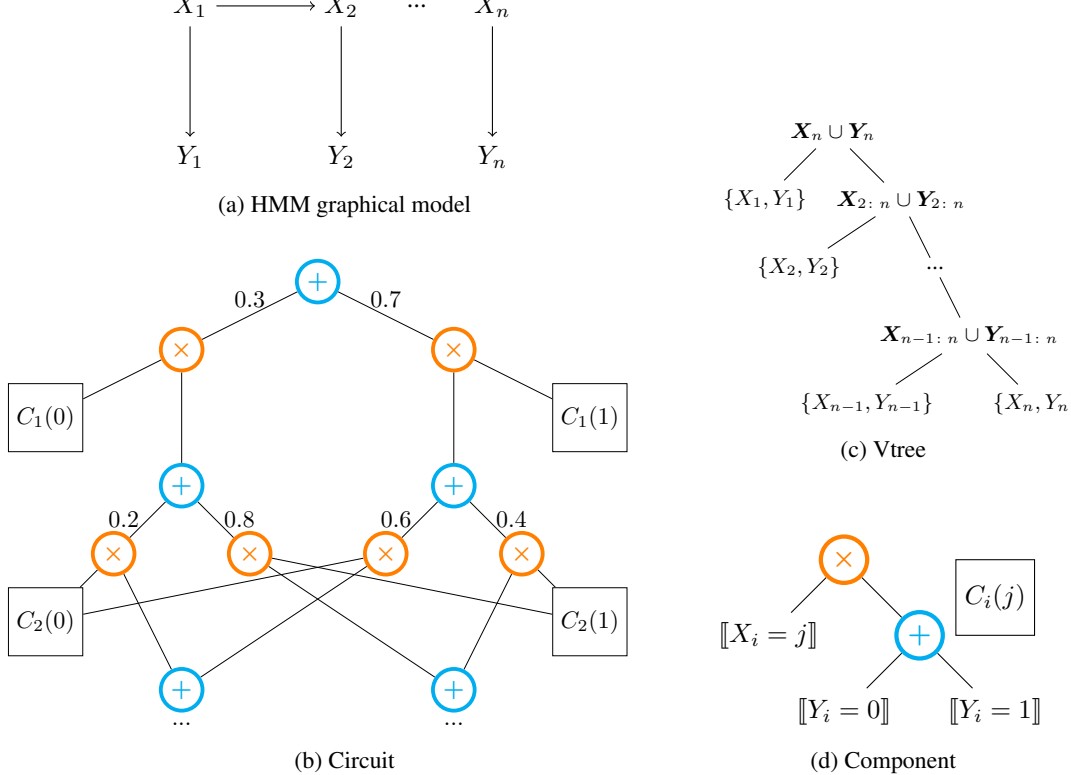

(a) HMM graphical model

(c) Vtree

(b) Circuit

(d) Component

Figure 4: Illustration of PC computing hidden Markov model (HMM)

---

**Algorithm 5:** 2AMC

---

**Input:** Decomposable, smooth, deterministic, $\boldsymbol{X}$-first and $\boldsymbol{X}$-deterministic logic circuit $C$ over
$\boldsymbol{X} \cup \boldsymbol{Y}$, weight circuits $\omega_{\boldsymbol{X}}, \omega_{\boldsymbol{Y}}$, semirings $\mathcal{S}_{\boldsymbol{X}}, \mathcal{S}_{\boldsymbol{Y}}$, mapping function $\tau_{\mathcal{S}_{\boldsymbol{Y}} \to \mathcal{S}_{\boldsymbol{X}}}$
**Output:** 2AMC value (scalar in semiring $\mathcal{S}_{\boldsymbol{X}}$)

1   $C_{\mathcal{S}_{\boldsymbol{Y}}}(\boldsymbol{X}, \boldsymbol{Y}) \leftarrow \texttt{MAPPING}(C(\boldsymbol{X}, \boldsymbol{Y}); [\![\cdot]\!]_{\mathcal{B} \to \mathcal{S}_{\boldsymbol{Y}}})$
2   $C_{\mathcal{S}_{\boldsymbol{Y}}, \omega_{\boldsymbol{Y}}}(\boldsymbol{X}, \boldsymbol{Y}) \leftarrow \texttt{PROD-CMP}(C_{\mathcal{S}_{\boldsymbol{Y}}}(\boldsymbol{X}, \boldsymbol{Y}), \omega_{\boldsymbol{Y}})$
3   $C_{\mathcal{S}_{\boldsymbol{Y}}, \omega_{\boldsymbol{Y}}}(\boldsymbol{X}) \leftarrow \texttt{AGG}(C_{\mathcal{S}_{\boldsymbol{Y}}, \omega_{\boldsymbol{Y}}}(\boldsymbol{X}, \boldsymbol{Y}); \boldsymbol{Y})$
4   $C_{\mathcal{S}_{\boldsymbol{X}}, \omega_{\boldsymbol{Y}}}(\boldsymbol{X}) \leftarrow \texttt{MAPPING}(C_{\mathcal{S}_{\boldsymbol{Y}}, \omega_{\boldsymbol{Y}}}(\boldsymbol{X}); \tau_{\mathcal{S}_{\boldsymbol{Y}} \to \mathcal{S}_{\boldsymbol{X}}})$
5   $C_{\mathcal{S}_{\boldsymbol{X}}, \omega_{\boldsymbol{Y}}, \omega_{\boldsymbol{X}}}(\boldsymbol{X}) \leftarrow \texttt{PROD-CMP}(C_{\mathcal{S}_{\boldsymbol{X}}}(\boldsymbol{X}), \omega_{\boldsymbol{X}})$
**Result:** $\texttt{AGG}(C_{\mathcal{S}_{\boldsymbol{X}}, \omega_{\boldsymbol{Y}}, \omega_{\boldsymbol{X}}}(\boldsymbol{X}); \boldsymbol{X})$

---

*Proof.* In Algorithm 5, we show the algorithm for 2AMC, which is simply a composition of aggregations, products, and elementwise mappings. To show tractability of 2AMC, we simply need to show that the input circuits to each of these operators satisfy the requisite tractability conditions.

We start with a smooth, decomposable, deterministic, $\boldsymbol{X}$-deterministic, and $\boldsymbol{X}$-first circuit $C(\boldsymbol{X}, \boldsymbol{Y})$.

- In line 1, we use the support mapping (Definition 6) from the Boolean to $\mathcal{S}_{\boldsymbol{Y}}$ semiring; this is tractable by Corollary 1 due to determinism, and the output $C_{\mathcal{S}_{\boldsymbol{Y}}}(\boldsymbol{X}, \boldsymbol{Y})$ retains all the properties by Table 3.

- In line 2, we take the product of $C_{\mathcal{S}_{\boldsymbol{Y}}}(\boldsymbol{X}, \boldsymbol{Y})$ and $\omega_{\boldsymbol{X}}(\boldsymbol{X})$. $\omega_{\boldsymbol{X}}$ is omni-compatible, so we can apply $\texttt{PROD-CMP}$. This results in a circuit $C_{\mathcal{S}_{\boldsymbol{Y}}, \omega_{\boldsymbol{Y}}}(\boldsymbol{X}, \boldsymbol{Y})$ that is smooth, decomposable and $\boldsymbol{X}$-first. $\omega_{\boldsymbol{X}}(\boldsymbol{X})$ is both deterministic and $\boldsymbol{X}$-deterministic as it has no sum nodes, so this output circuit is also deterministic and $\boldsymbol{X}$-deterministic by (5.2).

- In line 3, we aggregate $C_{\mathcal{S}_{\boldsymbol{Y}}, \omega_{\boldsymbol{Y}}}(\boldsymbol{X}, \boldsymbol{Y})$ over $\boldsymbol{Y}$. The output circuit $C_{\mathcal{S}_{\boldsymbol{Y}}, \omega_{\boldsymbol{Y}}}(\boldsymbol{X})$ is smooth and decomposable. It is also $\boldsymbol{X}$-deterministic by (5.1), as $\boldsymbol{Y} \cap \boldsymbol{X} = \emptyset$.

Since $C_{\mathcal{S}_{\boldsymbol{Y}},\omega_{\boldsymbol{Y}}}(\boldsymbol{X},\boldsymbol{Y})$ satisfied $\boldsymbol{X}$-firstness, each product node $\alpha = \alpha_1 \times \alpha_2$ in that circuit had at most one child (say $\alpha_1$) with scope overlapping with $\boldsymbol{Y}$. Then, in the product in the previous step, $\alpha_2$ must have been produced through Lines 1-2 (otherwise it would contain some variable in $\boldsymbol{Y}$); thus it was produced by applying $[\![\cdot]\!]_{\mathcal{B}\to\mathcal{S}_{\boldsymbol{Y}}}$ to some node in $C$. Thus, for any value $\boldsymbol{v} \in \mathrm{Assign}(\alpha_2)$, $p_{\alpha_2} \in \{0_{\mathcal{S}_{\boldsymbol{Y}}}, 1_{\mathcal{S}_{\boldsymbol{Y}}}\}$. So (Prod 0/1) is satisfied.

- In line 4, we apply the mapping $\tau_{\mathcal{S}_{\boldsymbol{Y}}\to\mathcal{S}_{\boldsymbol{X}}}$ to $C_{\mathcal{S}_{\boldsymbol{Y}},\omega_{\boldsymbol{Y}}}(\boldsymbol{X})$. This circuit is over $\boldsymbol{X}$ and is $\boldsymbol{X}$-deterministic, i.e. deterministic and satisfies (Additive). As shown in the previous step, it also satisfies (Prod 0/1). Thus the mapping algorithm produces the correct result, producing a smooth, decomposable and determinsitic circuit $C_{\mathcal{S}_{\boldsymbol{X}},\omega_{\boldsymbol{Y}}}(\boldsymbol{X})$ as output.

- In line 5, we take the product of $C_{\mathcal{S}_{\boldsymbol{X}},\omega_{\boldsymbol{Y}}}(\boldsymbol{X})$ with $\omega_{\boldsymbol{X}}(\boldsymbol{X})$. $\omega_{\boldsymbol{X}}$ is omni-compatible so we can apply PROD-CMP, producing a circuit $C_{\mathcal{S}_{\boldsymbol{X}},\omega_{\boldsymbol{Y}},\omega_{\boldsymbol{X}}}$ that is smooth and decomposable (and also deterministic).

- Finally, we aggregate $C_{\mathcal{S}_{\boldsymbol{X}},\omega_{\boldsymbol{Y}},\omega_{\boldsymbol{X}}}(\boldsymbol{X})$ over $\boldsymbol{X}$, producing a scalar.

$\square$

**Theorem 9** (Exponential Separation). *Given sets of variables $\boldsymbol{X} = \{X_1, ..., X_n\}$, $\boldsymbol{Y} = \{Y_1, ..., Y_n\}$, there exists a smooth, decomposable and $\boldsymbol{X}$-deterministic circuit $C$ of size $poly(n)$ such that the smallest smooth, decomposable, and $\boldsymbol{X}$-first circuit $C'$ such that $p_C \equiv p_{C'}$ has size $2^{\Omega(n)}$.*

*Proof.* Consider representing the distribution given by a hidden Markov model (HMM) over (hidden) variables $X_{\leq n} = \{X_1, ..., X_n\}$ and (observed) variables $Y_{\leq n} = \{Y_1, ..., Y_n\}$, as depicted in Figure 4a. Figure 4b shows a structured decomposable circuit that computes the hidden Markov model distribution, where the components $C_i(j)$ have scope $\{X_i, Y_i\}$. The corresponding vtree/scope-decomposition (with nodes notated using their scopes) is shown in Figure 4c. It can easily be checked that the circuit is $X_{\leq n}$-deterministic, and that the circuit size is linear in $n$.

It remains to show that the smallest $X_{\leq n}$-first and $X_{\leq n}$-deterministic circuit computing the HMM distribution is exponential in size. Explicitly, we will choose a HMM such that the emission distribution is given by $p(Y_i|X_i) = \mathbb{1}_{Y_i=X_i}$. Then we have that $p_{C'}(x_{\leq n}, Y_{\leq n}) = p_{C'}(x_{\leq n})p_{C'}(Y_{\leq n}|x_{\leq n}) = p_{C'}(x_{\leq n})\mathbb{1}_{Y_{\leq n}=x_{\leq n}}$, for any circuit $C'$ that expresses the distribution of the HMM.

Consider any such circuit $C'$. Then, let $\boldsymbol{\alpha} = \{\alpha_1, ..., \alpha_K\}$ be the set of nodes with scope $Y_{\leq n}$ in the circuit. We will need the following lemma:

**Lemma 2.** *For any value $x_{\leq n}$ of $X_{\leq n}$, there exists constants $c_1, .., c_K \in \mathbb{R}^{\geq 0}$ such that:*

$$p_{C'}(x_{\leq n}, Y_{\leq n}) \equiv \sum_{k=1}^{K} c_k p_{\alpha_k}(Y_{\leq n}) \qquad (6)$$

In other words, the output of the circuit is a linear function of the nodes with scope $Y_{\leq n}$.

*Proof.* We show this proof by bottom-up induction (child before parent), for the set of nodes whose scope *contains* $Y_{\leq n}$:

- **Input node**: If the scope is $Y_{\leq n}$, then it must be some node $\alpha_k \in \boldsymbol{\alpha}$; then we take $c_k = 1$ and $c_{k'} = 0$ for all $k' \neq k$.

- **Sum node**: By smoothness, all the children must have the same scope (containing $Y_{\leq n}$). The sum node is then just a linear combination of its children, so the result holds by the inductive hypothesis.

- **Product node** $P$: Let $P_1, P_2$ be the children of $P$. By $X_{\leq n}$-firstness, either both children are pure (have scope entirely contained in $X_{\leq n}$ or $Y_{\leq n}$), or one of them is pure, and the scope of the other one (say $P_1$) contains $Y_{\leq n}$.

  In the first case, if there is exactly one node (say $P_1$), with scope contained in $Y_{\leq n}$, then it must have scope exactly $Y_{\leq n}$. Then we have that:

  $$p_P(x_{\leq n}, Y_{\leq n}) = p_{P_1}(Y_{\leq n})p_{P_2}(x_{\leq n} \cap \mathrm{vars}(P_2))$$

$p_{P_2}(x_{\leq n} \cap \text{vars}(P_2))$ here is a constant, so by the inductive hypothesis we are done. If both nodes have scope contained in $Y_{\leq n}$, then $P$ is in $\boldsymbol{\alpha}$, say $P = \alpha_k$. Then we set $c_k = 1$ and $c_{k'} = 0$ for $k' \neq k$.

In the second case, we have that:

$$p_P(x_{\leq n}, Y_{\leq n}) = p_{P_1}(x_{\leq n} \cap \text{vars}(P_1), Y_{\leq n}) p_{P_2}(x_{\leq n} \cap \text{vars}(P_2))$$

Here $p_{P_2}(x_{\leq n} \cap \text{vars}(P_2))$ is a constant, so by the inductive hypothesis we are done.

Note that $X_{\leq n}$-firstness was crucial to avoid the case where a product has two mixed nodes (containing variables in $X_{\leq n}$ and $Y_{\leq n}$) as children.

$\square$

For any $k = 1, .., K$, define $v_k \in \mathbb{R}_{\geq 0}^{2^n}$ to be the vector with entries $v_{k,i} = \alpha_k(i)$ (where we interpret $i$ as a value of $Y_{\leq n}$). Then we have the following Corollary:

**Corollary 2.** *The set of vectors $\{v_1, ..., v_K\}$ forms a spanning set for $\mathbb{R}^{2^n}$.*

*Proof.* By the Lemma and the fact that $C'$ expresses the HMM distribution, we have that for any $x_{\leq n} \in \{0,1\}^n$, there exists $c_1, .., c_k \in \mathbb{R}^{\geq 0}$ such that:

$$p_{C'}(x_{\leq n}) \mathbb{1}_{Y_{\leq n} = x_{\leq n}} \equiv \sum_{k=1}^{K} c_k p_{\alpha_k}(Y_{\leq n})$$

Rearranging, and writing in vector form, we have:

$$e_{x_{\leq n}} = \sum_{k=1}^{K} \frac{c_k}{p_{C'}(x_{\leq n})} v_k$$

where $e_{x_{\leq n}} \in \mathbb{R}_{\geq 0}^{2^n}$ is the standard basis vector corresponding to the value $x_{\leq n}$. Thus $\{v_1, ..., v_K\}$ is a spanning set. $\square$

Any spanning set for $\mathbb{R}^{2^n}$ must contain at least $2^n$ elements. Thus, $K \geq 2^n$, and the circuit $C'$ must be exponentially sized. $\square$

One might attempt to remedy the situation by replacing $\boldsymbol{X}$-firstness with $\boldsymbol{X}$-determinism. For the general case, that however is insufficient:

**Theorem 10** (Hardness of 2AMC with $\boldsymbol{X}$-determinism). *2AMC is #P-hard even for decomposable, smooth, deterministic and $\boldsymbol{X}$-deterministic circuits, and a constant-time elementwise transformation function.*

*Proof.* By reduction from the counting version of number partitioning: Given positive integers $k_1, \ldots, k_n$, count the number of index sets $S \subseteq \{1, \ldots, n\}$ such that $\sum_{i \in S} k_i = \sum_{i \notin S} k_i = c$. That problem is known to be #P-hard [47]. Define $\phi = \bigwedge_{i=1}^{n}(X_i \Leftrightarrow Y_i)$. Then $\phi$ is a deterministic, $\boldsymbol{X}$-deterministic, decomposable and smooth circuit.[4] Let the inner labeling function be $\omega'(y_i) = k_i/c$ and $\omega'(\neg y_i) = 1$. Then for a fixed configuration $x$ of the variables $X = \{X_1, \ldots, X_n\}$, we have exactly one model for $\phi$, whose value is $\otimes_{i:x_i=1} k_i/c$. If we select the inner semiring so that $\otimes$ is addition (e.g., the max tropical semiring or log semiring), then the inner AMC problem returns $\sum_{i:x_i=1} k_i/c$, which equals 1 iff $S = \{i : x_i = 1\}$ is a solution to the number partitioning instance. Now, define the outer labeling function to be $\omega = 1$, and let the transformation function be $\tau(s) = 1$ if $s = 1$ and $\tau(s) = 0$ otherwise. Then the 2AMC problem with the probability semiring as outer semiring counts the number of solutions of the number partitioning instance. $\square$

---

[4]While this circuit is not $\boldsymbol{X}$-first, it does satisfy a property known as $\boldsymbol{X}$-firstness modulo definability [29]; thus that property is insufficient for 2AMC even together with $\boldsymbol{X}$-determinism.

Table 4: Tractability Conditions and Complexity for Compositional Inference Problems. We denote new results with an asterisk.

| | Problem | Tractability Conditions | Complexity |
|---|---|---|---|
| **2AMC** | PASP (Max-Credal)* | Sm, Dec, $\boldsymbol{X}$-Det | $O(|C|)$ |
| | PASP (MaxEnt)*, MMAP | Sm, Dec, Det, $\boldsymbol{X}$-Det | $O(|C|)$ |
| | SDP* | Sm, Dec, Det, $\boldsymbol{X}$-Det, $\boldsymbol{X}$-First | $O(|C|)$ |
| **Causal Inference** | Backdoor* | Sm, Dec, SD, $(\boldsymbol{X} \cup \boldsymbol{Z})$-Det | $O(|C|^2)$ |
| | | Sm, Dec, $\boldsymbol{Z}$-Det, $(\boldsymbol{X} \cup \boldsymbol{Z})$-Det | $O(|C|)$ |
| | Frontdoor* | Sm, Dec, SD, $\boldsymbol{X}$-Det, $(\boldsymbol{X} \cup \boldsymbol{Z})$-Det | $O(|C|^2)$ |
| **Other** | MFE* | Sm, Dec, $\boldsymbol{H}$-Det, $\boldsymbol{I}^-$-Det, $(\boldsymbol{H} \cup \boldsymbol{I}^-)$-Det | $O(|C|)$ |
| | Reverse-MAP | Sm, Dec, $\boldsymbol{X}$-Det | $O(|C|)$ |

# B    Case Studies

In this section, we provide more details about the compositional inference problems in Table 2 (reproduced in Table 4) for convenience, and prove the tractability conditions for each (Theorem 6). For all of them, we assume that we are given a Boolean formula represented as a circuit. That would usually come from knowledge compilation from some source language such as Bayesian Networks [9] or probabilistic logic programs [24]; our results thus show what properties the compiled circuit must have in order a query of interest to be tractable. Note that the problems are generally computationally hard [19, 10] on the source language, which means there do not exist compact circuits satsifying the properties in the worst-case.

**Theorem 6** (Tractability of Compositional Queries). *The results in Table 2 hold.*

## B.1    2AMC Queries

Firstly, we consider instances of 2AMC queries. Recall the general form of a 2AMC query. Given a partition of the variables $\boldsymbol{V} = (\boldsymbol{X}, \boldsymbol{Y})$, a Boolean function $\phi(\boldsymbol{X}, \boldsymbol{Y})$, *outer* and *inner* semirings $\mathcal{S}_{\boldsymbol{X}}, \mathcal{S}_{\boldsymbol{Y}}$, labeling functions $\omega_{\boldsymbol{Y}}(\boldsymbol{Y}) = \bigotimes_{Y_i \in \boldsymbol{Y}} \omega_{\boldsymbol{Y},i}(Y_i)$ over $\mathcal{S}$ and $\omega_{\boldsymbol{X}}(\boldsymbol{X}) = \bigotimes_{X_i \in \boldsymbol{X}} \omega_{\boldsymbol{X},i}(X_i)$ over $\mathcal{S}'$, and an elementwise mapping $\tau_{\mathcal{S}_{\boldsymbol{Y}} \to \mathcal{S}_{\boldsymbol{X}}} : \mathcal{S}_{\boldsymbol{Y}} \to \mathcal{S}_{\boldsymbol{X}}$, the 2AMC problem is given by:

$$\bigoplus_{\boldsymbol{x}} \left( \tau_{\mathcal{S}_{\boldsymbol{Y}} \to \mathcal{S}_{\boldsymbol{X}}} \left( \bigoplus_{\boldsymbol{y}} [\![\phi(\boldsymbol{x}, \boldsymbol{y})]\!]_{\mathcal{B} \to \mathcal{S}_{\boldsymbol{Y}}} \otimes \omega(\boldsymbol{y}) \right) \otimes \omega'(\boldsymbol{x}) \right) \qquad (1, \text{revisited})$$

By Theorem 8, any 2AMC problem is tractable if $\phi$ is given as a smooth, decomposable, deterministic, $\boldsymbol{X}$-deterministic, and $\boldsymbol{X}$-first circuit $C$. However, in some instances, we can relax these conditions, as we show shortly.

### B.1.1    Marginal MAP

In the *Marginal Maximum A Posteriori inference* (MMAP), we are given a Boolean function $\phi(\boldsymbol{V})$, a (unnormalized) fully factorized distribution $p(\boldsymbol{V}) = \prod_i p_i(V_i)$, a partition $\boldsymbol{X} \cup \boldsymbol{Y} = \boldsymbol{V}$ and some evidence $\boldsymbol{e}$ on $\boldsymbol{E} \subset \boldsymbol{V}$. The goal is to compute the probability of the maximum probability assignment of $\boldsymbol{X}$ consistent with $\boldsymbol{e}$:

$$\max_{\boldsymbol{x}} p(\boldsymbol{X} = \boldsymbol{x}, \boldsymbol{E} = \boldsymbol{e}) = \max_{\boldsymbol{x}} \sum_{\boldsymbol{y} \models \phi(\boldsymbol{x}, \boldsymbol{Y}) \wedge \boldsymbol{e}} \prod_i p_i(v_i).$$

To cast it as a 2AMC problem, take the inner semiring $\mathcal{S}_{\boldsymbol{Y}}$ to be the probability semiring and define the inner labelling function to assign $\omega_{\boldsymbol{Y}}(Y_i) = 0$ if $Y_i \in \boldsymbol{E}$ and $Y_i$ is *in*consistent with $\boldsymbol{e}$ and $\omega_{\boldsymbol{Y}}(Y_i) = p_i(Y_i)$ otherwise. The outer semiring is the $(\max, \cdot)$ semiring with labeling function $\omega_{\boldsymbol{X}}(X_i) = 1$. The elementwise mapping function $\tau_{\mathcal{S}_{\boldsymbol{Y}} \to \mathcal{S}_{\boldsymbol{X}}}(a) = a$ is the identity function.

The proof of the tractability conditions follows Theorem 8, except that we note that the mapping function $\tau_{\mathcal{S}_{\boldsymbol{Y}} \to \mathcal{S}_{\boldsymbol{X}}}$ from the outer to inner semiring satisifies (Multiplicative). As such, we do not need the (Prod 0/1) circuit property, which was the reason we needed the $\boldsymbol{X}$-firstness condition.

### B.1.2 Probabilistic Answer Set Programming (PASP)

The *Probabilistic Answer Set Programming Inference* (PASP) query takes a Boolean formula $\phi(\mathbf{V})$, a partition $\mathbf{X} \cup \mathbf{Y} = \mathbf{V}$, a (unnormalized) fully factorized distribution $p(\mathbf{X}) = \prod_i p(X_i)$, and query variable and value $\{Q = q\}$, for some $Q \in \mathbf{V}$. The goal is to compute:

$$p(Q = q) = \sum_{\mathbf{x}} \left( \prod_i p(X_i) \right) \sum_{\mathbf{y} \models \phi(\mathbf{x}, \mathbf{Y}) \wedge q} p^*(\mathbf{y}|\mathbf{x}).$$

The function $p^*(\mathbf{Y}|\mathbf{X})$ depends on the semantics adopted. Let $\mathrm{mod}(\mathbf{Y}|\mathbf{X}) := \{\mathbf{y} : \phi(\mathbf{X}, \mathbf{y})\}$ be the set of assignments of $\mathbf{Y}$ such that $\phi(\mathbf{X}, \cdot)$ is true. In the *Maximum Entropy Semantics* (MaxEnt) [6, 51, 45], one distributes the probability mass $p(\mathbf{X})$ uniformly over the models of $\phi$ consistent with $\mathbf{X}$, i.e. $p^*(\mathbf{y}|\mathbf{X}) = \frac{1}{|\mathrm{mod}(\mathbf{Y}|\mathbf{X})|}$. On the other hand, in the *Credal Semantics* [33, 14] (Max-Credal), one places all probability mass $p(\mathbf{X})$ on some assignment $\mathbf{y}$ of $\mathbf{Y}$ consistent with $\mathbf{X}$ and $q$. To obtain an upper bound on the query probability regardless of which $\mathbf{y}$ is chosen, one sets $p^*(\mathbf{y}|\mathbf{X}) := 1$ for all $\mathbf{y}$ if there exists an assignment $\mathbf{Y} \models \phi(\mathbf{X}, \mathbf{Y}) \wedge q$, and $p^*(\mathbf{Y}|\mathbf{X}) = 0$ otherwise.

The 2AMC formulation of the problem uses the probability semiring as outer semiring $\mathcal{S}_{\mathbf{X}}$, with labeling function $\omega_{\mathbf{X}}(X_i) = p(X_i)$ for $X_i \in \mathbf{X}$.

- In the (MaxEnt) semantics, for the inner semiring, we take as the semiring of pairs of naturals $\mathcal{S}_{\mathbf{Y}} = (\mathbb{N}^2, +, \cdot, (0,0), (1,1))$, with coordinatewise addition and multiplication. The inner labeling function sets $\omega_{\mathbf{Y}}(Q) = (\mathbb{1}_{Q=q}, 1)$, and sets $\omega_{\mathbf{Y}}(Y_i) = (1,1)$ for all other variables $Y_i \in \mathbf{Y}$. The mapping function is defined by $\tau_{\mathcal{S}_{\mathbf{Y}} \to \mathcal{S}_{\mathbf{X}}}((a,b)) = a/b$ (with $0/0 = 0$).

- In the (Max-Credal) semantics, we simply set the inner semiring to be the Boolean semiring $\mathcal{S}_{\mathbf{Y}} = \mathcal{B}$. The inner labeling function sets $\omega_{\mathbf{Y}}(Q) = \begin{cases} \top & \text{if } Q = q \\ \bot & \text{otherwise} \end{cases}$, and sets $\omega_{\mathbf{Y}}(Y_i) = \top$ for all other variables $Y_i \in \mathbf{Y}$. The mapping function is defined by $\tau_{\mathcal{S}_{\mathbf{Y}} \to \mathcal{S}_{\mathbf{X}}}(a) = [\![a]\!]_{\mathcal{S}_{\mathbf{Y}} \to \mathcal{S}_{\mathbf{X}}}$.

As with marginal MAP, we can see that in both cases, the mapping function $\tau_{\mathcal{S}_{\mathbf{Y}} \to \mathcal{S}_{\mathbf{X}}}$ satisfies (Multiplicative), so $\mathbf{X}$-firstness of the circuit is not required. In particular, for (MaxEnt) we have $\tau_{\mathcal{S}_{\mathbf{Y}} \to \mathcal{S}_{\mathbf{X}}}((a,b) \otimes (c,d)) = \tau_{\mathcal{S}_{\mathbf{Y}} \to \mathcal{S}_{\mathbf{X}}}((a \cdot c, b \cdot d)) = \frac{a \cdot c}{b \cdot d} = \frac{a}{b} \cdot \frac{c}{d} = \tau_{\mathcal{S}_{\mathbf{Y}} \to \mathcal{S}_{\mathbf{X}}}(a,b) \cdot \tau_{\mathcal{S}_{\mathbf{Y}} \to \mathcal{S}_{\mathbf{X}}}(c,d) = \tau_{\mathcal{S}_{\mathbf{Y}} \to \mathcal{S}_{\mathbf{X}}}(a,b) \otimes \tau_{\mathcal{S}_{\mathbf{Y}} \to \mathcal{S}_{\mathbf{X}}}(c,d)$ (this holds also if $(a,b) = (0,0)$ and/or $(c,d) = (0,0)$). Meanwhile, for (Max-Credal) we have $\tau_{\mathcal{S}_{\mathbf{Y}} \to \mathcal{S}_{\mathbf{X}}}(a \otimes b) = \tau_{\mathcal{S}_{\mathbf{Y}} \to \mathcal{S}_{\mathbf{X}}}(a \wedge b) = [\![a \wedge b]\!]_{\mathcal{S}_{\mathbf{Y}} \to \mathcal{S}_{\mathbf{X}}} = [\![a]\!]_{\mathcal{S}_{\mathbf{Y}} \to \mathcal{S}_{\mathbf{X}}} \cdot [\![b]\!]_{\mathcal{S}_{\mathbf{Y}} \to \mathcal{S}_{\mathbf{X}}} = \tau_{\mathcal{S}_{\mathbf{Y}} \to \mathcal{S}_{\mathbf{X}}}(a) \cdot \tau_{\mathcal{S}_{\mathbf{Y}} \to \mathcal{S}_{\mathbf{X}}}(b) = \tau_{\mathcal{S}_{\mathbf{Y}} \to \mathcal{S}_{\mathbf{X}}}(a) \otimes \tau_{\mathcal{S}_{\mathbf{Y}} \to \mathcal{S}_{\mathbf{X}}}(b)$.

For the (Max-Credal) semantics, we note additionally since $\mathcal{S}_{\mathbf{Y}}$ is just the Boolean semiring, we do *not* need determinism in Line 1 of Algorithm 5. So the only conditions required are smoothness, decomposability, and $\mathbf{X}$-determinism.

### B.1.3 Same-Decision Probability

In the *Same Decision Probability* (SDP) query [37], we are given a Boolean formula $\phi(\mathbf{V})$, a fully factorized distribution $p(\mathbf{V}) = \prod_i p(V_i)$, a partition $\mathbf{X}, \{Y\}$ of $\mathbf{V}$, a query $\{Y = y\}$, some evidence $\mathbf{e}$ on a subset $\mathbf{E} \subseteq \mathbf{X}$ of variables and a threshold value $T \in (0, 1]$. The goal is to compute a confidence measure on some threshold-based classification made with the underlying probabilistic model:

$$\sum_{\mathbf{x}} p(\mathbf{x}|\mathbf{e}) \mathbb{1}_{p(Y=y|\mathbf{x}, \mathbf{e}) \geq T},$$

To cast this as a 2AMC instance, we use the inner semiring $\mathcal{S}' = (\mathbb{R}^2_{\geq 0}, +, \cdot, (0,0), (1,1))$, with coordinate-wise addition and multiplication. The inner labeling function assigns $\omega_{\mathbf{Y}}(Y) = (p(Y)\mathbb{1}_{Y=y}, p(Y))$. The outer semiring is the probability semiring and the mapping $\tau_{\mathcal{S}_{\mathbf{Y}} \to \mathcal{S}_{\mathbf{X}}}$ from inner to outer semirings is $\tau_{\mathcal{S}_{\mathbf{Y}} \to \mathcal{S}_{\mathbf{X}}}((a,b)) = [\![a \geq bT]\!]$. Last, the outer labeling function assigns $\omega_{\mathbf{X}}(X_i) = \mathbb{1}_{X_i \models \mathbf{e}}$ if $X_i \in \mathbf{E}$, and $\omega_{\mathbf{X}}(X_i) = p(X_i)$ otherwise.

Unlike marginal MAP and PASP inference, there is no special structure in SDP that allows us to relax the general tractability conditions for 2AMC. However, it is still a 2AMC instance, and we have the tractability conditions from Theorem 8. In particular this justifies the use of $\mathbf{X}$-constrained sentential decision diagrams for this problem.

## B.2 Causal Inference

In Section 4.2, we discussed computing causal interventional distributions. In particular, in the backdoor and frontdoor cases, we had the following formulae:

$$p(\boldsymbol{y}|do(\boldsymbol{x})) = \sum_{\boldsymbol{z}} p(\boldsymbol{z})p(\boldsymbol{y}|\boldsymbol{x},\boldsymbol{z}), \tag{2}$$

$$p(\boldsymbol{y}|do(\boldsymbol{x})) = \sum_{\boldsymbol{z}} p(\boldsymbol{z}|\boldsymbol{x}) \sum_{\boldsymbol{x}'} p(\boldsymbol{x}')p(\boldsymbol{y}|\boldsymbol{x}',\boldsymbol{z}). \tag{3}$$

### B.2.1 Backdoor query

The backdoor query can be written as a compositional query as follows:

$$\texttt{BACKDOOR}(p;\boldsymbol{x},\boldsymbol{y}) := \bigoplus_{\boldsymbol{z}}\left(\left(\bigoplus_{\boldsymbol{x},\boldsymbol{y}} p(\boldsymbol{v})\right) \otimes p(\boldsymbol{v}) \otimes \tau_{-1}\left(\bigoplus_{\boldsymbol{y}} p(\boldsymbol{v})\right)\right). \tag{7}$$

where $\boldsymbol{V} = (\boldsymbol{X},\boldsymbol{Y},\boldsymbol{Z})$, and $\tau_{-1}(a) = \begin{cases} a^{-1} & \text{if } a \neq 0 \\ 0 & \text{if } a = 0 \end{cases}$. Note that $\tau_{-1}$ satisfies (Multiplicative),

and so for this mapping to be tractable we just need the circuit it is applied to to be deterministic.

Assume that $p(\boldsymbol{V})$ is given as a smooth, structured decomposable, and $(\boldsymbol{X} \cup \boldsymbol{Z})$-deterministic circuit (over the probabilistic semiring). We now show that this query is tractable, by showing that each operator in the composition is tractable. For readability, we label each circuit constructed with the function that it represents ( boxed ).

- $\boxed{p(\boldsymbol{X},\boldsymbol{Z})}$ $C_1(\boldsymbol{X},\boldsymbol{Z}) := \texttt{AGG}(C,\boldsymbol{Y})$ is tractable by smoothness and decomposability. By (5.1) in Table 3, since $\boldsymbol{Y} \cap (\boldsymbol{X} \cup \boldsymbol{Z}) = \emptyset$, $C_1$ is $(\boldsymbol{X} \cup \boldsymbol{Z})$-deterministic (i.e. deterministic).

- $\boxed{\frac{1}{p(\boldsymbol{X},\boldsymbol{Z})}}$ $C_2(\boldsymbol{X},\boldsymbol{Z}) := \texttt{MAPPING}(C_1,\tau_{-1})$ is tractable since $C_1$ is deterministic.

- $\boxed{p(\boldsymbol{Y}|\boldsymbol{X},\boldsymbol{Z})}$ $C_3(\boldsymbol{X},\boldsymbol{Y},\boldsymbol{Z}) := \texttt{PROD-SCMP}(C(\boldsymbol{X},\boldsymbol{Y},\boldsymbol{Z}),C_2(\boldsymbol{X},\boldsymbol{Z}))$. $C$ is $(\boldsymbol{X} \cup \boldsymbol{Z})$-support-compatible with itself as it is $(\boldsymbol{X} \cup \boldsymbol{Z})$-deterministic $\implies$ $C$ is also $(\boldsymbol{X} \cup \boldsymbol{Z})$-support-compatible with $C_1$ by (5.9) $\implies$ $C$ is also $(\boldsymbol{X} \cup \boldsymbol{Z})$-support-compatible with $C_2$ by (5.11). As $C$ and $C_2$ share variables $(\boldsymbol{X} \cup \boldsymbol{Z})$, this means they are support-compatible. Thus this product is tractable in linear time.

- $\boxed{p(\boldsymbol{Z})}$ $C_4(\boldsymbol{Z}) := \texttt{AGG}(C,\boldsymbol{X} \cup \boldsymbol{Y})$ is tractable by smoothness and decomposability.

- $\boxed{p(\boldsymbol{Z})p(\boldsymbol{Y}|\boldsymbol{X},\boldsymbol{Z})}$ $C_5(\boldsymbol{X},\boldsymbol{Y},\boldsymbol{Z}) := \texttt{PROD-CMP}(C_4,C_3)$. $C$ is $\boldsymbol{V}$-compatible with itself (structured decomposable) $\implies$ $C$ is $\boldsymbol{Z}$-compatible with itself by Proposition 1 $\implies$ $C$ is also $\boldsymbol{Z}$-compatible with $C_4$ by (5.5) $\implies$ $C_4$ is $\boldsymbol{Z}$-compatible with $C_1$ by (5.5) $\implies$ $C_4$ is $\boldsymbol{Z}$-compatible with $C_2$ by (5.8) $\implies$ $C_4$ is $\boldsymbol{Z}$-compatible with $C_3$ by (5.6). Since $C_4$ and $C_3$ share variables $\boldsymbol{Z}$, this means they are compatible and so this product is tractable in quadratic time.

- $\boxed{\sum_{\boldsymbol{z}} p(\boldsymbol{z})p(\boldsymbol{Y}|\boldsymbol{X},\boldsymbol{z})}$ $C_6(\boldsymbol{X},\boldsymbol{Y}) = \texttt{AGG}(C_5,\boldsymbol{Z})$ is tractable by smoothness and decomposability.

Thus, we have recovered the tractability conditions derived by [49], with the same complexity of $O(|C|^2)$ (induced by the compatible product to construct $C_5$). However, we also have an alternative tractability condition. Suppose that $C$ were additionally $\boldsymbol{Z}$-deterministic, but not necessarily structured decomposable. Then we could replace the derivation of $C_5$ above with the following:

- $\boxed{p(\boldsymbol{Z})p(\boldsymbol{Y}|\boldsymbol{X},\boldsymbol{Z})}$ $C_5(\boldsymbol{X},\boldsymbol{Y},\boldsymbol{Z}) := \texttt{PROD-SCMP}(C_4,C_3)$. $C$ is $\boldsymbol{Z}$-support-compatible with itself as it is $\boldsymbol{Z}$-deterministic $\implies$ $C$ is also $\boldsymbol{Z}$-support-compatible with $C_4$ by (5.9) $\implies$ $C_4$ is $\boldsymbol{Z}$-support-compatible with $C_1$ by (5.9) $\implies$ $C_4$ is $\boldsymbol{Z}$-compatible with $C_2$ by (5.11) $\implies$ $C_4$ is $\boldsymbol{Z}$-compatible with $C_3$ by (5.10). Since $C_4$ and $C_3$ share variables $\boldsymbol{Z}$, this means they are compatible and so this product is tractable in linear time.

In this case, the overall complexity is also reduced to $O(|C|)$.

### B.2.2 Frontdoor query

Now, consider the frontdoor case. In this case, we have the following compositional query:

$$\texttt{FRONTDOOR}(p; \boldsymbol{x}, \boldsymbol{y}, \boldsymbol{z}) = \bigoplus_{\boldsymbol{z}} \left( \left( \bigoplus_{\boldsymbol{y}} p(\boldsymbol{v}) \right) \otimes \tau_{-1} \left( \bigoplus_{\boldsymbol{y}, \boldsymbol{z}} p(\boldsymbol{v}) \right) \otimes \texttt{BACKDOOR}(p; \boldsymbol{z}, \boldsymbol{y}) \right) \quad (8)$$

Assume that $p(\boldsymbol{V})$ is given as a smooth, structured decomposable, $\boldsymbol{X}$-deterministic, and $(\boldsymbol{X} \cup \boldsymbol{Z})$-deterministic circuit (over the probabilistic semiring). We continue the analysis from the backdoor case:

- $\boxed{p(\boldsymbol{X})}$ $C_7(\boldsymbol{X}) := \texttt{AGG}(C, \boldsymbol{Y} \cup \boldsymbol{Z})$ is tractable by smoothness and decomposability. By (5.1) in Table 3, since $(\boldsymbol{Y} \cup \boldsymbol{Z}) \cap \boldsymbol{X} = \emptyset$, $C_7$ is $\boldsymbol{X}$-deterministic (i.e. deterministic).

- $\boxed{\frac{1}{p(\boldsymbol{X})}}$ $C_8(\boldsymbol{X}) := \texttt{MAPPING}(C_7, \tau_{-1})$ is tractable since $C_7$ is deterministic.

- $\boxed{p(\boldsymbol{Z}|\boldsymbol{X})}$ $C_9(\boldsymbol{X}, \boldsymbol{Z}) := \texttt{PROD-SCMP}(C_8, C_1)$. $C$ is $\boldsymbol{X}$-support-compatible with itself as it is $\boldsymbol{X}$-deterministic $\implies$ $C$ is $\boldsymbol{X}$-support-compatible with $C_1$ by (5.9) $\implies$ $C_1$ is $\boldsymbol{X}$-support-compatible with $C_7$ by (5.9) $\implies$ $C_1$ is $\boldsymbol{X}$-support-compatible with $C_8$ by (5.11). Thus this product is tractable in linear time.

- $\boxed{\sum_{\boldsymbol{x}} p(\boldsymbol{x}) p(\boldsymbol{Y}|\boldsymbol{x}, \boldsymbol{Z})}$ $C_{10}(\boldsymbol{Y}, \boldsymbol{Z})$. This is just like $C_6$, but with variables $\boldsymbol{X}$ and $\boldsymbol{Z}$ swapped. Thus it is tractable for a smooth, $\boldsymbol{X}$-deterministic and $(\boldsymbol{X} \cup \boldsymbol{Z})$-deterministic circuit in linear time.

- $\boxed{p(\boldsymbol{Z}|\boldsymbol{X}) \sum_{\boldsymbol{x}'} p(\boldsymbol{x}') p(\boldsymbol{Y}|\boldsymbol{x}', \boldsymbol{Z})}$ $C_{11}(\boldsymbol{X}, \boldsymbol{Y}, \boldsymbol{Z}) := \texttt{PROD-CMP}(C_9, C_{10})$. We can chain applications of (5.5), (5.7) and (5.8) in a similar way to the other steps to show that $C_9, C_{10}$ are $\boldsymbol{Z}$-compatible (i.e. compatible), so this product is tractable in quadratic time.

- $\boxed{\sum_{\boldsymbol{z}} p(\boldsymbol{z}|\boldsymbol{X}) \sum_{\boldsymbol{x}'} p(\boldsymbol{x}') p(\boldsymbol{Y}|\boldsymbol{x}', \boldsymbol{z})}$ $C_{12}(\boldsymbol{X}, \boldsymbol{Y}) := \texttt{AGG}(C_{11}; \boldsymbol{Z})$. This is tractable by smoothness and decomposability.

Thus, this algorithm has complexity $O(|C|^2)$, as opposed to the $O(|C|^3)$ complexity algorithm in [49]. The key difference is that we exploit support compatibility for a linear time product when constructing $C_{10}$.

### B.3 Other Problems

#### B.3.1 Most Frugal Explanation

In [31], the most frugal explanation (MFE) query was introduced. Given a partition of variables $\boldsymbol{V}$ into $(\boldsymbol{H}, \boldsymbol{I}^+, \boldsymbol{I}^-, \boldsymbol{E})$, some evidence $\boldsymbol{e} \in \text{Assign}(\boldsymbol{E})$, and a probability distribution $p(\boldsymbol{V})$, the MFE query asks for the following:

$$\max_{\boldsymbol{h}} \sum_{\boldsymbol{i}^-} \mathbb{1}[\boldsymbol{h} \in \arg\max_{\boldsymbol{h}'} p(\boldsymbol{h}', \boldsymbol{i}^-, \boldsymbol{e})] \quad (9)$$

In words, we want the explanation (assignment to $\boldsymbol{H}$) that is the most probable for the most number of assignments to $\boldsymbol{I}^-$, when $\boldsymbol{I}^+$ is marginalized out. We can rewrite as follows:

$$\max_{\boldsymbol{h}} \sum_{\boldsymbol{i}^-} \mathbb{1}\left[ \frac{p(\boldsymbol{h}, \boldsymbol{i}^-, \boldsymbol{e})}{\max_{\boldsymbol{h}'} p(\boldsymbol{h}', \boldsymbol{i}^-, \boldsymbol{e})} = 1 \right] \quad (10)$$

This can be written as a compositional query as follows.

$$\bigoplus_{\boldsymbol{h}} \tau_{\mathcal{S}''' \to \mathcal{S}'} \bigoplus_{\boldsymbol{i}^-} \tau_{\mathcal{S}'' \to \mathcal{S}'''} \left( \tau_{-1} \left( \tau_{\mathcal{S}' \to \mathcal{S}''} \left( \bigoplus_{\boldsymbol{h}'} \tau_{\mathcal{S} \to \mathcal{S}'} (p(\boldsymbol{h}', \boldsymbol{i}^-, \boldsymbol{e})) \right) \right) \otimes p(\boldsymbol{h}, \boldsymbol{i}^-, \boldsymbol{e}) \right) \quad (11)$$

where $\mathcal{S}$ is the probability semiring, $\mathcal{S}'$ is the $(\max, \cdot)$-semiring, $\mathcal{S}''$ is $([0, 1], +, \cdot, 0, 1)$ (i.e. the probability semiring with domain $[0, 1]$), and $\mathcal{S}'''$ is the counting semiring $(\mathbb{N}, +, \cdot, 0, 1)$, and the mapping functions are defined as follows:

- $\tau_{\mathcal{S} \to \mathcal{S}'}(a) = a$
- $\tau_{\mathcal{S}' \to \mathcal{S}''}(a) = a$
- $\tau_{-1}(a) = \begin{cases} a^{-1} & \text{if } a \neq 0 \\ 0 & \text{if } a = 0 \end{cases}$
- $\tau_{\mathcal{S}'' \to \mathcal{S}'''}(a) = \mathbb{1}_{a=1}$
- $\tau_{\mathcal{S}''' \to \mathcal{S}'}(a) = a$

Suppose we are given a probabilistic circuit representing $p(\boldsymbol{H}, \boldsymbol{I}^-, \boldsymbol{e})$. While this query appears extremely intimidating at first glance, we note that the only operators we need to consider are the mappings and single product. Note that all of these mappings satisfy (Multiplicative) ($\tau_{\mathcal{S}'' \to \mathcal{S}'''}$ because the domain of $\mathcal{S}''$ is $[0,1]$ so $\tau_{\mathcal{S}'' \to \mathcal{S}'''}(a \cdot b) = 1$ iff $a = b = 1$); thus the mappings are tractable if the input circuits are deterministic. By checking the scopes of the inputs to each mapping, we can see that $(\boldsymbol{H} \cup \boldsymbol{I}^-)$-determinism, $\boldsymbol{I}^-$-determinism, and $\boldsymbol{H}$-determinism suffices. This also enables tractability of the product in linear time by support compatibility.

No tractability conditions for exact inference for this query were previously known. While the motivation behind the MFE query is as a means of approximating marginal MAP, and so this exact algorithm is not practically useful in this case, this example illustrates the power of the compositional framework to tackle even very complex queries.

### B.3.2  Reverse MAP

Recently, in [27], the reverse-MAP query was introduced, defined by:

$$\max_{\boldsymbol{X}} p(\boldsymbol{e_1} | \boldsymbol{X}, \boldsymbol{e_2}) \tag{12}$$

where the variables are partitioned as $\boldsymbol{V} = (\boldsymbol{E_1}, \boldsymbol{E_2}, \boldsymbol{X}, \boldsymbol{H})$. In our compositional framework, this can be written as:

$$\bigoplus_{\boldsymbol{x}} \tau_{\mathcal{P} \to \mathcal{M}} \Big( \bigoplus_{\boldsymbol{h}} p(\boldsymbol{e_1}, \boldsymbol{x}, \boldsymbol{e_2}, \boldsymbol{h}) \otimes \tau_{-1} \big( \bigoplus_{\boldsymbol{h}, \boldsymbol{e_1'}} p(\boldsymbol{e_1'}, \boldsymbol{x}, \boldsymbol{e_2}, \boldsymbol{h}) \big) \Big) \tag{13}$$

Here, the mapping $\tau_{-1}$ is tractable if the circuit for $p$ is $\boldsymbol{X}$-deterministic. Since $p$ is $\boldsymbol{X}$-deterministic, it is $\boldsymbol{X}$-support-compatible with itself; chaining this with (5.9) and (5.11) in Table 3, the inputs to the product are $\boldsymbol{X}$-compatible; since they have scope $\boldsymbol{X}$, this means the product is tractable by support-compatibility. The resulting circuit remains $\boldsymbol{X}$-deterministic (i.e. deterministic as the scope is $\boldsymbol{X}$), which means that the mapping $\tau_{\mathcal{P} \to \mathcal{M}}$ from the probability to $(\max, \cdot)$ semiring is tractable. Thus, this query is tractable for smooth, decomposable and $\boldsymbol{X}$-deterministic circuits in linear time (same as derived by the authors).

