# OpenReview forum: "A Compositional Atlas for Algebraic Circuits"
_NeurIPS.cc/2024/Conference — NeurIPS 2024 poster_

### Official Review · Reviewer_oPB2 · 2024-07-08

**Soundness:** 4
**Presentation:** 3
**Contribution:** 2
**Rating:** 6
**Confidence:** 3

**Summary:**

This paper unifies two lines of work:
1) a compositional approach to tractability of queries over logical [12], probabilistic [35] and causal circuits [36] and
2) a (commutative) semiring-based perspective over different computational tasks ([12] among many others).

The benefit is the characterization of certain probabilistic and causal tasks in terms of combinations of simple operations on algebraic circuits. Concretely, this approach is used to derive new complexity results in second-level algebraic model counting (including highlighting a wrong claim in [20]) and causal inference, among others.

**update after rebuttal**

The response clarified a bit the scope of the contribution. I think that this is solid work overall and that adding parts of their response to the main text would make the paper more accessible. I raised my score accordingly.

**Strengths:**

- The paper is overall well-written, albeit dense (which I think it is a necessity due to the limited space)
- The novel properties of AC introduced here could lead to more effective compilation algorithms

**Weaknesses:**

- Novel and existing notions/definitions are mixed together, making it hard to give proper credit where it is due.
- The high-level contributions are not very clear from the text (in particular in the first sections). [35] clearly listed their contributions at the end of their introduction, in contrast, I found lines 39-42 more vague in this sense, giving the (wrong) idea that the compositional approach is a novelty.
- I think that the practical impact of this contribution is not be very clear to a general AI/ML audience.

**Questions:**

1. Could you clearly outline what are the novel notions that enable this generalized compositional framework?
2. Could you similarly outline what are the key ideas / existing work this work builds upon?
3. Could you provide examples (for the general NeurIPS audience rather than experts in AMC and AC) of the practical impact of this contribution?


## Minors

Multiple typos on Def. 1

**Limitations:**

The limitations of this work are adequately addressed in the conclusion.

---

> ### Author Rebuttal · Authors · 2024-08-07
>
> > The high-level contributions are not very clear from the text (in particular in the first sections). [35] clearly listed their contributions at the end of their introduction, in contrast, I found lines 39-42 more vague in this sense, giving the (wrong) idea that the compositional approach is a novelty.
>
> We will add a contributions statement at the end of the introduction. To summarize, the key contributions are (1) a more general compositional inference framework that covers circuits over arbitrary semirings, including logical and probabilistic circuits; (2) a conceptually simple, yet general formulation in terms of just three basic operations (aggregation, products, elementwise mappings), with associated tractability and composability conditions (Table 1); (3) novel and systematically derived tractability conditions for a range of existing inference problems (Table 2).
>
> > **Q2** Could you similarly outline what are the key ideas / existing work this work builds upon?
>
> Our work mainly draws on three lines of existing work.
>
> - Firstly, work on new structural properties for tractable inference on circuits (e.g. $\textbf{X}$-determinism [6] for marginal MAP and compatibility [35] for products).
> - Secondly, the compositional approach to inference [12, 35], which characterizes *tractability conditions* for individual operations  and which properties are maintained by the output of the operations (*composability conditions*).
> - Finally, algebraic model counting [21, 20], which casts many inference queries as aggregation over a particular semiring.
>
> Our conceptual contribution is to combine the generality of compositional inference with semiring structure. In the process, we make use of many of the recently introduced circuit properties (and extend them where required).
>
> > **Q1** Could you clearly outline what are the novel notions that enable this generalized compositional framework?
>
> The key novel notions that enable the generalized compositional framework are (1) the elementwise mapping operation and (2) extended composability conditions. In more detail:
>
> - **Elementwise Mapping:** A key novel contribution is our definition of elementwise mappings as a *modular operation*, which covers e.g. the (implicit) transformation from the Boolean circuit in algebraic model counting, 2AMC when moving from the inner to the outer semiring, and various mappings on the probability semiring such as reciprocals.
>
>     * This abstraction is crucial in that it allows us to extend the compositional approach to inference to a much wider range of problems that were not addressed in [35]. In contrast to prior works on algebraic inference, this is independent of particular formalisms (e.g. labeling functions in AMC), whether the input circuit is a logic circuit (DNNF) or PC, etc.
>
>     * In terms of the practical impact, the compositional approach provides a *simple* and *robust* recipe for finding sufficient tractability conditions: simply write down the inference query in math, and apply Table 1 to each operation.  For example, this enabled us to discover a mistake in the 2AMC tractability conditions [20], and elegantly extends compositional inference for probabilistic circuits to maximization, which was mentioned in the conclusion of the PC atlas paper [35] as an open problem.
>
> - **Composability Conditions:** A limitation of [35] is that the analysis of *composability conditions* was incomplete. In particular, there was no result on transitivity of compatibility (in fact, we show in Example 3 in the Appendix that this is actually not true). This is critical for the proofs of tractability for the queries in Table 2 using the compositional approach. To this end, we introduced $\textbf{X}$-compatibility and $\textbf{X}$-support-compatibility as new properties which do satisfy ``transitivity'' conditions, and prove these in Table 1/Section 3.2.
>
> > **Q3** Could you provide examples (for the general NeurIPS audience rather than experts in AMC and AC) of the practical impact of this contribution?
>
> The compositional queries we consider in this paper have many applications which are of interest to the NeurIPS community. For example, PCs have been used to compute interventional distributions in causal inference [36]. By reducing the complexity of backdoor adjustment for PCs from quadratic to linear (given $\textbf{Z}$-determinism), we open up the possibility of scaling up PC models for causal inference. The probabilistic ASP queries we consider are of significant interest in neuro-symbolic AI
> [b,c,d], where circuit representations are widely used to encode symbolic knowledge. Our work clarifies which circuit types can be used depending on the desired semantics.
>
> Given the connections that we establish between these disparate queries, we believe that our work will motivate more research on compiling as well as learning expressive circuits satisfying the properties that we identify.
>
> We believe that our work will also open up new applications for circuits within ML. For example, recent work has examined using circuits for generating text satisfying logical conditions [a], with quality comparable to LLM approaches; this relies on tractable products between the logical and probabilistic circuits. The PC used is a HMM, which is $\textbf{X}$-deterministic where $\textbf{X}$ are the hidden states (e.g. Figure 4 in the Appendix); our compositional algebraic atlas shows that it would be tractable to find the most likely hidden state conditional on the logical constraint, which could be used for model explainability.
>
> [a] Zhang et al. ``Tractable Control for Autoregressive Language Generation'' ICML 2023.
>
> [b] Zhun et al. ``NeurASP: Embracing Neural Networks into Answer Set Programming'' IJCAI 2020.
>
> [c] Mahaeve et al. ``Neural probabilistic logic programming in DeepProbLog'' AIJ 2021.
>
> [d] Huang et al. ``Scallop: From Probabilistic Deductive Databases to Scalable Differentiable Reasoning'' NeurIPS 2021.

---

> > ### Comment · Reviewer_oPB2 · 2024-08-11
> > **Response to the authors**
> >
> > Thank you for clarifying some aspects of your contribution, I don't have further questions at the present time.

---

### Official Review · Reviewer_oGJN · 2024-07-10

**Soundness:** 3
**Presentation:** 4
**Contribution:** 3
**Rating:** 7
**Confidence:** 3

**Summary:**

The paper mainly focuses on the problem of deriving tractability conditions for compositional operations over circuits as to solve a number of queries. By fixing the language of circuits over semirings, the paper generalizes known results and introduce novel ones regarding the tractability of queries solved via algebraic circuits. In addition, it analyzes scenarios where milder structural conditions might be required, when compared to previous results.

**Strengths:**

The results of this paper are part of a series of many theoretical results regarding the tractability of complex operations over circuits. I think the paper positively contributes to the current research area, with a more general understanding of already known results but also new ones. For instance, the exponential separation (Theorem 9) provides an example where lifting the X-first property not needed for computing 2AMC queries on particular semirings might provide an exponential expressiveness advantage. However, note that I lack background regarding the causal probabilistic queries being considered in Section 4.2, and I am not able to fully check the claims regarding their complexity.

I believe the paper is very well written: the notation is very clear, most of the claims are well motivated from the beginning, and there are concise examples across the paper. The proofs in the appendix are also easily readable once having understood the structural properties being used.

The reason of the lower scores related to soundness and contribution are listed in the weaknesses section. However, note that I am somewhat confident that the authors can solve the following issues during the rebuttal phase and therefore potentially improving my score.

**Weaknesses:**

I think that some results requiring particular semirings and homomorphisms require so many properties that make me wonder their actual contribution.
Tractable mapping (Theorem 4) requires the following combination of properties to be both true:
1. $\tau$ is additive OR $C$ is deterministic
2. $\tau$ is multiplicative OR ($\tau$ maps the multiplicative identity AND for each product $\alpha$, one of its input must output zero or one as a constant).
I think that the authors should have stressed the limitations of the given properties on $\tau$. E.g., if $C$ is not deterministic and there are no product units satisfying the above, then which functions $\tau$ are we restricted to in practice? See the question 1 below for more on this.

Theorem 3 claims that one can compute the product of support compatible circuits linearly in the maximum circuit size for both time and memory. However, I think the authors should state that this complexity assumes that (i) we already know the isomorphism between nodes, and (ii) we already know how to "match" the supports of sum nodes in the algorithm. I think that without (ii) one would definitely need linear space and quadratic time instead.

In line L298-L300, the authors claim that for particular semirings and mappings, one could relax the determinism property from the circuit for tractable 2AMC. However, this does not seem to be reflected in table 2, where X-Det is present in all 2AMC queries. I think the authors should either make a milder claim or explicitly show in table 2 a case of such a query.

Looking at Algorithm 2 and 3, I think there are corner cases that are not covered neither by them nor by the proofs. For example, the case of multiplying a sum node and a product node is missing. The case of multiplying input nodes and sum/product nodes is also missing. In fact, these algorithms are way too small when compared to the algorithm of the product showed in [35] (Algorithm 3 in the appendix). I believe the cases must be covered for completeness.

I believe a minor weakness of the presentation of this paper is that it mixes two kinds of theoretical results. It contains some results that leverage algebraic circuits and specific properties regarding semirings and morphisms (e.g., Theorem 4). However, it seems the majority of the other results do not actually need any specific property of the semiring being chosen, and could have been part of a work following the compositional atlas paper on circuits [35]. When reading the paper, I often found myself asking whether the theorems and proof I was looking at were the result of a property of semirings I was missing. This is because of the title and the definition of algebraic circuits is the first definition appearing, and I was expected the rest of the paper being very dependent on it. Instead, it seems many theorems can stand on their own with the usual circuit definition given in [35]. Although minor, I think it would be beneficial if the authors stressed at the beginning that the chosen semiring is not really important for many of the results being shown.

**Questions:**

1. If we restrict  $\tau$ to be both additive and multiplicative homomorphisms, then does this imply it can only be identity function? It looks like it can only be identity function as for Lemma 3.8 in the appendix of [35]. Do the authors agree on this? If yes, then I think it is necessary to refine the conditions required by Theorem 4 as to include these results about the satisfaction of Cauchy functional equations.
2. Following Question 1, are there examples of both additive and multiplicative homomorphisms for some particular choices of source and destination semirings?
3. It seems the support-compatibility definition does not consider how scopes of products are factorized. What is the relationship between support compatibility between circuits and compatibility + determinism? Is structured decomposability PLUS determinism in a circuit more restrictive than support-compatibility with itself?

**Limitations:**

I have discussed the presence of limitations in the weaknesses section.

---

> ### Author Rebuttal · Authors · 2024-08-07
>
> > I think that some results requiring particular semirings and homomorphisms require so many properties that make me wonder their actual contribution. Tractable mapping (Theorem 4) requires the following combination of properties to be both true...
>
> Please see response to Q1/Q2 below.
>
> > Theorem 3 claims that one can compute the product of support compatible circuits linearly in the maximum circuit size for both time and memory...
>
> This is a good point, thanks. We agree and will make this clear in the revised draft.
>
> > In line L298-L300, the authors claim that for particular semirings and mappings, one could relax the determinism property from the circuit for tractable 2AMC. However, this does not seem to be reflected in table 2, where X-Det is present in all 2AMC queries...
>
> By this we mean that one can relax determinism (i.e. $\textbf{V}$-determinism), while $\textbf{X}$-determinism is always required. For example, for the PASP (Max-Credal) query, determinism is not required. We will rephrase to make this clearer.
>
> > Looking at Algorithm 2 and 3, I think there are corner cases that are not covered neither by them nor by the proofs. For example, the case of multiplying a sum node and a product node is missing...
>
> Thanks for pointing this out. For Algorithm 2, we have now added the cases of multiplying an input or product node with a sum node (this corresponds simply to multiplying the input/product with the children of the sum). We have also made minor changes to the proofs for Thm 2 and Thm 5.2 to account for this corner case. Please see the PDF response for the full details. As for multiplying an input with a product, this also seems to be an omission in [35]. We will address this by adding the condition that input nodes do not have scope overlapping with both children of a product to the compatibility definition.
>
> As for Algorithm 3 (support-compatible products), these cases are disallowed by the isomorphism scope condition (i), since for smooth and decomposable circuits, sum nodes have children with the same scope, product nodes have children with disjoint scopes, and input nodes do not have children. We will add this point in the proof of Theorem 3.
>
> > I believe a minor weakness of the presentation of this paper is that it mixes two kinds of theoretical results. It contains some results that leverage algebraic circuits and specific properties regarding semirings and morphisms (e.g., Theorem 4)...
>
> Thank you, this is very helpful feedback presentation-wise. The results for aggregation and products are indeed semiring-agnostic, depending only on the generic semiring $\oplus, \otimes$ and circuit scope/support properties, while the results for elementwise mappings do depend on the specific mapping/semirings. We will clarify this at the start of Section 3.
>
> > **Q1/Q2**: If we restrict to be both additive and multiplicative homomorphisms, then does this imply it can only be identity function?... Following Question 1, are there examples of both additive and multiplicative homomorphisms for some particular choices of source and destination semirings?
>
> As noted in [35], over the probability semiring the only additive and multiplicative homomorphism is the identity function. For other semirings, there are more such functions (known as semiring homomorphisms). For example:
> - Corollary 1 shows that the support mapping $\[\[p\]\]$$_{\mathcal{S} \to \mathcal{S}'}$ always satisfies (Multiplicative), and satisfies (Additive) if (a) no element except $0 _{\mathcal{S}}$ has an additive inverse in $\mathcal{S}$ and (b) $\mathcal{S}'$ is idempotent.
> Examples of such $\mathcal{S}$ include the Boolean, probability and $(\max, \cdot)$ semirings, while examples of $\mathcal{S}'$ include the Boolean, $(\max, \cdot)$, and tropical $(\min, +)$ semirings. The support mapping is a semiring homomorphism in these cases, and is not the identity function unless $\mathcal{S}, \mathcal{S}'$ are both the Boolean semiring.
> - For the semiring $\mathcal{S} = \mathcal{S}' = (\max, \times)$ semiring, any function $f(x) = x^{\beta}$ is a semiring homomorphism.
> - For the tropical semiring $\mathcal{S} = \mathcal{S}' = (\min, +)$, any function $f(x) = c \cdot x$ is a semiring homomorphism.
> - For the rings $\mathcal{S} = \mathbb{Z}$ and $\mathcal{S'} = \mathbb{Z} _{12}$ (the rings of integers, and integers modulo  $12$), then $f(x) = 4 \cdot x \text{ mod } 12$ is a (semi)ring homomorphism.
>
> That said, semiring homomorphisms are a very restricted class of functions and for most of the useful mappings we consider (aside from the support mapping) at least one of the circuit-specific conditions (determinism or the product node condition) will need to hold.
>
> > **Q3** It seems the support-compatibility definition does not consider how scopes of products are factorized. What is the relationship between support compatibility between circuits and compatibility + determinism? Is structured decomposability PLUS determinism in a circuit more restrictive than support-compatibility with itself?
>
> The reason that support-compatibility does not consider factorization of scopes is because of the isomorphism which requires scopes of isomorphic nodes to match; thus the necessary scope factorization is already satisfied. Support-compatibility and compatibility+determinism are different properties and incomparable in general; in particular, support-compatibility does not require compatibility (e.g. any decomposable, smooth, deterministic circuit is compatible with itself), while two circuits can be compatible and both deterministic, but not necessarily share the same "support decomposition". On the other hand, structured decomposability + determinism is strictly stronger than support-compatibility with itself (just decomposability + determinism would suffice).

---

> > ### Comment · Reviewer_oGJN · 2024-08-09
> >
> > Thank you for the very good responses to all my questions.
> > I have decided to raise my overall score to full acceptance.

---

### Official Review · Reviewer_41Ra · 2024-07-11

**Soundness:** 4
**Presentation:** 3
**Contribution:** 2
**Rating:** 5
**Confidence:** 4

**Summary:**

The paper presents sufficient conditions under which certain problems (e.g. 2AMC are tractable when performed on circuits.

**Strengths:**

The paper is generally well-written and technically sound. It also makes a solid effort in trying to unify tractability conditions in the context of algebraic circuits.

**Weaknesses:**

1) My main problem with the paper is that it only discusses sufficient conditions (also acknowledged by the authors). This is constrast to teh probabilistic circuit atlas [1]. This in itself is not a problem, however, the paper does not discuss the problem of eighted model integratio, which has also been formulated as an AMC problem with integration as an extra operation [2]. In general WMI is intractable, however, it has been shown that there exist tractable fragments [3]. If I understand the paper correctly, the tractable fragment of WMI does not fall within the proposed framework. This is quite limiting as the framework does not even cover all tractable problems that can be formulated as 2AMC.

2) Tractable algebraic circuits seem to be highly related to performing tractable inference on first-order circuits. [4] This relationship is not discussed at all.


[1] Vergari, Antonio, et al. "A compositional atlas of tractable circuit operations for probabilistic inference." Advances in Neural Information Processing Systems 34 (2021): 13189-13201.

[2] Dos Martires, Pedro Zuidberg, Anton Dries, and Luc De Raedt. "Exact and approximate weighted model integration with probability density functions using knowledge compilation." Proceedings of the AAAI Conference on Artificial Intelligence. Vol. 33. No. 01. 2019.

[3] Zeng, Zhe, et al. "Probabilistic inference with algebraic constraints: Theoretical limits and practical approximations." Advances in Neural Information Processing Systems 33 (2020): 11564-11575.

[4] Van den Broeck, Guy, et al. "Lifted probabilistic inference by first-order knowledge compilation." IJCAI. 2011.

**Questions:**

Could the authors comment on the two points I raised under "weaknesses".

**Limitations:**

covered.

---

> ### Author Rebuttal · Authors · 2024-08-07
>
> > My main problem with the paper is that it only discusses sufficient conditions (also acknowledged by the authors). This is contrast to the probabilistic circuit atlas [1].
>
> Although we only discuss sufficient conditions, for Table 1, the necessity of these conditions in general follows from the results in the PC atlas, as those are special cases in the probabilistic semiring. The point we are trying to make is that, inevitably, even though the conditions are necessary in general, they might not be for a specific semiring (and a specific elementwise mapping).
>
> For the specific compositional queries in Table 2, we show hardness for 2AMC without these conditions in Theorem 7, while hardness results for the causal inference queries are given in [36].
>
> > This in itself is not a problem, however, the paper does not discuss the problem of eighted model integratio, which has also been formulated as an AMC problem with integration as an extra operation [2]. In general WMI is intractable, however, it has been shown that there exist tractable fragments [3]. If I understand the paper correctly, the tractable fragment of WMI does not fall within the proposed framework. This is quite limiting as the framework does not even cover all tractable problems that can be formulated as 2AMC.
>
> Thank you for bringing up these works on WMI. We will add these references and a discussion to our related work section.
>
> In [2], the authors propose to solve hybrid WMI problems by (i) performing AMC over the probability density semiring; and (ii) performing symbolic computation of the resulting expression. The first step (i) is captured by AMC, and thus in our framework (cf. lines 255-262), showing that d-DNNF suffices for valid evaluation (aggregation). We note that the integration step (ii) here uses a symbolic inference engine (or sampler), as opposed to being part of the AMC itself.
>
> [3] shows certain fragments of WMI are tractable through a message-passing algorithm; however, it is unclear if and how the WMI problems being considered could be represented as an AMC problem on circuits, and how the message-passing algorithm would translate to circuits. Our work concerns compositional inference problems (including AMC/2AMC) on circuit representations, and in particular it encompasses all AMC/2AMC problems on circuits, as we show in Theorems 7 and 8.
>
> > Tractable algebraic circuits seem to be highly related to performing tractable inference on first-order circuits. [4] This relationship is not discussed at all.
>
> Our work focuses on circuits where the nodes are algebraic expressions (e.g. propositional formulae in NNFs); in particular, the expressions are not defined over a domain as in first-order circuits. We would be happy to clarify any specific questions the reviewer has about the relationship of our work with this paper.

---

> > ### Comment · Reviewer_41Ra · 2024-08-11
> >
> > **On WMI and AMC**
> > Indeed the method by Zeng et al. solves WMI via a message passing scheme. As it is well-known that message passing is essentially performs a series of sum product evaluations this can be cast as an AMC problem, which was done by Zuidberg et al. Given these two works it should hence be possible to cast WMI as an inference problem in the AMC atlas.
> >
> > **On first order circuits**
> > You state that "you work focuses on propositoinal formulas". However, if you had pure propositional formulas you would not have any variables to perform the extra operation over. I.e. the max in a marginal map problem. In this regard you are not dealing with propositional formulas but with (a fragment of) weighted first-order logic.
> >
> > **In conclusion**
> >
> > I still believe the connections to WMI circuits and FO logic circuits should be studied more carefully if one were to claim an atlas of algebraic circuits.

---

> > > ### Author Response · Authors · 2024-08-12
> > >
> > > Thank you for the discussion.
> > >
> > > **On WMI and AMC** It is true that standard message-passing on discrete graphical models can be easily translated to circuits. However, the algorithm of Zeng et al. for WMI is different; for instance, integration is not even efficient if the diameter of the tree is large.
> > >
> > > As far as we are aware, a circuit interpretation of their algorithm does not currently exist, and would involve novel research. In particular, in Zuidberg et al., the *logical* (Boolean) part of the WMI problem is compiled to a circuit (which, as an AMC problem, is captured in our atlas), and has the standard exponential-in-treewidth complexity; while the *integration* uses an external symbolic integration engine (PSI-Solver).
> > >
> > > **On first-order circuits** We think the reviewer may be confusing propositional variables, of which we have many, with logical variables, of which we have none. Logical variables, when instantiated as arguments to predicates, give rise to propositional variables in the form of atoms in the work of Van den Broeck et al. Our work has no predicates and no logical variables, and is therefore only tangentially related. Our semiring operations, such as max, are over propositional variables, not logical variables.
> > >
> > > In weighted first-order model counting as defined in Van den Broeck et al., one sums a *product of weights* over *grounded predicates satisfying the first-order formula*. In contrast, in algebraic circuits, we sum/aggregate a *function* over *assignments of the propositional variables*.

---

### Official Review · Reviewer_x2Dq · 2024-07-12

**Soundness:** 3
**Presentation:** 3
**Contribution:** 3
**Rating:** 6
**Confidence:** 3

**Summary:**

They investigate algebraic circuits on semi-rings (with sums and products).
They give criterions that allow efficient combinations of circuits and aggregation of the variables of one circuit (e.g. the sum over all inputs of the circuit).
They give algorithms and hardness results for Algebraic Model Counting.

**Strengths:**

they give an unified view on different important problems which I did not expect to be related

foundamental research

efficient algorithms

well-written

**Weaknesses:**

a lot of technical definitions

the conditions under which the algorithms are efficiently applicable seem rather restrictive

**Questions:**

what is an "atlas"? the word is never used in the text


the classic circuit classes have an uniformity constraint, on the complexity of an algorithm that computes the circuit for a certain input length.
do these algebraic circuits not need uniformity ?


>Theorem 2
why not just add one new product node as root node, and then make both C and C' children of this new node.
it is a much simpler construction and would also compute the product.

**Limitations:**

yes

---

> ### Author Rebuttal · Authors · 2024-08-07
>
> >the conditions under which the algorithms are efficiently applicable seem rather restrictive
>
> We agree that the sufficient conditions for tractability of many compositional queries can be strong. However, part of our contribution is that our framework is able to derive weaker tractability conditions than were previously known. For example, we show that $\textbf{X}$-firstness is not necessary for probabilistic answer-set programming inference (under the max-credal or max-ent semantics), and that this can lead to exponentially smaller circuits (Thm 9).
>
> Despite the high complexity of these problems, there already exist tools for compiling or learning circuits satisfying these properties (e.g. PySDD [a] and PSDD [b, c] for $\textbf{X}$-firstness and $\textbf{X}$-determinism, MDNets [36] for $\textbf{X}$-determinism). We believe that our work should motivate further study on compilation and learning algorithms for circuits satisfying these properties, given the generality of these properties across multiple inference tasks. j
>
> [a] Meert \& Choi (2017) *PySDD.* In Recent Trends in Knowledge Compilation, Report from Dagstuhl Seminar 17381, Sep 2017.
>
> [b] Kisa et al. (2014) *Probabilistic Sentential Decision Diagrams.* In KR.
>
> [c] Liang et al. (2017) *Learning the structure of probabilistic sentential decision diagrams.* In UAI.
>
> >what is an "atlas"? the word is never used in the text
>
> We use "atlas" to refer to the collection of operations and corresponding tractability conditions we derive (Table 1); it also refers to a closely related recent work [35] that considered compositional inference for probabilistic circuits.
>
> > the classic circuit classes have an uniformity constraint, on the complexity of an algorithm that computes the circuit for a certain input length. do these algebraic circuits not need uniformity ?
>
> Indeed, for Theorem 9, we use a family of algebraic circuits $C_1, ..., C_n$ to specify functions, such that the smallest $\textbf{X}$-first circuits have an exponential lower bound on size. This family is indeed logspace-uniform; note however that we don't make any claims in terms of membership of languages in complexity classes, merely on the size of the circuits computing the function. We will make this point clearer in the revised version.
>
> > Theorem 2 why not just add one new product node as root node, and then make both C and C' children of this new node. it is a much simpler construction and would also compute the product.
>
> This is possible, but the resulting circuit would not be decomposable if $C$ and $C'$ share variables. As the queries we (and the majority of other works in the knowledge compilation/tractable model literature) consider require aggregation subsequent to products, we restrict to algorithms producing decomposable and smooth circuits.

---

### Author Rebuttal · Authors · 2024-08-07

We would like to thank all the reviewers for their time and effort spent reviewing our paper, and for their helpful feedback and comments. Please find individual responses to each reviewer below.

We attach a PDF here addressing corner cases in Algorithm 2 mentioned by Reviewer oGJN.

---

### Decision · Program_Chairs · 2024-09-25

**Decision:**

Accept (poster)

**Comment:**

The paper provides a unifying framework for algebraic circuits using compositional operation and claims to show new tractable classes for various probabilistic queries.  This is done by extending circuits to semi-rings.

The reviewers comments say that the strength is in the technical work. Yet the weakness is that it is not clear what the practical contribution is. It is not clear if the tractability conditions can be satisfied by any real circuits. Also, there are complaints that there is no clear separation between earlier and new contributions. There was a discussion with the authors and reviewers seemed to accept the clarifications provided.

There is a very large literature missing here on semiring, by Bistareli et. al: in the area of constraint and probabilistic reasoning by

Semiring-based constraint satisfaction and optimization
Authors: Stefano Bistarelli, Ugo Montanari, Francesca Rossi
Journal of the ACM (JACM), Volume 44, Issue 2
Pages 201 - 236
https://doi.org/10.1145/256303.256306